# Spontaneous behaviour is structured by reinforcement without explicit reward

Jeffrey E. Markowitz[1,7,8], Winthrop F. Gillis[1,8], Maya Jay[1,8], Jeffrey Wood[1], Ryley W. Harris[1], Robert Cieszkowski[1], Rebecca Scott[1], David Brann[1], Dorothy Koveal[1], Tomasz Kula[1], Caleb Weinreb[1], Mohammed Abdal Monium Osman[1], Sandra Romero Pinto[2,3], Naoshige Uchida[2,3], Scott W. Linderman[4,5], Bernardo L. Sabatini[1,6] & Sandeep Robert Datta[1✉]

Spontaneous animal behaviour is built from action modules that are concatenated by the brain into sequences[1,2]. However, the neural mechanisms that guide the composition of naturalistic, self-motivated behaviour remain unknown. Here we show that dopamine systematically fluctuates in the dorsolateral striatum (DLS) as mice spontaneously express sub-second behavioural modules, despite the absence of task structure, sensory cues or exogenous reward. Photometric recordings and calibrated closed-loop optogenetic manipulations during open field behaviour demonstrate that DLS dopamine fluctuations increase sequence variation over seconds, reinforce the use of associated behavioural modules over minutes, and modulate the vigour with which modules are expressed, without directly influencing movement initiation or moment-to-moment kinematics. Although the reinforcing effects of optogenetic DLS dopamine manipulations vary across behavioural modules and individual mice, these differences are well predicted by observed variation in the relationships between endogenous dopamine and module use. Consistent with the possibility that DLS dopamine fluctuations act as a teaching signal, mice build sequences during exploration as if to maximize dopamine. Together, these findings suggest a model in which the same circuits and computations that govern action choices in structured tasks have a key role in sculpting the content of unconstrained, high-dimensional, spontaneous behaviour.

Spontaneous behaviour exhibits structure. Ethologists have long argued that the self-motivated behaviour of animals in the wild is flexibly built from modular components that are linked together over time in a predictable yet probabilistic manner[1]. Many well-studied laboratory behaviours—including chemotaxis, grooming, prey seeking, courtship, birdsong and exploratory locomotion—are similarly characterized by modularity and predictability[2–5]. However, it remains unclear how the brain regulates the expression of individual behavioural modules at any given moment, or how it dynamically composes these modules into the fluid behaviours observed when animals act of their own volition in the absence of experimental restraint, task structure or exogenous reward.

Given that the loss of dopaminergic neurons from the substantia nigra pars compacta (SNc) causes diffuse deficits in action initiation and sequencing, it is likely that the neuromodulator dopamine influences the architecture of spontaneous behaviour[6–8]. Yet we know little about the precise relationship between dopamine and behaviour when animals freely explore an environment. Although dopamine is thought to motivate spontaneous behaviour and to influence the vigour with which actions are expressed, evidence is mixed as to whether phasic dopamine transients are permissive or causal for movements, whether dopamine rises or falls when animals initiate a movement, and whether

dopamine fluctuations specify movement kinematics in freely behaving animals[6,9–19]. By contrast, during structured tasks in which animals seek explicit and often cued rewards, phasic dopamine clearly conveys information related to reward and reward-prediction errors, reinforces reward-associated actions, and influences the choices made between alternative actions[20–25].

Dopamine may have distinct roles during spontaneous and task-structured behaviours, given the many ways in which they differ; for example, spontaneous behaviours generally exhibit a greater variety of expressed behavioural modules, include more complex behavioural sequences, and tend to emphasize self-initiated movements associated with active sensing[2,4,26]. Nevertheless, both spontaneous behaviour and structured tasks demand that animals choose actions on an ongoing basis from a distribution of possibilities, suggesting that dopamine may influence the continuous assembly of naturalistic sequences through mechanisms similar to those used to support goal-driven action selection in response to rewards.

To test this hypothesis, here we characterize mouse spontaneous behaviour using motion sequencing (MoSeq)—which uses 3D imaging and unsupervised machine learning to atomize behaviour into sub-second modules referred to as 'syllables'—while simultaneously

[1]Department of Neurobiology, Harvard Medical School, Boston, MA, USA. [2]Department of Molecular and Cellular Biology, Harvard University, Cambridge, MA, USA. [3]Center for Brain Science, Harvard University, Cambridge, MA, USA. [4]Wu Tsai Neurosciences Institute, Stanford University, Stanford, CA, USA. [5]Department of Statistics, Stanford University, Stanford, CA, USA. [6]Howard Hughes Medical Institute, Chevy Chase, MD, USA. [7]Present address: Wallace H. Coulter Department of Biomedical Engineering, Georgia Institute of Technology and Emory University, Atlanta, GA, USA. [8]These authors contributed equally: Jeffrey E. Markowitz, Winthrop F. Gillis, Maya Jay. ✉e-mail: Srdatta@hms.harvard.edu

assessing and manipulating dopamine transients in DLS, a region of the basal ganglia implicated in the composition of natural behaviours[27–29]. As SNc dopaminergic neurons acutely influence the population activity of DLS spiny projection neurons (SPNs) and induce plasticity in corticostriatal synapses[30], DLS dopamine fluctuations may be particularly relevant to syllable expression and/or sequencing.

We find that DLS dopamine systematically fluctuates during the expression of behavioural syllables, and—through calibrated closed-loop manipulations of DLS dopamine—demonstrate that these fluctuations are causally related to syllable usage, sequencing and vigour. A simple computational model in which syllable-associated dopamine transients shape sequence composition effectively reproduces the observed behavioural choices made by mice during spontaneous behaviour. These results reveal that DLS dopamine transients act as a continuous teaching signal, one that affords spontaneous behaviour its moment-to-moment structure; our observations further suggest a broad model in which the composition of spontaneous behaviour from elemental components is supported by the same computations and circuits that govern action choices in more structured tasks.

### Relating DLS dopamine to spontaneous behavior

To characterize the relationship between striatal dopamine release and spontaneous behaviour, we virally expressed the dopamine reporter dLight1.1 in DLS neurons, and then assessed dopamine fluctuations via photometry as mice explored a featureless open field in the dark[31]. In concert with these neural measurements, we both quantified conventional movement parameters and performed MoSeq[4] (Fig. 1a–d and Extended Data Fig. 1). In this setting, MoSeq identified 37 commonly used behavioural syllables, whose median duration was 400 ± 636 ms, ranging from pause syllables in which mice adopted different static poses, to dynamic syllables in which mice reared, groomed or explored (Extended Data Fig. 2a–c). Syllables varied in terms of how often they were used and the order in which they occurred as time unfolded during each experiment (Extended Data Fig. 2b,d–g).

Photometry revealed pervasive, fast dopamine fluctuations in DLS that occurred throughout each experiment regardless of whether the mouse was actively moving or relatively still (average rise time during spontaneous behaviour = 67 ms, decay time = 100 ms; Fig. 1d and Extended Data Fig. 1e,h). These dopamine fluctuations (measured as both dLight transient rates and average binned amplitudes) were only weakly correlated with many aspects of movement kinematics (for example, turning, rearing and acceleration) but were significantly negatively correlated with translational velocity at short timescales (<10 s); this correlation reversed and became positive at longer timescales, consistent with the idea that dopamine broadly invigorates and motivates movement[6,9,17,32] (Fig. 1e). We validated the relationship between translational velocity and dopamine transients using 3D keypoint tracking, which also demonstrated that forelimb movement per se (that is, independent of translation) only negligibly correlated with dopamine transients (Extended Data Fig. 3).

As has been observed previously, dopamine systematically fluctuated when mice initiated a movement after pausing in the arena[9] (Fig. 1f). However, we also observed that dopamine fluctuated as mice transitioned from one behavioural syllable into the next. These fluctuations exhibited a characteristic dip-then-peak pattern surrounding each syllable transition; as dLight fluorescence changes lag dopamine release by tens of milliseconds[31] (Extended Data Fig. 1e), it is likely that the observed dopamine dip occurs at the end of the previous syllable, and the peak in dopamine occurs during expression of the subsequent syllable (Fig. 1f,g). Consistent with this hypothesis, time-warping the dopamine trace to accommodate differences in syllable duration revealed that on average, dopamine peaked near the middle of each syllable and decayed as the syllable ended, reminiscent of the 'burst–pause' firing pattern of dopamine neurons previously observed at movement initiations[10,11,14,33] (Extended Data Fig. 4a). Nearly every syllable instance

was accompanied by a positive dopamine transient, whose amplitude varied (Extended Data Fig. 4b–e).

### DLS dopamine does not predict syllable identity

Syllable-specific dopamine waveforms exhibited relatively stereotyped shapes and amplitudes (when averaged either within or across mice and experiments), suggesting that dopamine waveforms predict either the identity of the associated syllable or the kinematics of movement associated with its expression (Fig. 1h and Extended Data Fig. 4c–h). However, kinematically similar syllables (for example, two rears) could exhibit very different average dLight waveforms, whereas kinematically different syllables (for example, turning and investigating) could exhibit similar average waveforms (Fig. 1i). Aggregating different syllables into categories (such as rearing, grooming or diving) revealed that different categories of behaviour exhibited broadly overlapping average dopamine transient amplitudes (Extended Data Fig. 5a). Furthermore, syllable-associated dopamine transient amplitudes only weakly correlated with the movements actually expressed during a given syllable (Fig 1i,j and Extended Data Fig. 5b).

Consistent with a potential dissociation between DLS dopamine and syllable-related kinematics, dopamine waveforms often changed shape and amplitude across different instances of the same syllable (Fig. 1k). This was true even for syllables known to correlate with high SNc and striatal activity such as contralateral turns[20,34,35] (Extended Data Fig. 5c). Indeed, random forest classifiers were unable to predict which syllable was being expressed by the mouse at any moment based on the coincident dLight waveform amplitudes or shapes[9] (Fig. 1l). Syllable-associated dopamine transient amplitudes were also unrelated to many other features of behaviour, including the rendition quality of each syllable, differences in speed between syllables, the biomechanical difficulty of transitioning into a given syllable, the position of the mouse in the arena, and the specific identity of the preceding syllable (Extended Data Fig. 5d–i). Thus, although dopamine systematically fluctuates during the expression of behavioural syllables and syllables are associated on average with different dopamine waveform shapes and amplitudes, individual dopamine transients do not appear to reliably encode information about syllable identity, kinematics or related features of syllable-associated behaviour.

### Dopamine predicts future syllable use and sequencing

We therefore considered whether DLS dopamine transients instead have a more flexible role in specifying ongoing patterns of syllable usage, given that the usage of specific syllables and sequences varies during the course of each experiment (Extended Data Fig. 2e–g). Notably, dopamine transient amplitudes observed during spontaneous behaviour roughly matched those observed while mice consumed an unexpected (and presumably rewarding) chocolate chip placed in the open field; this finding suggests that syllable-associated dopamine transients occurring during self-initiated behaviour, even in the absence of reward, may reinforce the expression of associated syllables (Fig. 2a).

Consistent with this hypothesis, those syllables that were on average associated with more DLS dopamine during a given experiment tended to be used more during that experiment; furthermore, variation in the average level of DLS dopamine associated with a given syllable across experiments predicted variation in the use of that same syllable (relative to others) across experiments (Fig. 2b). Dopamine transient amplitudes also predicted the near-term use of associated behavioural syllables: if a given instance of a syllable coincided with a relatively high amplitude dopamine transient, that syllable was then used more over the next several minutes, whereas if an instance of that same syllable coincided with a relatively low amplitude transient it was used less (Fig. 2c–g). The relationship between dopamine transient amplitudes and syllable usage was not due to autocorrelation in the dLight signal or to correlations between dopamine and velocity, both of which declined

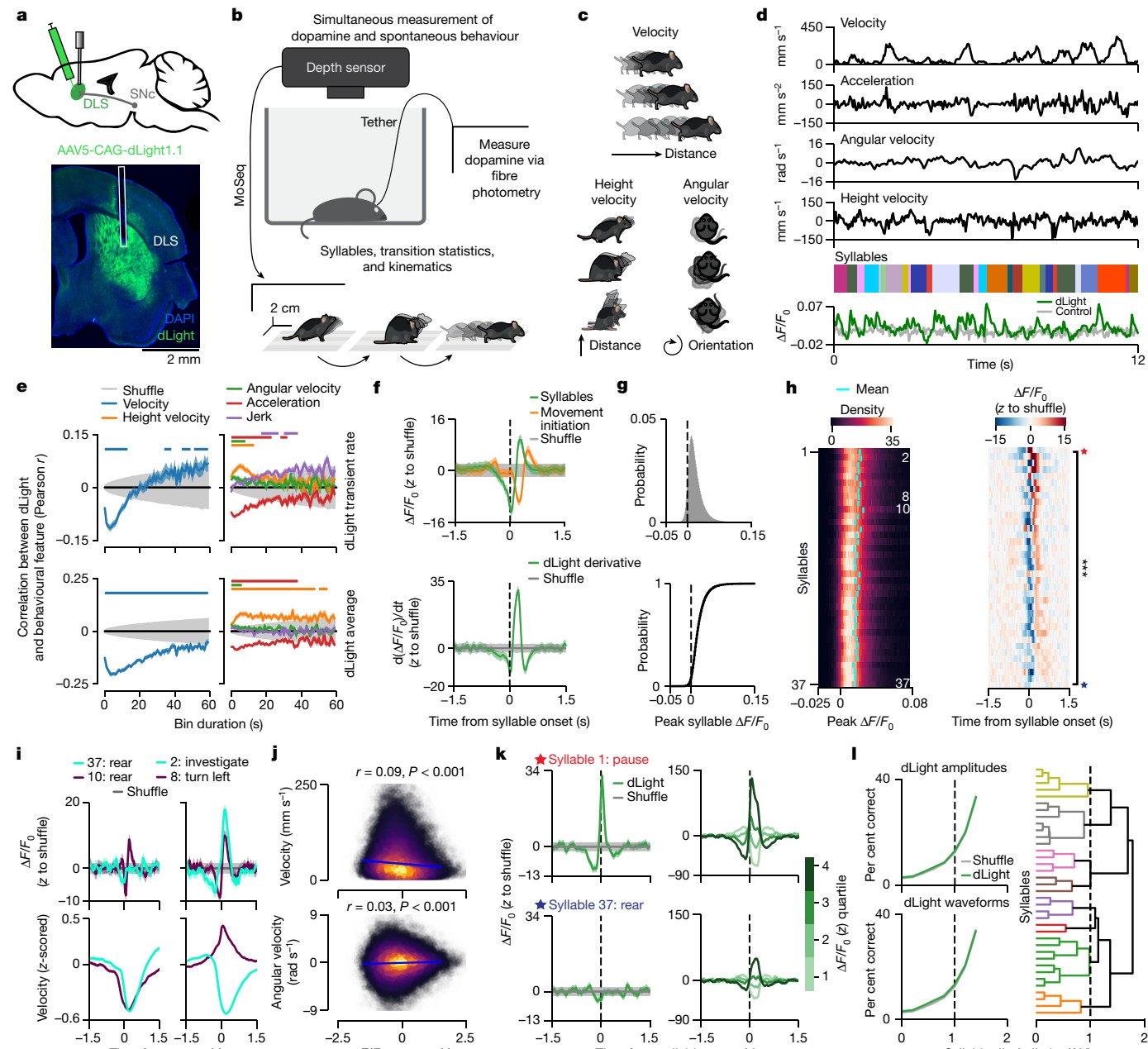

**Fig. 1 | Behaviour is associated with dopamine transients in DLS. a**, dLight expression and fibre placement in DLS (Methods). **b**, The behavioural characterization pipeline using MoSeq ($n = 14$ mice for MoSeq, 216 experiments; Methods). **c**, Examples of measured kinematic variables. **d**, Aligned kinematic variables, MoSeq syllables and dLight fluorescence from an example experiment. **e**, Top, average correlation between kinematic variables and dLight transient rate. Bottom, correlations with dLight fluorescence. Coloured shading denotes bootstrapped s.e.m.; grey shading indicates the 95% shuffle confidence interval. Solid bars indicate statistical significance at $P < 0.05$ (shuffle test; Methods). **f**, Top, average fluorescence ($z$-scored to shuffle; Extended Data Fig. 4c and Methods) aligned to movement initiation or syllable onsets ($n = 100$ shuffles). The average syllable-associated dLight transient exceeds that associated with movement initiation ($P = 0.0006$, $z = 3$, effect size $r = 0.8$, two-sided Wilcoxon signed-rank test; $n = 14$ averages). Bottom, derivative of top panel. Green shading represents the 95% bootstrap confidence interval; grey shading represents the 95% shuffle confidence interval. **g**, The distribution of all syllable-associated dLight peaks. Bottom, the cumulative distribution. **h**, Left, the distribution of syllable-associated dLight peaks for across all experiments. Right, $z$-scored average syllable-associated waveforms, sorted by peak fluorescence. The blue and red stars indicate the syllable waveforms

shown in **k**. ***, Kruskal–Wallis $H$ test on average syllable-associated fluorescence amplitudes: $P < 10^{-25}$, $H = 209.29$, $n = 518$ mouse–syllable pairs. $y$-axis syllable sorting is shared across panels. **i**, Left, example syllables with different average (across experiments) waveforms (top) but similar velocity (bottom). Right, example syllables with similar waveforms (top) but different velocities (bottom). Shading represents the 95% confidence interval. **j**, Robust linear regression between syllable-associated dLight and velocity (top) or angular velocity (bottom; Methods). Each point is a sampled syllable instance ($n = 28,000$ points; $n = 2,000$ points per syllable drawn randomly from each mouse). Regression line (shading indicates the 95% bootstrap confidence interval) and kernel density estimate are shown. $P$-values estimated by shuffle test. **k**, Left, average syllable-associated waveforms for starred syllables in **h** (right). Coloured shading represents the 95% bootstrap confidence interval; grey shading represents the 95% shuffle confidence interval. Right, syllable-associated waveforms from the left panel binned into quartiles of peak magnitudes. **l**, Left, held-out classifier performance predicting syllable identity from syllable-associated dLight peak amplitudes (top) or waveforms (bottom). Right, dendrogram showing syllables organized by MoSeq distance (Methods). AU, arbitrary units.

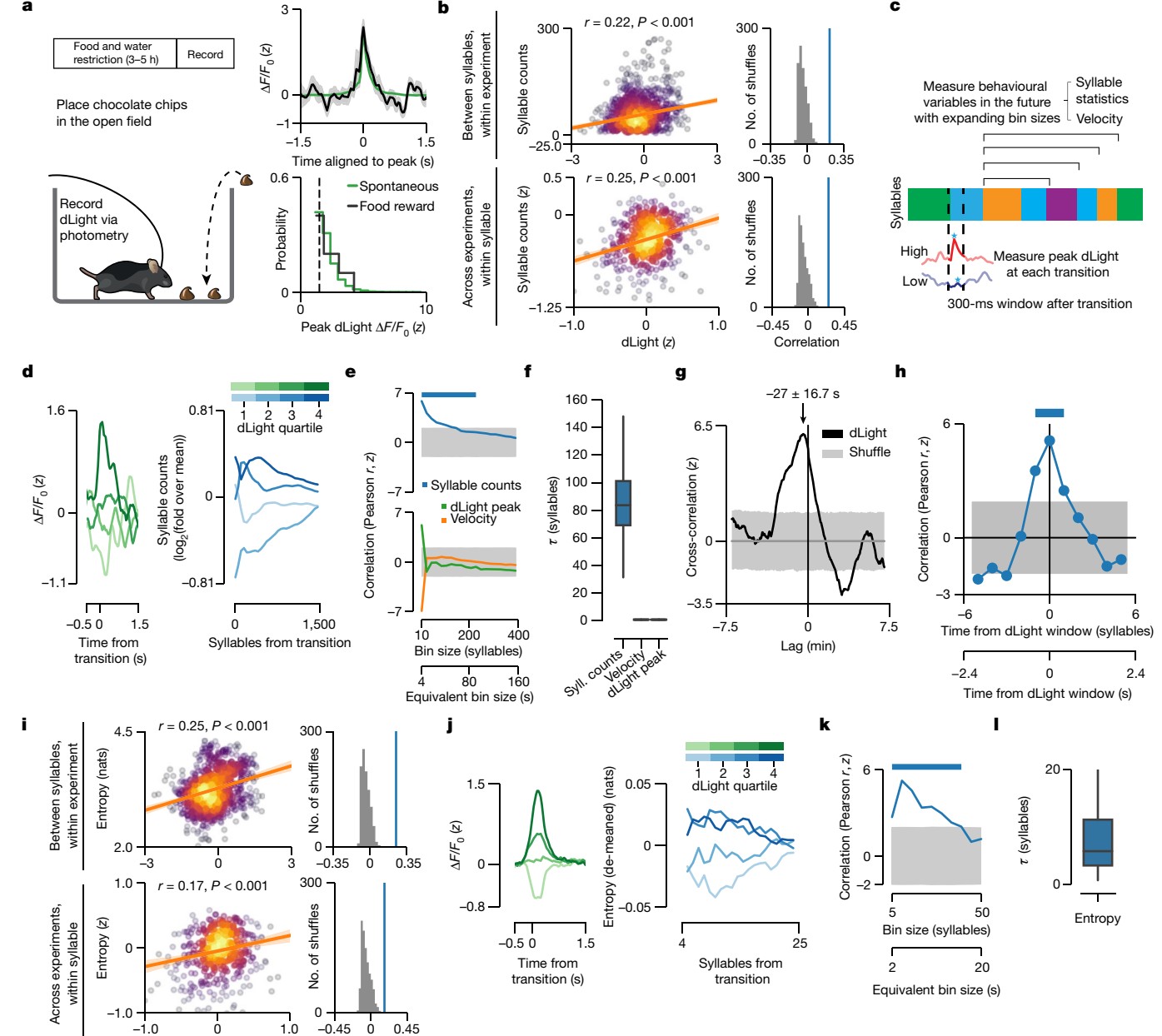

**Fig. 2 | Endogenous dopamine transients predict average syllable use and sequence variability. a**, The unexpected food reward protocol. Top right, average spontaneous versus food reward-associated transients (Methods). Shading represents bootstrap s.e.m. Bottom right, probability density function of dLight transient amplitudes. The dotted line indicates the threshold for detecting dLight transient peaks. **b**, Left, robust linear regression between syllable-associated dLight and average syllable counts per syllable (each dot is a syllable–mouse pair). Regression line and kernel density estimate are shown ($r$ is Pearson correlation between held-out predictions and actual data). Right, the distribution of Pearson correlations using models fit to shuffled data compared with the observed correlation (blue line). Shading indicates the 95% bootstrap confidence interval. $P$-values estimated by one-sided shuffle test. **c**, Schematic depicting the hypothesis that dopamine predicts changes in future behaviour. Blue star indicates the syllable-associated dopamine peak. **d**, Left, average dLight waveforms for each fluorescence quartile at syllable onset for an example syllable–experiment pair. Right, $\log_2$ fold change compared to average syllable counts after example syllable onset computed over increasing bin sizes (in syllables) after onset. **e**, The average Pearson correlation between syllable-associated dLight and syllable counts or velocity, and the dLight signal autocorrelation, computed using a set of increasing bin

sizes after syllable onset. Grey shading represents the shuffled 95% confidence interval. The two $x$-axes reflect time in syllables and approximated in seconds. Solid bars indicate statistical significance ($P < 0.05$, one-sided shuffle test). **f**, The distributions of exponential decay timescales ($\tau$) for the correlations plotted in Fig. 2e ($n = 1,000$ bootstrap samples). In all box plots in this Article, the horizontal line represents the median, box edges delineate the first and third quartiles, and whiskers include the furthest data point within 1.5 times the interquartile range of the first or third quartile. **g**, Average cross-correlation between binned syllable counts and syllable-associated dLight fluorescence (from all mice and experiments) across lags ($P < 0.001$, one-sided shuffle test; the arrow indicates average peak lag, error is 68% confidence interval). Grey shading represents the shuffled 95% confidence interval. **h**, Overall correlation between syllable-associated dLight and syllable usage for syllables temporally adjacent to the index syllable. Grey shading represents the shuffled 95% confidence interval. The solid bar denotes statistical significance ($P < 0.05$, one-sided shuffle test). **i**, As in **b**, but for average entropy per syllable. Nat, natural unit of information. **j**, As in **d**, but for sequence entropy for an example syllable–experiment pair. **k**, As in **e**, but for sequence entropy. **l**, Fitted $\tau$ values for the correlation curve in **k** ($n = 1,000$ bootstrap samples).

sharply after a few hundred milliseconds (Fig. 2e). Furthermore, the possible consequences of dopamine transients were largely restricted to the specific syllable with which they were associated, as the size of a given dopamine transient did not substantially correlate with the use of syllables neighbouring in time (Fig. 2h).

Each syllable expressed during spontaneous behaviour is associated with a specific set of possible subsequent syllables whose likelihoods vary substantially—some subsequent syllables are very likely, and therefore contribute to creating predictable (that is, more deterministic) behavioural sequences, whereas others are much less likely and therefore participate in less predictable (that is, more probabilistic) behavioural sequences (Extended Data Fig. 2d). Given that treatment with dopaminergic agonists has been shown to increase the variability of ongoing behavioural sequences[36–38], we tested whether syllable-associated dopamine transients can influence the choice of next expressed syllable. Examining the transition patterns between syllables revealed that mice tended to string together more deterministic behavioural sequences when average syllable-associated dopamine was relatively low rather than high (Fig. 2i and Extended Data Fig. 6a). Furthermore, syllable-associated dopamine levels correlated with sequence predictability on a moment-to-moment basis: if a given instance of a syllable was associated with a relatively high amplitude dopamine transient, the mouse tended to make less predictable syllable choices over the next several seconds, whereas if that same syllable was associated with a relatively low amplitude transient, syllable sequences were more deterministic in the near future (Fig. 2j–l).

Both syllable usage and sequence variability were themselves correlated: those syllables that were used the most also tended to participate in the most variable behavioural sequences (Extended Data Fig. 6a). However, syllable usage and sequence variability contributed independently to the ability of an encoding model to predict dopamine fluctuations during behaviour (Extended Data Fig. 6b–g; Methods). Our findings regarding syllable usage and sequence variability were specific to DLS, as dLight recordings in dorsomedial striatum (DMS) revealed less frequent dopamine transients that do not predict future syllable usage (Extended Data Fig. 7). However, dLight signals were effectively predicted by velocity in both DMS and DLS, consistent with previous findings that dopamine fluctuations in dorsal striatum correlate with movement[25] (Extended Data Fig. 7e). Taken together, these findings suggest that moment-to-moment fluctuations in DLS dopamine influence the usage of associated behavioural syllables over timescales of minutes, and the choice of what to do next—and thus the predictability of behavioural sequences—on timescales of seconds.

## Syllable-associated Opto-DA influences behaviour

To test directly whether syllable-associated DLS dopamine is sufficient to drive increases in syllable usage and sequence variability—and to determine whether fast dopamine fluctuations also influence movements per se—we built a platform that enables us to trigger an optogenetic pulse during the expression of a specific, targeted syllable (Methods and Extended Data Fig. 8a–e). We used this approach to manipulate syllable-associated phasic dopamine levels (Opto-DA) by optically stimulating dopamine axons in the DLS in mice expressing channelrhodopsin-2 (ChR2) in all dopamine-releasing neurons in the midbrain (Fig. 3a,b; Methods). Our stimulation protocol was calibrated to mimic typical syllable-associated dLight amplitudes observed during spontaneous behaviour (Fig. 3c; Methods). We assessed spontaneous mouse behaviour before, during and after syllable-specific stimulation; in each mouse we serially repeated this stimulation protocol for six different target syllables, which collectively exhibited substantial variability in syllable kinematics, usage and sequencing (Fig. 3d and Extended Data Fig. 9a–e). In nearly all instances, optogenetic stimulation was limited to the targeted syllable and did not extend into subsequent syllables (Extended Data Fig. 9f).

Pairing syllable expression with Opto-DA rapidly increased the use of target syllables, evoking a stable increase in syllable usage per unit time (rather than continuously increasing the rate of syllable use) (Fig. 3e–g and Extended Data Fig. 9g,h). This increased usage persisted in experiments after stimulation was terminated, demonstrating that mice learned an association between dopamine stimulation and the specific targeted syllable (Fig. 3h). The reinforcing effects of Opto-DA were specific to the targeted syllable, as non-target syllables were not affected (Extended Data Figs. 8f and 10). Thus, Opto-DA is sufficient to reinforce the expression of targeted behavioural syllables in the absence of task structure or external sensory cues, suggesting that mice can recognize their own movements on a moment-to-moment basis and use this information to upregulate actions that are associated with exogenous dopamine[39].

Consistent with the observed correlations between endogenous dopamine fluctuations and sequence variability, behavioural sequences observed immediately after Opto-DA stimulation were more unpredictable than the those observed on catch trials (Fig. 3i). Of note, this increase in sequence variability was transient and only apparent during Opto-DA sessions; in the two subsequent sessions—after optogenetic stimulation had ceased but during which target syllable use remained upregulated—behavioural sequences surrounding the target syllable became more predictable, as the most likely transitions into and out of the target became even more likely (Fig. 3j,k). Thus Opto-DA increases sequence variability over seconds-long timescales, whereas sequence variability decreases over the longer timescales at which Opto-DA supports syllable reinforcement.

We also considered whether pairing Opto-DA with specific syllables changed the vigour with syllables were expressed, given prior experiments demonstrating that dopamine can control the speed (that is, vigour) of future movements by reinforcing the expression of fast (or slow) versions of a given movement, or the pitch of a note in a zebra finch's song[40–43]. To address whether Opto-DA influences syllable-associated vigour, we tailored our Opto-DA experiment such that optogenetic stimulation was delivered only on those target syllable instances in which syllable speed was in the top quartile of its overall distribution; this manipulation systematically increased the velocity with which the target syllable was later expressed (Extended Data Fig. 10d). Conversely, Opto-DA during the slowest quartile of the syllable velocity distribution caused future instances of the targeted syllable to slow down.

In contrast to its dynamic effects on syllable usage, sequence variability and syllable vigour, Opto-DA did not prompt switching from one syllable to the next, alter movement parameters associated with the targeted syllable, change movement during the stimulation experiment in general, or induce a preference for a spatial location in the arena (Fig. 3l and Extended Data Fig. 10e–k). However, extending the duration of optogenetic stimulation to several seconds (which caused dLight signals that are of substantially higher amplitude than typically observed during spontaneous behaviour) caused mice to increase their velocity, consistent with previous reports demonstrating that strong perturbations of dopamine are sufficient to cause movements[9,10,14,20] (Fig. 3m and Extended Data Fig. 10l).

## Dopamine–behaviour relationships predict learning

Only a fraction of mice successfully associated Opto-DA with targeted syllables, and among those that learned this association, the degree of learning varied (Fig. 3e). We wondered whether this distribution in learning reflected differences in the sensitivity of individual mice to dopamine (Fig. 4a). Consistent with this possibility, the ability of endogenous, syllable-associated dopamine transients to support syllable use and induce sequence variability was similar within an individual mouse but differed across mice (Fig. 4b). Furthermore, mice that were strongly influenced by endogenous dopamine fluctuations were also those that were particularly avid learners of the association between Opto-DA and

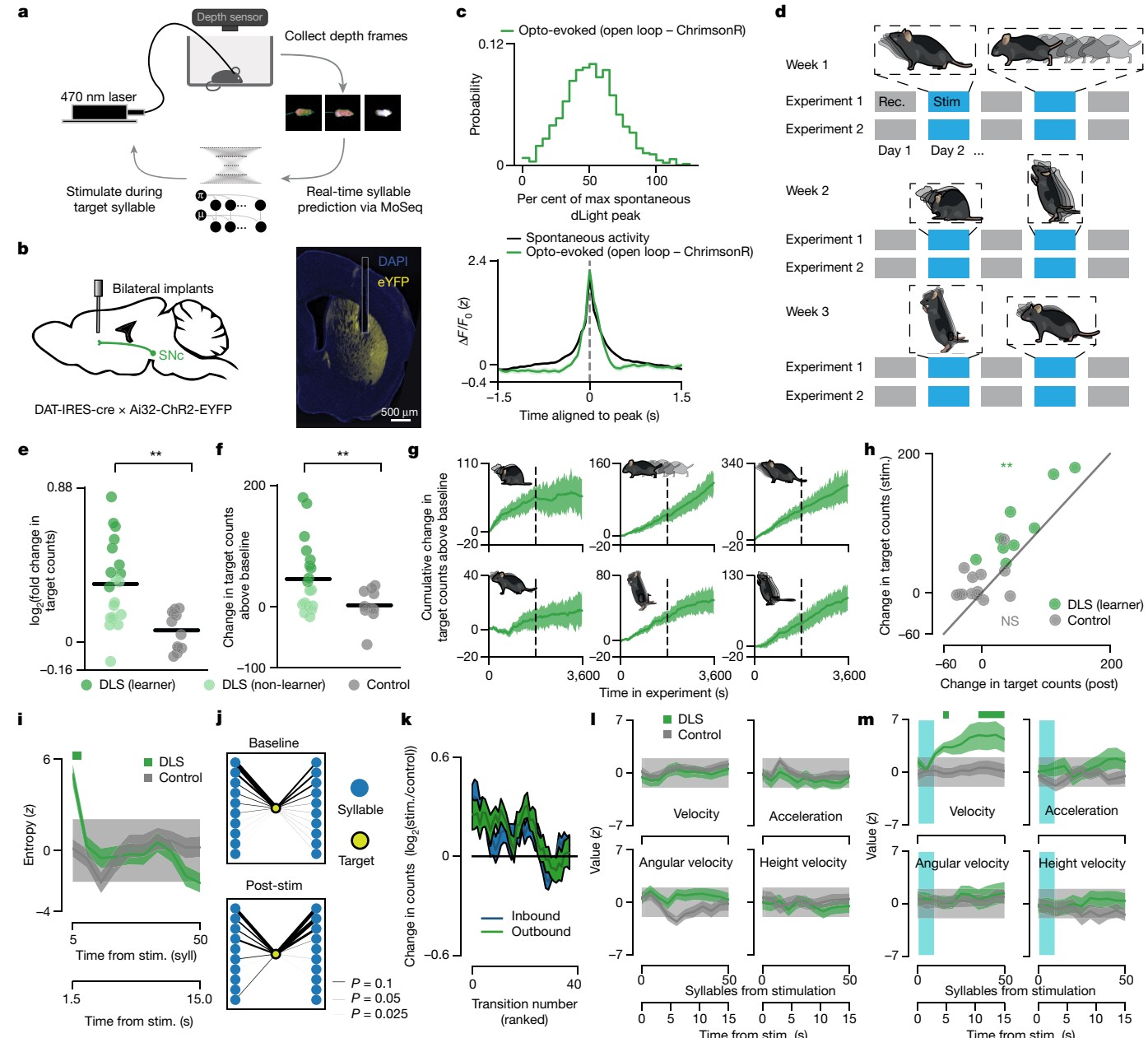

**Fig. 3 | Optogenetically evoked dopamine release in DLS reinforces syllable use and increases sequence variability. a**, The closed-loop MoSeq pipeline. **b**, Schematic and representative brain section of fibre cannulae over DLS dopamine axons. **c**, Top, normalized optogenetically evoked dLight peak magnitude distribution ($n = 842$ peaks). Bottom, mean waveforms from spontaneous and Opto-DA transients (Methods). Shading represents the 95% bootstrapped confidence interval. Max, maximum. **d**, Experimental schedule describing baseline (rec.) and stimulation (stim.) sessions for 'target' syllables (Methods). **e**, $\log_2$ fold change in target counts compared with baseline, per mouse, averaged across targets ($P = 0.002$, $U = 197$, $f = 0.82$, one-sided Mann–Whitney $U$ test) (see Methods for definition of 'learner'). **f**, Cumulative increase in target counts relative to baseline in Opto-DA and control mice ($P = 0.007$, $U = 184$, $f = 0.77$, one-sided Mann–Whitney $U$ test). **g**, Cumulative counts over concatenated stimulation sessions per target. Shading indicates bootstrap s.e.m. **h**, The relationship between target syllable usage changes from baseline during stimulation experiments versus post-stimulation experiments per mouse (Pearson $r = 0.89$, $P = 0.005$ for learners and $P = 0.082$ for controls,

one-sided shuffle test). NS, not significant. **i**, Average transition entropy following stimulation. Shading indicates bootstrap s.e.m. The light grey band indicates the 95% confidence interval of the pre-stimulation average. Data are binned using five-syllable-wide non-overlapping bins. The bar indicates significance ($P < 0.05$ for Opto-DA mice, $P > 0.05$ for controls, two-sided Mann–Whitney $U$ test comparing stimulation with catch trials). Syll, syllable. **j**, Sequence context changes from baseline to post-stimulation for an example mouse–target pair. Sequences proceed from left (incoming syllables) to right (outgoing syllables). Nodes are sorted by decreasing frequency at baseline. **k**, Average change in inbound and outbound transitions for target syllables on stimulation day sorted by the baseline rank of the transition. Traces are smoothed with a five-point rolling average. Shading indicates bootstrap s.e.m. **l**, Average kinematic parameters aligned to stimulation in Opto-DA mice and controls. Shading as in **i**. No comparisons between stimulation and catch trials in any of the mice were significant ($P > 0.05$, one-sided Mann–Whitney $U$ test). **m**, As in **l**, but following 3-s-long stimulation. The solid bar indicates significance ($P < 0.05$, one-sided Mann–Whitney $U$ test).

targeted syllables (Fig. 4c). These findings suggest that inter-mouse variability in Opto-DA learning reflects differences in the sensitivity of DLS to dopamine transients in general (as reflected by behaviour).

In addition, not all syllables were equally reinforceable in both the endogenous dopamine and Opto-DA experiments (Fig. 4d). There was no discernable relationship between the specific type of behaviour

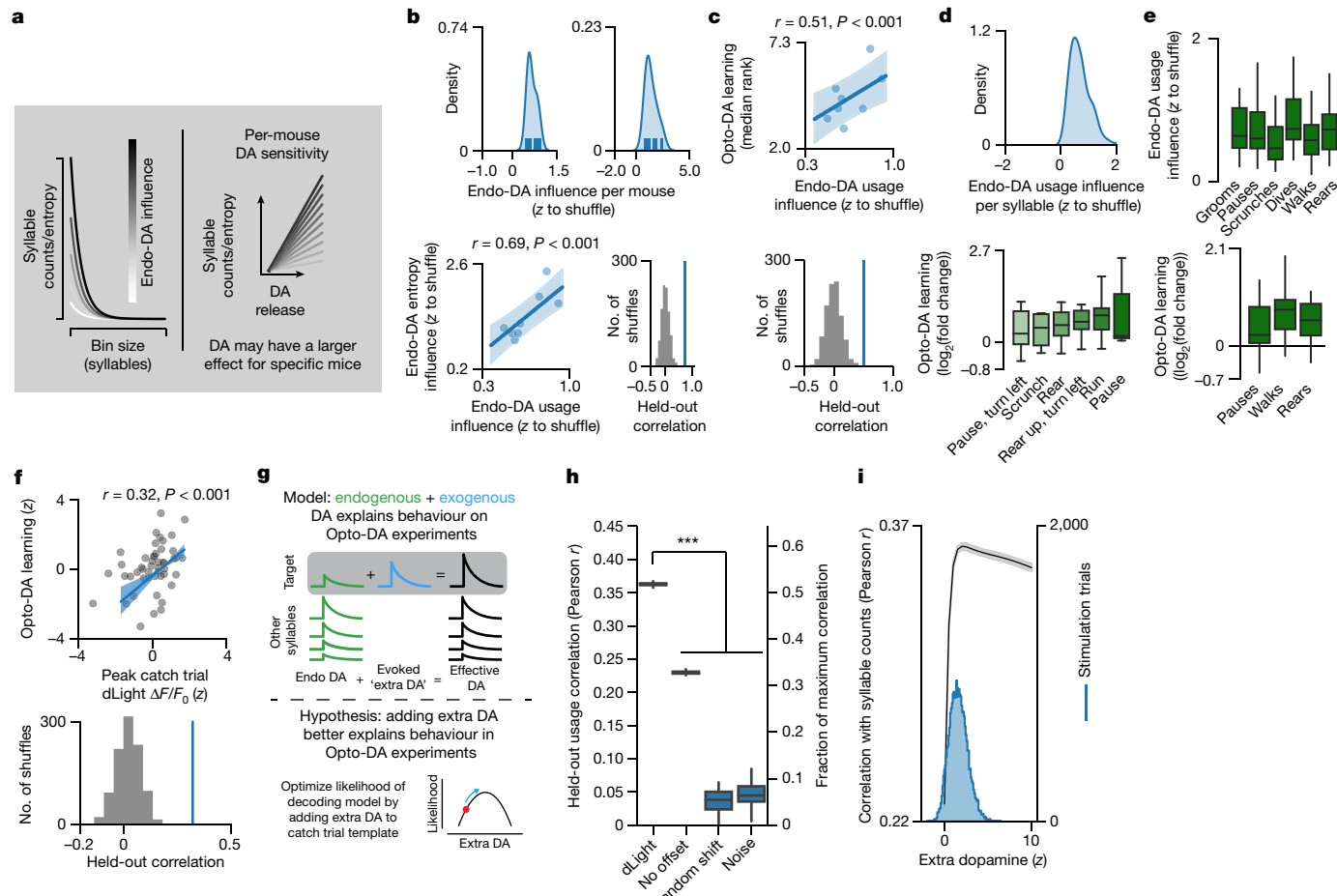

**Fig. 4 | Optogenetic syllable reinforcement varies predictably across mice and syllables. a**, Schematic depicting the relationship between observed endogenous dopamine-syllable usage correlations and per-mouse dopamine sensitivity. Dopamine (DA) sensitivity refers to the ability of endogenous, syllable-associated dopamine peaks to influence changes in syllable counts (Endo-DA count influence) or sequencing (Endo-DA entropy influence) within an experiment (see Methods for how indices were computed). **b**, Top, the distribution of per mouse Endo-DA count influence (left) and Endo-DA entropy influence (right) averaged across all syllables. Bottom, scatter plot (including linear regression model fit) of per mouse average Endo-DA count influence and Endo-DA entropy influence (Pearson $r = 0.69$ computed from model predictions on leave-two-out held-out data; $P = 0.001$, $P$-value computed via one-sided shuffle test). Shading indicates the 95% bootstrap confidence interval. **c**, Scatter plot (including regression line) of per mouse average Opto-DA learning versus Endo-DA count influence (Pearson $r = 0.51$ (computed from model predictions on leave-two-out held-out data); $P = 0.001$, $P$-value computed via one-sided shuffle test). Shading indicates the 95% bootstrap confidence interval. **d**, Top, the distribution of per syllable average Endo-DA count influence ($n = 296$ mouse–syllable pairs). Bottom, Opto-DA learning

plotted syllable-by-syllable for 'learner' mice ($n = 9$ mice). **e**, Top, average Endo-DA count influence across syllable categories. Bottom, average Opto-DA learning across syllable categories. **f**, Top, scatter plot (including regression line) of catch trial syllable-associated dLight and Opto-DA learning for each mouse–syllable pair ($r = 0.32$ over held-out data; $P < 0.001$, estimated via one-sided shuffle test). Bottom, model performance (evaluated with five times fivefold cross-validation) using actual versus shuffled data. **g**, Hypothesis that evoked dopamine release combines with ongoing endogenous release to alter behavioural choices. **h**, Model-based likelihood of predicting held-out syllable choices on Opto-DA stim experiments (blue) relative to control models (Methods) ($P = 7 \times 10^{-18}$ across all model comparisons relative to dLight model, $U = 2,500$, $f = 1$, two-sided Mann–Whitney $U$ test; $n = 50$ model restarts; Methods). The right $y$-axis indicates the model performance as a fraction the of maximum correlation. **i**, The relationship between average model accuracy (correlation between predicted syllable usage and actual usage) and the 'extra dopamine' free parameter (black). Shading indicates the 95% bootstrap confidence interval. The distribution of empirically measured optically evoked dLight fluorescence is shown in blue.

encoded by each syllable (for example, rearing, running or grooming) and the degree of reinforcement by endogenous dopamine or Opto-DA (Fig. 4e). By contrast, the ability of a particular syllable to be reinforced by exogenous dopamine was well predicted by that syllable's average endogenous dopamine transient amplitude: syllables that on average were associated with high amplitude dopamine transients during spontaneous behaviour were more easily reinforced by Opto-DA than those typically associated with low amplitude dopamine transients (Fig. 4f).

This observation suggests that the endogenous dopamine fluctuations that naturally occur during expression of a targeted syllable sum together with the exogenous dopamine induced by Opto-DA to promote syllable usage. To test this idea, we built a dynamic decoding

model that predicted moment-to-moment syllable expression on the basis of syllable-associated dopamine. In this model, syllable-associated dopamine was represented as the sum of two components: endogenous dopamine (that is, the observed syllable-associated dopamine observed at baseline) and a free parameter representing the extra dopamine afforded by Opto-DA. We then fit this model to data observed during catch trials in the Opto-DA experiments, in which optogenetic stimulation during the target syllable was not provided (Methods). This additive model could reliably predict moment-to-moment syllable choices during the Opto-DA experiment (which included stimulation trials), but made less effective predictions when dopamine transients were shuffled in time relative to syllables (Fig. 4g,h). In addition, the amount of

extra dopamine required for the model to accurately predict actual syllable usage patterns closely matched the amount of dopamine elicited by Opto-DA in our calibration experiments (Figs. 3c and 4i). Together, these data support a model in which endogenous and exogenous DLS dopamine sum together to influence syllable expression, and provide a possible explanation for the differential reinforcement of syllables observed in the Opto-DA experiment.

## An RL model reproduces spontaneous behaviour

The observation that both exogenously induced and endogenous fluctuations in DLS dopamine levels drive changes in syllable usage and sequencing raises the surprising possibility that, during spontaneous exploration, mice are optimizing their behaviour—as they do in more structured tasks—to maximize the amount of total dopamine obtained during an experiment. If so, this would suggest that mice interpret fast DLS dopamine fluctuations during spontaneous behaviour as a teaching signal capable of shaping action choices.

To test this hypothesis, we built a reinforcement learning (RL) model in which an agent is trained—based on observed syllable sequences and dopamine transients—to predict the syllable choices expressed by actual mice during spontaneous behaviour (Fig. 5a). Whereas RL agents typically seek to maximize overall reward by optimizing their action choices in a particular state, here the RL agent seeks to maximize dopamine by optimizing the choice of which syllable to express next given its current syllable (Methods). We train this RL agent using standard Q-learning rules, which govern how syllable-associated dopamine transients update the transition probabilities between syllables[44], and formulated this model such that dopamine is both rewarding and injects transient variability into subsequent syllable choices.

Inspection revealed that after training the model converged on a syllable transition matrix similar to that emitted by actual mice exploring an open field; comparing alternative formulations of this model revealed that maximal model performance depended on dopamine both reinforcing specific syllable transitions and briefly increasing the variability of syllable choices (Fig. 5b–d). These findings are consistent with a model in which mice structure spontaneous behaviour to maximize DLS dopamine. We note that although our models assume that dopamine acts as a reward, there are alternative formulations under which dopamine acts as a reward-prediction error that are also consistent with our data (Fig. 5e; Methods). This caveat is important given that mice rapidly modify syllable usage in response to Opto-DA stimulation, preventing us from formally distinguishing between these possibilities.

## Discussion

Goal-oriented behaviours are purposive and yield explicit rewards, whereas spontaneous behaviour can often appear to be aimless and inscrutable. And yet, even the behaviour of mice placed in a dark empty bucket exhibits remarkable structure. Here we show that this structure is governed by ongoing fast fluctuations in the neuromodulator dopamine. DLS dopamine transients both correlate with and causally influence how often each syllable is used and in what order, despite the absence of an explicit task or exogenous reward. The ability of a simple RL model, in which dopamine transients are substituted for the reward signal, to predict syllable choices suggests that dopamine acts as a moment-to-moment teaching signal, one that enables DLS to dynamically assemble behavioural sequences. Together, our observations identify a neural mechanism that actively shapes the trajectory of spontaneous behaviour as it unfolds, and propose an unexpected functional role for DLS dopamine during self-motivated action that is centred around choice and reinforcement rather than movement initiation or instantaneous kinematic control.

Recordings of dopamine neurons during spontaneous behaviour have revealed a variety of correlations with movement initiation,

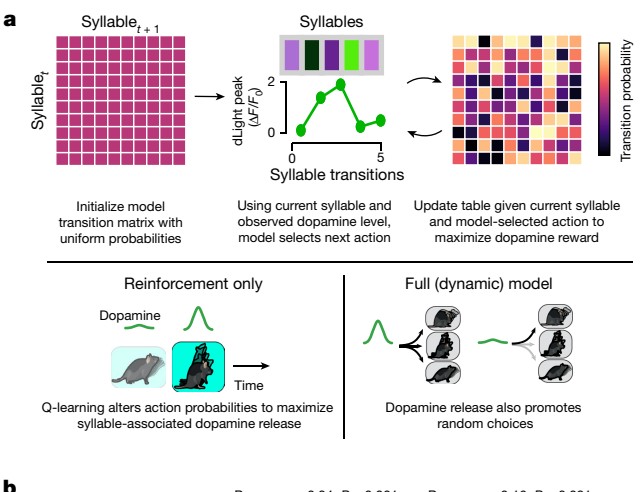

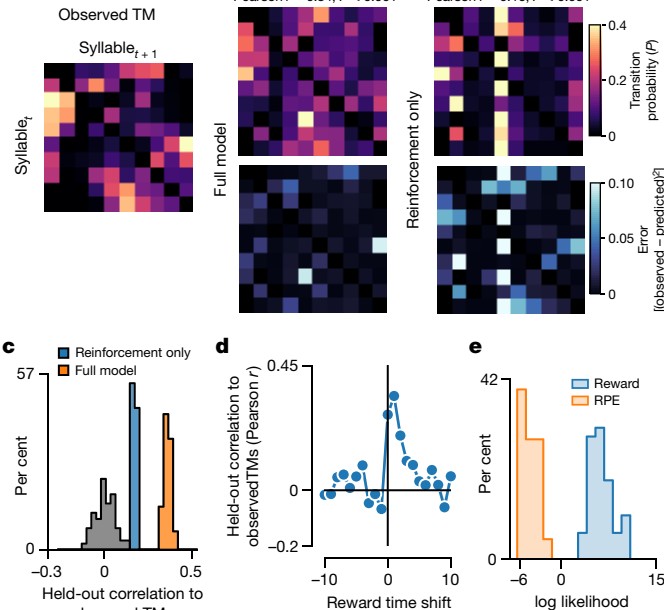

**Fig. 5 | RL models suggest that mice attempt to maximize dopamine during spontaneous behaviour. a**, Top, schematic describing modification of a standard RL model to explore relationships between DLS dopamine fluctuations and behavioural choices. Bottom, schematics of 'reinforcement-only' and 'full' model variants (Methods). **b**, Left, empirical transition matrix (TM) observed during open field behaviour. Centre, an example transition matrix learned by the full model (top), along with the squared difference between the empirically observed transition matrix and the example learned transition matrix (bottom). Right, as in centre, except for the reinforcement-only model. The average correlation between the observed transition matrix and the transition matrix learned by each model, along with the associated *P*-value computed via shuffle test, are given for each model type. For visualization, the model transition matrix is estimated by taking a softmax (see Methods) over the Q-table learned by the model. Here, the temperature parameter was set to 0.1 for visualization only. **c**, The distribution of correlations between the learned and observed transition matrices for both the reinforcement-only (blue) and full (orange) models, compared to a histogram of correlations between transition matrices learned with time-shuffled dLight traces (all models are statistically significant according to a shuffle test, defined as model fits exceeding 95% of shuffle correlation values, *n* = 100 shuffles). **d**, The performance of the full model after temporally shifting syllable-associated dLight amplitudes across syllables over various lags. **e**, The distribution of log likelihoods for models that consider dopamine as a reward versus a reward-prediction error (RPE) signal (Methods). The log likelihoods shown are for the best parameterization for each model type across 50 bootstraps of the dataset. On the basis of this relationship, we formulated models that treated dopamine transients as representing reward rather than reward-prediction error.

speed and kinematics, and strong optogenetic stimulation of dopamine neurons elicits movement, suggesting that dopamine may cause movements per se[9,11,12,14,20,33]. The failure of our calibrated optogenetic experiments to causally influence movement initiation or execution at the moment of stimulation argues that DLS dopamine organizes—rather than commands—movement by influencing the overall statistical structure of behaviour. Notably, DLS dopamine bidirectionally influences future syllable vigour, demonstrating that dopamine regulates both discrete (that is, which syllable to express next) and continuous (that is, how fast to express a particular syllable) aspects of spontaneous behaviour.

Our data suggest four broad, non-mutually exclusive models for how DLS dopamine transients may arise and thereby influence future behavioural choices. First, dopamine may represent an output of the motor system that enables it to modulate (and invigorate) the future expression of some syllables relative to others. In this model, DLS dopamine acts to implement a motor plan articulated by the cortex, thalamus and basal ganglia (all of which send projections to SNc[45]); our observations that syllable-associated dopamine causally influences both future syllable use and subsequent syllable choices is consistent with this model. Although the mechanisms that enable dopamine to briefly increase sequence variability are not yet clear, it is possible that this phenomenon relates to the ability of dopamine to increase SPN excitability, which may decrease the fidelity with which cortical inputs are transformed into ensemble SPN activity[30,46].

Second, DLS dopamine may represent the output of a circuit that evaluates the content of spontaneous behaviour. Although dopamine has classically been thought to report reward-prediction errors—which by definition require the provision of reward—it has recently been argued that dopamine may also encode action-prediction errors[47,48] (APEs). APEs are proposed to occur as animals either execute or plan to execute a behaviour that is unexpected in a given context; in the setting of spontaneous behaviour, an APE-like model would predict that DLS dopamine represents the comparison between the expressed (or soon-to-be-expressed) behavioural syllable and that which would have been expressed at a particular moment given an idealized transition matrix. Our finding (similar to that in ref. [9]) that syllable-associated dopamine transients reflect the probability of the next expressed behavioural syllable—but convey no information about syllable identity—is also consistent with the proposed role of APEs in conveying information about action errors that is independent of the specific identity of the expressed behaviour.

Third, DLS DA may encode an error signal unrelated to APEs. Given that most syllables are probably associated with some degree of active sensory sampling, it is possible that DLS dopamine reflects the difference between expected and experienced sensory cues in the environment that are encountered during each syllable. Similarly, DLS dopamine might evaluate the actual quality of execution of each syllable (akin to performance prediction errors observed in birdsong[49,50]), although our data suggest that this is not the case. Finally, the brain may be misinterpreting random fluctuations in DLS dopamine as a reward-like signal, thus inadvertently structuring behaviour; such stochastic fluctuations in cortical dopamine have recently been shown to support RL[51]. However, spontaneous behaviour in the open field evolves during our experiments with characteristic dynamics, arguing against this possibility. Future work will be required to arbitrate among these models.

We note that the midbrain dopamine system targets many brain areas with a diverse array of functions, and our experiments were deliberately designed to focus on the influence of dopamine on the DLS[14]. Given differences in their anatomical inputs and intrinsic timescales at which dopamine fluctuates[52], it is likely that the relationship that we observe between DLS dopamine and syllable usage does not apply to the DMS or the ventral striatum, which vary in their functional roles in movement, motivation and value assignment[24,53,54]. Indeed, it is possible that phasic dopamine fluctuations in these areas are responsible for initiating new movements or controlling their kinematics; alternatively, it is possible that tonic dopamine is broadly permissive but not causal for movements per se, as is suggested by the ability of L-DOPA therapy to revert movement deficits in humans with Parkinson's disease[55]. The causal relationship between dopamine and movement may also depend on the specific task in which an animal is engaged and the extent of its training; the observation that during goal-oriented tasks many individual dopamine neurons adopt task-related tuning to movements is consistent with this idea[11,16,19]. Note also that although both the endogenous dopamine and calibrated Opto-DA experiments argue that DLS dopamine is unrelated to movement initiation or kinematics (similar to observations in refs. [9–11,14,19]), it is possible that our bulk measurements and manipulations obscure a subtle role for specific SNc dopaminergic axons in controlling instantaneous movement.

Despite training, animals often fail to learn to perform structured tasks, with some animals, tasks or behaviours being more resistant to learning than others; understanding brain–behaviour relationships under these circumstances often requires considering only those animals whose behavioural responses exceeds some threshold for accuracy after training[56]. Although our experiments contain no overt goal or task structure, we also observe variability in Opto-DA learning, both across individual mice and across syllables. A substantial part of this variability can be explained by dopamine itself: mice that are more sensitive to endogenous dopamine fluctuations are better able to associate Opto-DA with targeted syllables, and syllables more effectively associated with Opto-DA are also associated with higher average endogenous dopamine transients. Our results also suggest that some mice are more sensitive to dopamine than others, despite being genetically identical and housed in similar conditions; further work is required to characterize the source of this inter-mouse variation.

Previous work has demonstrated that DLS SPNs encode information about the current syllable and the sequence context in which that syllable is expressed[27–29]. This information is probably inherited from thalamic and cortical inputs to DLS[57,58]. The ability of mice to upregulate syllables in response to DLS dopamine axon stimulation demonstrates that mice can recognize and reinforce the movements of their own body in the absence of external sensory cues (such as the click commonly used as a sensory trigger during clicker training); this self-recognition is remarkably specific, as dopamine stimulation does not substantially reinforce syllables that are kinematically related to the target, or syllables that are adjacent in time. Our findings naturally lead to a hypothesis in which dopamine encourages the use of associated syllables by inducing short-term plasticity in sensorimotor inputs to DLS[59]. Notably, our dopamine stimulation coincides with only a fraction of each targeted syllable, yet syllables remain temporally intact and are coherently reinforced as a whole; this observation supports the speculation that syllables are natural units of spontaneous behaviour used by the brain to structure action.

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

## Methods

A list of reagents and resources is provided in Extended Data Table 1.

### Ethical compliance
All experimental procedures were approved by the Harvard Medical School Institutional Animal Care and Use Committee (Protocol Number 04930) and were performed in compliance with the ethical regulations of Harvard University as well as the Guide for Animal Care and Use of Laboratory Animals.

### MoSeq
**Overview.** MoSeq (described previously in refs. [4,27,60]) is an unsupervised machine learning method that identifies brief, re-used behavioural motifs that mice perform spontaneously. MoSeq takes as its input 3D imaging data of mice and returns a set of behavioural 'syllables' that characterizes the expressed behaviour of those mice, and the statistics that govern the order in which those syllables were expressed in the experiment. MoSeq was used as it is originally described to explore relationships between endogenous DLS dopamine release and behaviour. This technology was further adapted to accommodate real-time syllable identification for closed-loop manipulations of neural activity as described below. Importantly, the underlying fitted autoregressive hidden Markov model (AR-HMM) for both the 'offline' and 'online' variants of MoSeq used in this study are the same, enabling comparisons of neural activity associated with syllables that were recognized and performed across multiple experiments.

**Pre-processing.** MoSeq consists of two essential workflows: one for pre-processing depth data and converting it into a low-dimensional time series that describes pose dynamics, and another for modelling the low-dimensional time-series data. As previously described, in order to focus on pose dynamics, raw depth frames were first background-subtracted to convert depth units from distance to height from the floor (in millimeters). Next, the location of the mouse was identified by finding the centroid of the contour with the largest area using the OpenCV findcontours function. An 80 × 80 pixel bounding box was drawn around the identified centroid, and the orientation was estimated using an ellipse fit (with a previously described correction for ±180-degree ambiguities[4,27]). The mouse was rotated in the bounding box to face the right side. The 80 × 80 pixel depth video of the centred, oriented mouse was then used to estimate pose dynamics.

**Size-normalizing deep network.** To accommodate noise in online syllable estimation and other sources of variation in the depth images not due to changes in pose dynamics (for example, occluding objects such as fibre-optic cables), we designed a denoising convolution autoencoder. The network was designed using TensorFlow to process images in <33 ms, the time between frame captures on the Microsoft Kinect V2[61]. On the encoder side, 4 layers of 2D convolutions (ReLu activation) followed by max pooling were used to downsample the 80 × 80 images to 5 × 5. Another 4 layers of 2D convolutions with successive upsampling layers were used on the decoding side to reconstruct the 80 × 80 images (10,310,041 total parameters). Batch normalization was used during training with a batch size of 128. In order to train the network, we used a size- and age-matched dataset (7–8 weeks of age). Mouse images were corrupted through rotation, position jitter, zooming in and out (that is, changing size), and superimposing depth images of fibre-optic cables. The network was fed corrupted mouse images as input and was trained to minimize the reconstruction loss of the original, corresponding uncorrupted mouse images (Extended Data Fig. 8a–c). The model was trained for 100 epochs using stochastic gradient descent with early stopping. Both online and offline variants of MoSeq included the size-normalizing network to ensure results were comparable.

**Dimensionality reduction and AR-HMM training.** In order to represent pose dynamics in a common space for all experiments, principal components and an AR-HMM time-series model were trained offline on a sample dataset of genotype- and age-matched mice. The parameters describing the principal components and AR-HMM model were saved. All depth videos acquired for this paper were then projected onto these same principal components for all experiments, whether they used the online or offline variant. As previously described, principal components were estimated from cropped, oriented depth videos, and the AR-HMM was trained on the top 10 principal components. Since the denoising autoencoder was used for all experiments, mouse videos from the size-and-age-matched dataset were fed through the denoising autoencoder prior to principal component estimation.

**Offline variant.** In the offline variant, the Viterbi algorithm was used to estimate the most probable discrete latent state sequence according to the trained AR-HMM for each experiment post hoc. This variant was used to analyse all data except for the Opto-DA experiments shown in Figs. 3 and 4.

**Online variant.** In the online variant, syllable likelihoods were computed and updated by computing the forward probabilities of the discrete latent states for each frame as they arrived from the depth sensor. To avoid spurious syllable detections, the targeted syllable probability had to cross a user-defined threshold for three consecutive frames.

### Histological verification
Mice were euthanized following completion of behavioural tests. Mice were first perfused with cold 1× PBS and subsequently with 4% paraformaldehyde. Fifty-micrometre sections of extracted brain tissue were sliced on a Leica VT1000 vibratome. All slices were mounted on glass slides using Vectashield with DAPI (Vector Laboratories) and imaged with an Olympus VS120 Virtual Slide Microscope.

### dLight validation and variant selection
dLight1.1 was selected to visualize dopamine release dynamics in the DLS owing to its rapid rise and decay times, comparatively lower dopamine affinity (so as to not saturate binding), as well as its responsiveness over much of the physiological range of known DA concentrations in freely moving rodents[31,62–64].

Since dopamine-free and dopamine-bound excitation spectra have yet to be reported for the dLight1.1 sensor, a series of in vitro experiments was performed to identify an excitation wavelength whose fluorescence was stable and independent of dopamine levels, and which therefore could be used for post hoc motion artefact correction. Like GCaMP, dLight1.1 uses cpGFP as a chromophore, and various generations of GCaMP have been shown to: (1) have an increase in ligand-free fluorescence when excited with 400 nm wavelengths and (2) have an isosbestic wavelength in the UV to blue region[65–67]. To test whether UV excitation could be a suitable reference wavelength for dLight1.1, HEK 293 cells (ATCC, cells were validated by ATCC via short tandem repeat analysis and were not tested for mycoplasma) were transfected with the dLight1.1 plasmid (Addgene 111067-AAV5) using Mirus TransIT-LT1 (MIR 2304). Cells were imaged using an Olympus BX51WI upright microscope and a LUMPlanFl/IR 60×/0.90W objective. Excitation light was delivered by an AURA light engine (Lumencor) at 400 and 480 nm with 50 ms exposure time. Emission light was split with an FF395/495/610-Di01 dichroic mirror and bandpass filtered with an FF01-425/527/685 filter (all filter optics from Semrock). Images were collected with a CCD camera (IMAGO-QE, Thermo Fisher Scientific), at a rate of one frame every two seconds, alternating the excitation wavelengths in each frame. Image acquisition and analysis were performed using custom-built software written in MATLAB[68] (Mathworks). Cells were segmented from maximum-projection fluorescence images using

Cellpose[69]. Cells with a diameter of less than 30 pixels were excluded from downstream analysis. Fluorescence traces were denoised using a hampel filter (window size 10 and threshold set to 2 median absolute deviations from the median) and normalized to $\Delta F/F_0$. Cells were included if their maximum $\Delta F/F_0$ exceeded 5%. $F_0$ was computed by fitting a bi-exponential function to the time series.

**Stereotaxic surgery for open field photometric recordings.** Eight- to ten-week-old C57BL/6J ($n = 6$ mice, The Jackson Laboratory stock no. 000664) mice of either sex were anaesthetized using 1–2% isoflurane in oxygen, at a flow rate of 1 l min$^{-1}$ for the duration of the procedure. AAV5.CAG.dLight1.1 (Addgene #111067, titre: $4.85 \times 10^{12}$) was injected at a 1:2 dilution (either sterile PBS or sterile Ringer's solution) into the DLS (AP 0.260; ML 2.550; DV −2.40), in a total volume of 400 nl per injection. For all stereotaxic implants, AP and ML were zeroed relative to bregma, DV was zeroed relative to the pial surface, and coordinates are in units of mm. Injections were performed by a Nanoject II or a Nanoject III (Drummond) at a rate of 10 nl per 10 s, unilaterally in each mouse. A single 200-μm diameter, 0.37–0.57 NA fibre cannula was implanted 200 μm above the injection site at the DLS (DV −2.20) for photometry data collection. Finally, medical-grade titanium headbars (South Shore Manufacturing) were secured to the skull with cyanoacrylate glue (Loctite 454).

Mice were group-housed prior to stereotaxic surgery procedures, and following surgery were individually housed on a 12-hour dark–light cycle (09:00–21:00). All behavioural recordings were done between 010:00 and 17:00.

**Stereotaxic surgery for simultaneous photometric recordings and optogenetic stimulation.** Six- to 12-week old DAT-IRES-cre mice ($n = 10$ mice, The Jackson Laboratory stock no. 006660) of either sex were injected with the same dLight1.1 virus described above into the right hemisphere DLS. Additionally, using the same previously described surgical procedure, 350 nl of AAV1.Syn.Flex.ChrimsonR. tdTomato (UNC Vector Core, titre: $4.1 \times 10^{12}$) was injected into the right hemisphere SNc (AP −3.160; ML 1.400; DV −4.200 from pia), in a 1:2 dilution for calibration and stimulation experiments (see below). Mice were implanted unilaterally with a 200 μm core 0.37–0.57 NA fibre over the DLS for simultaneous stimulation and photometric data collection.

Two of the ten mice were used to calibrate optogenetic stimulation (see 'dLight calibration experiments'). The other 8 mice injected with dLight and ChrimsonR were also run through the 3 complete closed-loop experiments described in 'Closed-loop DLS dopamine stimulation experiments' (one experiment with 250 ms continuous wave (CW) stimulation, one with 2 s CW stimulation, and another with 3 pulsed stimulation, 25 Hz frequency with 5 ms pulse width). Baseline data from these experiments were combined with mice described in 'Fibre Photometry for dLight recordings', thus yielding a total of $n = 14$ mice. Two of the 12 dLight only mice did not pass our quality control criteria for dLight recordings and were thus excluded from all dLight analysis (note that they were included in Extended Data Fig. 2a–b,d only, which strictly used behavioural data). Baseline data were considered data from the day prior to a stimulation day, or the day after with the targeted syllable excluded (yielding $n = 378$ experiments total). If the targeted syllable could not be reasonably excluded then data from the day after a stimulation day was excluded entirely.

## dLight behaviour procedures
**OFA experiments.** Depth videos of mouse behaviour were acquired at 30 Hz using a Kinect 2 for Windows (Microsoft) using a custom user interface written in Python (similar to ref. [60]) on a Linux computer. For all OFA experiments, except where noted, mice were placed in a circular open field (US Plastics 14317) in the dark for 30 min per experiment, for 2 experiments per day. As described previously, the open field was sanded and painted black with spray paint (Acryli-Quik Ultra Flat Black; 132496) to eliminate reflective artefacts in the depth video.

**Food reward experiments.** To assess whether spontaneous dLight transients in the DLS were of appreciable magnitude compared to reward consumption-related transients, a series of separate dLight photometry experiments were run to measure reward consumption-related transient magnitudes ($n = 6$ mice). For two days prior to the experiment, mice were habituated to the open field arena for two 30-min experiments on each day. On the morning of the experiment, to increase the salience of food reward, mice were habituated to the experimental room and food and water restricted for 3–5 h prior to beginning the experiment. Mice were placed in the arena, and behaviour and photometry data were simultaneously acquired. Chocolate chips (Nestle Toll House Milk Chocolate) were divided into quarters and introduced into the arena at random intervals and locations decided by the experimenter (with an average of 1 chocolate chip piece every 4 min) for mice to freely consume for a total of 30 min. To identify reward consumption-related responses, a human observer indicated each moment in time during the experiment where mice began to consume the chocolate via post hoc inspection of the infrared video captured by the Kinect. Photometry signal peaks for Fig. 2a were identified at the onset of consumption. Mean spontaneous transient peak had observed magnitudes of $2.12 \pm 0.80 \Delta F/F_0 (z)$ ($n = 5,247$ transients). By comparison, mean reward consumption-associated transients had an approximate magnitude of $2.36 \pm 0.92 \Delta F/F_0 (z)$ ($n = 10$ transients).

**Fibre photometry for dLight recordings.** Photometry and behavioural data were collected simultaneously. A digital lock-in amplifier was implemented using a TDT RX8 digital signal processor as previously described[27]. A 470 nm (blue) LED and a 405nM (UV) LED (Mightex) were sinusoidally modulated at 161 Hz and 381 Hz, respectively (these frequencies were chosen to avoid harmonic cross-talk). Modulated excitation light was passed through a three-colour fluorescence mini-cube (Doric Lenses FMC7_E1(400-410)_F1(420-450)_E2(460-490)_F2(500-540)_E3(550-575)_F3(600-680)_S), then through a pigtailed rotary joint (Doric Lenses B300-0089, FRJ_1x1_PT_200/220/LWMJ-0.37_1.0m_FCM_0.08m_FCM) and finally into a low-autofluorescence fibre-optic patch cord (Doric Lenses MFP_200/230/900-0.37_0.75m_FCM-MF1.25_LAF or MFP_200/230/900-0.57_0.75m_FCM-MF1.25_LAF) connected to the optical implant in the freely moving mouse. Emission light was collected through the same patch cord, then passed back through the mini-cube. Light on the F2 port was bandpass filtered for green emission (500–540 nm) and sent to a silicon photomultiplier with an integrated transimpedance amplifier (SensL MiniSM-30035-X08). Voltages from the SensL unit were collected through the TDT Active X interface using 24-bit analogue-to-digital convertors at >6 kHz, and voltage signals driving the UV and blue LEDs were also stored for offline analysis.

The output of the PMT was then demodulated into the components generated by the blue and UV LEDs. The voltage signal was multiplied by the two driving signals—corresponding to the green emission due separately to blue and UV LED excitation—and low-passed using a third order elliptic filter (max ripple: 0.1; stop attenuation: 40 dB; corner frequency: 8 Hz). The UV component was used a reference signal.

**Synchronizing depth video and photometry.** To align photometry and behavioural data, a custom IR led-based synchronization system was implemented. Two sets of 3 IR (850 nm) LEDs (Mouser part # 720-SFH4550) were attached to the walls of the recording bucket and directed towards the Kinect depth sensor. The signal used to power the LEDs was digitally copied to the TDT. An Arduino was used to generate a sequence of pulses for each LED set. One LED set transitioned between on and off states every 2 s while the other transitioned into an on state randomly every 2–5 s and remained in the on state for 1 s. The sequences of on and off states of each LED set were detected in the photometry data acquired

with the TDT and IR videos captured by the Kinect. The timestamps of the sequences were aligned across each recording modality and photometry recordings were down sampled to 30 Hz to match the depth video sampling rate. This same mechanism was used to align photometry data to keypoints in Extended Data Fig. 3.

**Photometry pre-processing.** Demodulated photometry traces were normalized by first computing $\Delta F/F_0$. $F_0$ was estimated by calculating the 10th percentile of the photometry amplitude using a 5-s sliding window to account for slow, correlated fluorescence changes between dLight and the UV reference channels. Both the dLight and reference channels were normalized using this procedure. Since the UV reference signal captures non-ligand-associated fluctuations in fluorescence (deriving from hemodynamics, pH changes, autofluorescence, motion artefact, mechanical shifts, and so on), a fit reference signal was subtracted from the dLight channel (see 'Photometry active referencing'). Finally, referenced dLight traces were $z$-scored using a 20-s sliding window with a single sample step size slid over the entire experiment to remove slow trends in $\Delta F/F_0$ amplitudes due to long timescale effects—for example, photobleaching. Only experiments where the maximum percentage $\Delta F/F_0$ exceeded 1.5 and the dLight to reference correlation was below 0.6 were included for further analysis.

**Photometry active referencing.** In order to remove the effects of motion and mechanical artefacts from downstream analysis, a fit reference signal was subtracted from the demodulated dLight photometry trace as initially mentioned in 'Photometry pre-processing'[31,54] (Extended Data Fig. 1g). First, the reference signal was low-pass filtered with a second-order Butterworth filter with a 3 Hz corner frequency. Next, to account for differences in gain or DC offset, RANSAC ordinary least squares regression was used to find the slope and bias with which to transform the reference signal to minimize the difference between the reference and the dLight photometry traces. Finally, the transformed reference trace was subtracted from the dLight trace.

**Capturing 3D keypoints.** To capture 3D keypoints, mice were recorded in a multi-camera open field arena with transparent floor and walls. Near-infrared video recordings at 30 Hz were obtained from six cameras (Microsoft Azure Kinect; cameras were placed above, below and at four cardinal directions). Separate deep neural networks with an HRNet architecture were trained to detect keypoints in each view (top, bottom and side) using ~1,000 hand-labelled frames[70]. Frame-labelling was crowdsourced through a commercial service (Scale AI), and included the tail tip, tail base, three points along the spine, the ankle and toe of each hind limb, the forepaws, ears, nose and implant. After detection of 2D keypoints from each camera, 3D keypoint coordinates were triangulated and then refined using GIMBAL—a model-based approach that leverages anatomical constraints and motion continuity[71]. GIMBAL requires learning an anatomical model and then applying the model to multi-camera behaviour recordings. For model fitting, we followed the approach described in ref. [71], using 50 pose states and excluding outlier poses using the EllipticEnvelope method from sklearn. For applying GIMBAL to behaviour recordings, we again followed[71], setting the parameters obs_outlier_variance, obs_inlier_variance, and pos_dt_variance to 1e6, 10 and 10, respectively for all keypoints.

**Computing 2D and 3D velocity.** To compute 2D translational velocity, the centroid of the keypoints associated with the spine (approximating whole-body movement) was computed for the $x$ and $y$ planes (the $z$ plane was disregarded). Then, the velocity was computed from the difference in position between every 2 frames and divided by 2 (to provide a smoother estimate of velocity). 3D translational velocity was computed the same way, except the $z$ plane was included in the calculation. The average velocity of the keypoints associated with the forepaws were used to compute 3D forelimb velocity.

**Partialing kinematic parameters from dLight.** To compute the relationship between dLight and forelimb velocity, other kinematic parameters known to be correlated with dLight were partialed out of the dLight fluorescence signal. Specifically, 2D velocity, 3D velocity and height were partialed out of dLight using linear regression. Then, the correlation between the partialed dLight signal and 3D forelimb velocity were computed and compared to 1,000 bootstrapped shuffles.

**Movement initiation analyses.** A changepoint detection algorithm was used to find moments where mice transitioned from periods of relative stillness to movement. To capture long bouts of movement, the velocity of the 2D centroid of the mouse was $z$-scored across each experiment and then smoothed with a 50-point (1.67s) boxcar window. To find sharp changes in velocity, the derivative of smoothed velocity trace was computed, and the result was raised to the third power. Peaks in this velocity changepoint score were discovered using SciPy's findpeaks function with the following parameters: height 1, width 1, prominence 1 so that consecutive data points around each peak were disregarded.

**dLight time warping.** To account for variability in syllable duration, dLight traces were time warped for Extended Data Fig. 4a. Here, all dLight traces were linearly interpolated using the numpy.interp function to a duration of 0.83 s, or 25 samples. Thus, syllables longer than 0.83 s were linearly compressed, and syllables shorter than 0.83 s were linearly expanded. We obtained similar results time warping dLight traces to 0.4 s; thus, the duration of time warped instances did not affect interpretation of subsequent analyses.

**dLight average waveform $z$-scoring.** For dLight waveforms shown in Fig. 1f, top and bottom, h,i,k and Extended Data Figs. 4c–g, 5c,f and 7c, first onset-aligned waveforms were $z$-scored using the mean and s.d. of fluorescence values from 10 s prior to 10 s after onset. Next, to account for differences in the number of syllable instances (trials) in each average, waveforms were additionally normalized by $z$-scoring relative to the mean and s.d. of 1,000 shuffle averages, where individual trials were circularly permuted prior to averaging.

**Decoding syllable identity from dLight waveforms.** To decode syllable identity from dLight waveforms or dLight peaks, a random forest classifier[72] (cuRF = 1,000 trees, max depth = 1,000, number of bins = 128, with cross-validation on 5 folds of data) was trained to predict syllable and syllable group identity on held-out data (similar to ref. [27]). Syllable groups were created by hierarchically clustering syllables based on their pairwise MoSeq distance (see below) and thresholds were increased with a distance cut-off in steps of 0.2. The inputs to the random forest classifier were either: (1) the maximum $z$-scored dLight value from syllable onset to 300 ms after syllable onset for each syllable instance or (2) dLight waveforms and their derivatives starting at syllable onset up to 300 ms into the future for individual syllable instances. Held-out accuracy was compared to 100 shuffles of syllable identity.

**Decoding turning orientation from dLight waveforms.** To decode turning orientation from dLight waveforms (Extended Data Fig. 5c), a linear support vector machine was trained to classify whether a particular syllable instance is a left- or rightward turning syllable using cross-validation on five folds of data. To sample the behaviour space of turning syllables, eight syllables with the largest angular velocities were chosen, four for each turning orientation. The model was fit to dLight waveforms and their derivatives starting at syllable onset up to 300 ms after onset for individual syllable instances and was tested on held-out data.

**MoSeq distance.** The MoSeq distance between two syllables was computed as previously described[27]. In brief, the estimated autoregressive

matrices for each syllable were used to generate synthetic trajectories through principal component space (that is, in the space defined by the first ten principal components of the depth video). Then, the correlation distance between trajectories for all pairs of syllables were computed. Since the online and offline variants of MoSeq used the same autoregressive matrices, these distances are equivalent in the online and offline variants.

**Analysing the relationship between dLight and syllable statistics within an experiment across syllables.** The dLight fluorescence associated with syllable transitions was computed as the maximum $z$-scored dLight value from syllable onset to 300 ms after syllable onset for each syllable transition, to account for jitter in dopamine release or technical jitter in defining syllable changepoints. Throughout the text, we refer to syllable-associated waveform peak amplitudes in $z$-scored $\Delta F/F_0$ units as 'syllable-associated dLight'. These dLight values were then averaged for each syllable and for each experiment. To assess the correlation between syllable-associated dLight and syllable counts, the dLight averages were $z$-scored across syllables in each experiment. These normalized dLight peaks represented whether a syllable had relatively higher or lower dLight during a given experiment. Finally, experiment-normalized dLight values along with syllable counts were then averaged across experiments for each mouse, thus leaving a value for each mouse and each syllable.

In order to measure the linear relationship between dLight peak values and syllable counts, a robust linear regression using the Huber regressor[73] predicted average syllable counts from average dLight peaks. The regression model was evaluated using a fivefold cross-validation repeated 100 times. Reported correlation values in Figs. 1j and 2 were estimated over the held-out data. $P$-values were estimated by comparing held-out correlation values to those estimated from a linear model computed on shuffled data. To remove syllables that varied due to finite size effects, only syllables that occurred at least 100 times total across all experiments per mouse were included.

To compute syllable entropy (estimating the randomness of outgoing transitions associated with each syllable), the outgoing transition probabilities associated with each syllable for each mouse were computed by counting the number of occurrences a syllable transitions to all others within an experiment and expressing this as a probability distribution. Next, the Shannon entropy was estimated over the outgoing transition probabilities for each syllable. Finally, the linear regression was estimated using the exact same procedure used for syllable counts.

**Analysing the relationship between dLight and syllable statistics across experiments for each syllable.** This series of analyses queried a total of 379 experiments. To capture the correlation between syllable-associated dLight peaks and syllable-associated behavioural features (syllable frequency, syllable entropy) within syllables but across experiments, first, the maximum $z$-scored dLight amplitude from onset to 300 ms after syllable onset at each syllable transition was computed. These syllable-associated dLight peaks were averaged for each experiment and syllable. Then, the dLight peak averages for each syllable and mouse were $z$-scored separately across experiments. Additionally, to put variation of each syllable across experiments on the same scale, syllable frequency, and syllable entropy were also $z$-scored for each syllable and mouse across experiments (Fig. 2b,i, bottom). Next, to remove variability in the calculation, values were pooled across syllables for each experiment, thus leaving a value for each experiment and mouse. To remove syllables that varied due to finite size effects, first only syllables that occurred at least 50 times per session on average were considered for downstream analysis. Linear models (Huber regressors) were fit to the resulting average dLight peaks, syllable frequency, and syllable entropy and evaluated as described in the previous section.

**Analysing the moment-to-moment relationship between dLight and syllable statistics within an experiment.** This series of analyses queried a total of 760 syllable–experiment pairs. dLight peak values were estimated by taking the maximum dLight value from onset to 300 ms after onset at each syllable transition. Velocity, syllable counts, and dLight peak values were averaged per syllable and per mouse over an expanding bin size; that is, velocity, syllable counts, and dLight peak values were estimated over the subsequent $n$ syllables after the transition were dLight value was calculated, where $n$ varied from 5 syllables up to 400 (Fig. 2e). For sequence randomness, to avoid finite size effects, dLight values were binned into 20 equally spaced bins per syllable (Fig. 2k). Then, transition matrices were combined within each bin across all syllables per mouse and per time bin. Finally, Pearson correlation values were then calculated between dLight values and the behavioural features estimated at each bin size. Pearson coefficients were $z$-scored using the mean and s.d. from Pearson coefficients estimated after shuffling dLight peak values.

Note that, in order to prevent the measurement from being influence by consistent non-stationarities in behaviour, these correlations were computed within each of the five time segments shown by dashed lines in Extended Data Fig. 2e. Then, per-segment correlations were averaged.

Time-constants associated with the correlation between dLight values and behavioural features over increasing bin sizes were estimated by fitting an exponential decay curve to the correlation values at each bin size using the SciPy's curvefit function[74]. Decay functions were fit over 1,000 bootstrap resamples of the data; the depicted distributions are taus fit over each resample.

**Analysing the cross-correlation between syllable-associated dLight and syllable usage.** The dLight fluorescence associated with all instances of a given syllable was binned across a three-minute window (chosen based upon the decay in Fig. 2f) and correlated with the use of that same syllable across a 3-min window, with the windows shifted the indicated amounts ($x$-axis). Correlation values (in Fig. 2g,h) were $z$-scored using the mean and s.d. from shuffles. $P$-values were estimated via shuffle test.

**Analysing the relationship between syllable-associated dLight and syllable classes.** Syllables were manually classified into 6 classes by hand-labelling crowd videos summarizing model output[4,27,60]. Then, syllable-associated dLight was averaged for all syllables within each class.

**Encoding model predicting average dLight from behaviour.** As with the linear regression analysis (previous section), dLight peaks were estimated by taking the maximum $z$-scored dLight amplitude from syllable onset to 300 ms after onset. Behavioural features (entropy, velocity, and syllable counts) after each transition were computed across various bin sizes as described in 'Analysing the relationship between dLight and syllable statistics within an experiment'. The bin sizes used were 5, 10, 25, 50, 100, 200 300, 400, 800 and 1,600 syllables. Syllable frequency, syllable entropy and velocity were averaged for each experiment and syllable in each bin size. These syllable and experiment-wide average values were then $z$-scored separately for each mouse and then averaged for each mouse and each syllable. In order to remove correlations between behavioural features they were whitened using zero-phase component analysis (ZCA) whitening. Whitened behavioural features were then fed to a Bayesian linear regression model to predict average dLight peak amplitudes per syllable and per mouse according to the following equation:

$$p(y|X, \beta, \sigma^2) = N(\beta^T X, \sigma^2)$$

where $X$ is defined as features, $\beta$ is regression coefficients, $y$ is dLight peak values, $\sigma$ is the s.d., and $N$ is the normal distribution. A normal prior was placed on the regression coefficients, and an exponential prior was placed on the s.d. Samples from the posterior were drawn via the no u-turn sampler (NUTS) using NumPyro ($n = 1,000$ warmup samples, then $n = 3,000$ samples)[75]. To assess the temporal relationship between behavioural features and dLight, a separate model was fit at each lag (here, features were whitened separately within each lag, Extended Data Fig. 6c). Overall model performance was quantified by feeding in features at their approximate best bin size to the model. For kinematic parameters and for entropy, this bin size (lag) was 10 timesteps; for syllable counts, this bin size (lag) was 100 timesteps (in syllable time). Then, each feature was fed in separately to quantify the performance of feature subsets.

**Encoding model predicting instantaneous dLight from behaviour.** In order to predict instantaneous dLight amplitudes from syllable counts, syllable entropy, velocity (2D, angular and height velocity), and acceleration, a series of convolution kernels were estimated, each of which map from each behavioural feature to dLight amplitude. Mathematically, the model can be written as follows:

$$\text{dLight}(t) = \sum_{f \in F} \sum_{t'=-2s}^{2s} \beta_f(t - t')f(t)$$

where $\text{dLight}(t)$ corresponds to the dLight trace at time step $t$, $f(t)$ is the behavioural feature at time step $t$, and $\beta$ is the weight of the convolution kernel. Kernel weights were optimized using a Huber loss via the Jax library[76]. That is to say, the dLight amplitude at each time sample is predicted by convolving each behavioural feature (frequency, entropy, velocity, and acceleration) with a convolution kernel and then summing the result across features. The model was trained and evaluated using twofold cross-validation by recording experiment, and the Pearson correlation between predicted dLight amplitudes and actual amplitudes was assessed on held-out experiments. In order to remove the effects of high frequency noise on training and evaluation, the dLight traces were smoothed using a 60-sample (2-s) boxcar filter prior to training and evaluation.

**Decoding model predicting behaviour from dLight.** The decoding model was designed to capture the two main effects of dopamine on behavioural statistics—usage and sequencing. The goal of the decoding model is to predict the likelihood of a sequence of syllables given past dopamine. The model comprises two key features: (1) a component that scales syllable usage by past syllable-associated dopamine, and (2) a component that scales randomness of the next syllable choice by past global dopamine. This can be summed up with the following equation:

$$P(s_t = i) \propto \exp\left(\frac{\alpha_a \sum_{n=1}^{250}\left(\text{d}a_{t-n}\exp\left(\frac{-n}{\tau_a}\right)\delta\left(s_{t-n} = i\right)\right)}{\alpha_b \sum_{n=1}^{250}\left(\text{d}a_{t-n}\exp\left(\frac{-n}{\tau_b}\right)\right)}\right)$$

where $s_t$ is the syllable a mouse performs at time $t$ during a behaviour experiment, $\text{d}a_t$ is the peak dLight recorded for syllable $s_t$, $\tau_a$ and $\tau_b$ describe the timescale of the usage and choice randomness component respectively, $\alpha_a$ and $\alpha_b$ scale the usage and choice randomness components respectively, and $\delta$ is the Dirac delta function (that is, one-hot encoding) that returns 1 when $s_{t-1} = i$ and 0 otherwise.

The parameters $\alpha_b$, $\tau_a$ and $\tau_b$ were fixed using approximations of analysis of the behavioural data (Fig. 2), and only $\alpha_a$ was learned by maximizing the likelihood of the function above given the sequence of syllables mice perform across a group of experiments and peak dLight measurements associated with the syllable sequence $z$-scored across each experiment. This was done via evaluating the likelihood of the function over multiple values of $\alpha_a$. $\tau_a$ (describing the effect of dopamine on future syllable usage/counts) was fixed at 100 syllable timesteps, and $\tau_b$ (describing the effect of dopamine on syllable sequence entropy) was fixed at 10 syllable timesteps. These values were approximated from the median $\tau$ values reported in Fig. 2.

To test model performance, data were split into 5 folds of training and test experiments and repeated 100 times using repeated $K$-fold cross-validation. We then computed the Pearson correlation between syllable counts from model simulations and actual syllable counts after smoothing with a 50-point rolling average. The one free parameter was fit using the training dataset and assessed on the test dataset. To avoid degradation in performance due to syllable sparsity, the top 10 syllables were used. The model was compared to a suite of control models, each evaluated over the same folds. The dopamine phase shift model was evaluated on the same data, but with all dopamine traces circularly shifted by a random integer between 1 and 1,000, and the noise model was evaluated with dopamine traces replaced by numbers drawn from a unit variance random normal distribution (since the traces were $z$-scored). In order to determine the maximum possible performance, the per experiment number of counts per syllable was correlated with the across-experiment average. Here, the model performed significantly better than controls. Median Pearson correlation between held-out predictions and observed data: actual model $r = 0.20$, phase shift control $r = 0.04$, noise model $r = 0.04$. Comparison between actual model and controls, $P = 7 \times 10^{-18}$, $U = 2,500$, $f = 1$, Mann–Whitney $U$ test, $n = 50$ model restarts.

To test the hypothesis that endogenous and exogenous dopamine linearly combine to alter the future usage of single syllables of behaviour, the present decoding model was modified. Maximal correlations were identified between predicted and observed syllable usages when adding (or subtracting) extra dopamine (termed 'extra DA') to the syllable-associated dopamine amplitudes observed on catch trials (Fig. 4g–i). Model-based log likelihoods of held-out syllable choices from Opto-DA stimulation day experiments were then computed. Other versions of this model (shown in Fig. 4h) included: (1) a control model in which no 'extra DA' is added to the model ('no offset'), (2) a control that uses a phase-shifted version of the dLight trace ('random shift'), and (3) a model that uses random numbers from a normal distribution with mean and variance matched to the dLight signal ('noise').

**dLight calibration experiments.** In order to characterize the speed and magnitude of evoked dopamine transients in the open field, dLight transients were elicited using brief optogenetic stimulation of SNc axons in the DLS expressing ChrimsonR while mice freely explored an open field arena[77]. A number of stimulation parameters were tested, using varying light intensity, stimulation length, and whether the stimulus was delivered in as a single continuous-wave pulse or delivered as multiple rapid short pulses. A single, short (250 ms; roughly the timescale of syllables), continuous stimulation pulse of red light at 10 mW (Opto Engine MRL-III-635; SKU: RD-635-00500-CWM-SD-03-LED-0) most effectively matched the amplitude and dynamics of endogenous dLight transients observed in the open field. The mean Opto-DA peak was measured at $2.18 \pm 0.85 \, \Delta F/F_0 (z)$, mean spontaneous peak $= 2.23 \pm 0.62 \, \Delta F/F_0 (z)$ and 99th percentile spontaneous peak $= 3.40 \, \Delta F/F_0 (z)$ Pulsed stimulation was also disfavoured as numerous studies have shown that pulsed stimulation can cause synchrony in neural and axonal networks that can evoke prolonged release[78–80]. Note that when excited with 635 nm light, the efficiency with which light evokes spiking in neurons expressing ChrimsonR is similar to efficiency with which blue light evokes spiking in neurons expressing ChR2[77].

Once a single 250 ms continuous pulse of 10 mW light was preliminarily chosen as the desired optogenetic stimulus to evoke dopamine release from DLS dopamine axons, another round of open-loop stimulation with these stimulation parameters was performed in the open field in two of the 10 total mice injected with dLight and ChrimsonR.

In these two mice, the intervals between stimulation times were drawn by randomly choosing an integer delay between 6 and 17 s for each stimulation. This range was chosen to guarantee each animal received at least 100 stimulations during an experiment. This enabled analysis of more stimulation trials with intended parameters to verify that the amplitude of evoked transients were within the same order of magnitude as spontaneously evoked transients (Fig. 3c).

**DMS dLight recordings.** As a series of control experiments to establish the specificity of DLS dopamine encodings, dLight recordings were performed in the DMS using the same techniques described above. dLight stereotactic injections in wild-type mice of either sex (C57BL/6J, $n = 8$) were performed at AP: 0.26, ML: 1.5, and DV: −2.2. Fibres for photometry (in C57BL/6J mice, $n = 8$, $n = 64$ recording experiments) were implanted in the manner described above at coordinates: AP: 0.26, ML: 1.5, DV: −2.0. Open field behavioural recordings and encoding models were performed for these data exactly as described above.

**Stereotaxic surgery for optogenetics.** Eight- to fifteen-week-old DAT-IRES-cre::Ai32 mice resulting from the cross of DAT-IRES-cre mice (The Jackson Laboratory, 006660) and Ai32 mice (The Jackson Lab, 012569) of either sex were used. The double transgenic DAT-IRES-cre::Ai32 mouse line has previously been used to conduct specific dopaminergic neuron activation[10,81,82]. Similar surgical procedures were used as described above, except two 200 μm 0.37 NA multimode optical fibres were implanted bilaterally over DLS (AP 0.260; ML 2.550; DV −2.300), in DAT-IRES-cre::Ai32 mice ($n = 20$). Control animals (DAT-IRES-cre mice, $n = 12$) of either sex were implanted bilaterally at the same coordinates, with 6 of these animals implanted in the nucleus accumbens (AP 1.300; ML 1.000; DV −4.000). These animals are collectively termed 'no-opsin controls' throughout the manuscript. Medical-grade titanium headbars were secured to the skull using cyanoacrylate. Optical stimulation experiments were then performed 2–3 weeks post-surgery.

**Closed-loop stimulation behavioural paradigm.** For two days prior to the closed-loop stimulation schedule (Fig. 3d), mice were habituated to the bucket for two 30-min experiments on each day. To test the change in statistics of specific syllables via syllable-triggered optogenetic stimulation, experiments were performed in a three-day schedule for each of six chosen target syllables. On the first day, two 30-min experiments were run for each mouse to characterize baseline target syllable usage. On the second day, two 30-min 'stimulation' experiments were performed for each mouse. During these experiments blue light (470 nm, 10 mW, a single 250-ms continuous-wave pulse) was delivered on 75% of target syllable detections. Stimulation was not conditioned on syllables occurring before the target. Finally, on the third day, baseline experiment recordings were repeated to assess syllable usage memory and usage decay after reinforcement. For half of the targeted syllables for each mouse (randomized across mice), the pre-stimulation baseline experiment is the same experiment as the post-stimulation baseline experiment for a different syllable (see Fig. 3d). A three-day cadence with multiple, short behavioural recording experiments per day was chosen to both minimize non-stationarities in syllable usage within an experiment, as well as to not expose the mice to the behavioural arena for more than one total hour per day. To control for order effects on changes in target syllable usages over time, animals were randomly split into two groups, each of which had a unique ordering of target syllables across the six stimulation days of the three-week cadence. The time interval between the first experiment and the second experiment for the same mouse on each day (either recording or stimulation) was 195 min on average ±58 min (s.d.). Mice were euthanized following completion of behavioural tests, and histology was performed using procedures described above.

To assess the effect of increased dopamine release these experiments were repeated with 3-s pulsed stimulation (25 Hz, 5 ms pulse width) in $n = 3$ DAT-cre::Ai32 and $n = 2$ (DAT-IRES-cre) control animals.

**Closed-loop velocity modulation experiments.** DAT-IRES-cre::Ai32 mice ($n = 5$) of either sex underwent 90-min recording and manipulation experiments. For the first 30 min, we estimated the distribution of velocities for a specific target syllable. Then, for the next 30 min, optogenetic stimulation was triggered both when the syllable was expressed according to our closed-loop system and when the animal's syllable-specific velocity exceeded the 75th percentile or went below the 25th percentile. Experiments were analysed only if the mouse received at least 50 stimulations and they increased the usage of the target syllable on average relative to their average baseline (established via separate recording experiments with no stimulation).

**Quantifying changes in target syllable counts.** First, the number of times the targeted syllable was performed during a 30-s sliding window (non-overlapping) for each 30-min stimulation experiment was computed. Then, a cumulative sum was taken. To turn the result into an estimate of excess target counts, a cumulative sum was also computed from the morning and evening experiments from the most recent previous baseline day. Finally, the average of the morning and evening baseline estimates was averaged and subtracted off.

'Learner' mice were defined as mice whose average change in target counts above baseline across all syllables exceed the maximum average change in target counts exhibited by no-opsin control animals. These $n = 9$ animals were used for subsequent analyses of target kinematics and learning specificity (Extended Data Fig. 10).

**Quantifying effects on syllables near the target in time.** To assess whether syllables temporally adjacent to the target were reinforced as a result of optogenetic stimulation, syllables were identified that—on average—were near to the target in time. Specifically, the average time between all non-targeted syllables and the target was computed, along with their change in counts above baseline. Then, syllables were binned when they occurred on average relative to the target in syllable units in equally spaced bins from ten syllables before the target to ten syllables after. Finally, for each experiment, a weighted average of the change in counts above baseline for all syllables in each bin was computed, where a syllable's weight was defined by its relative frequency in an experiment.

**Quantifying effects on syllables whose velocity was similar to the target.** To understand whether syllables with similar velocity profiles to the target were also reinforced, the average velocity from onset to offset for each syllable was computed and $z$-scored across instances within an experiment. Then, the average velocity of the target was subtracted from each syllable's average velocity. Finally, the change in count above baseline for each syllable was binned by its target-velocity-difference.

**Quantifying Opto-DA effects on movement parameters and sequence randomness.** To quantify the effects of Opto-DA on movement parameters and sequence randomness over short timescales, sequence entropy, velocity (2D, angular and height velocity) and acceleration were estimated in five-syllable-long non-overlapping bins starting from stimulation onset. This window was chosen to minimize noise in downstream calculations while retaining reasonable time-resolution. To compensate for non-stationarities in behaviour across the experiments, mice, and targeted syllables, entropy, velocity and acceleration pre-stimulation-onset were subtracted from their values post-stimulation. Finally, these baseline-subtracted values were $z$-scored using the mean and s.d. estimated from catch trials.

**Analysing the influence of dopamine on optogenetic reinforcement**
**Mice used to assess the influence of endogenous dopamine fluctuations on optogenetic reinforcement.** As described above, eight mice

injected with dLight and ChrimsonR were also run through closed-loop reinforcement experiments. The reinforcement experiment run with 250 ms 10 mW CW stimulation enabled decoding analysis of how exogenous dopamine release altered usage of syllables during experiments in which 'extra DA' was added (Fig. 4g–i).

**Predicting the amount of exogenously added dopamine during Opto-DA experiments using the decoding model.** To predict the magnitude of exogenously evoked dLight fluorescence using the decoding model, dLight fluorescence on each instance in which the mouse expressed the target syllable and received stimulation was replaced with the average dLight fluorescence observed for the target syllable on catch trials during which there was no optogenetic stimulation. Then an offset (denoted as 'extra DA') was added to each syllable instance in which the mouse received stimulation. The likelihood of the syllable sequences expressed during Opto-DA experiments was computed for a range of extra DA offsets (and hence a range of exogenously added dopamine). The model was evaluated using the exact same procedure described in 'Decoding model predicting behaviour from dLight', except the repeated K-fold splits (5-fold split repeated 100 times) was performed over stimulation experiments. The 'extra DA' outputs of the model were compared to empirical photometric data collected from animals expressing dLight that underwent ChrimsonR-mediated closed-loop reinforcement (Fig. 4i).

**Using the influence of endogenous dopamine to predict Opto-DA reinforcement.** In order to assess whether the impact of dopamine at baseline could predict Opto-DA reinforcement, we used the correlation between dopamine fluctuations and syllable statistics (usage and entropy) within an experiment. Specifically, we computed the correlation between dLight levels and usage as outlined in 'Analysing the moment-to-moment relationship between dLight and syllable statistics within an experiment' (Fig. 2e,k), except correlations were assessed per mouse and per syllable. Values at each bin size were $z$-scored using the mean and s.d. correlations computed over shuffled data. Here, n = 100 shuffles were used for the correlation with entropy for computational efficiency. To determine the modulation depth of these correlation curves for each mouse and syllable, we used the s.d. of the correlation values across bin sizes. This resulted in a value that reflected the short-term influence of dopamine on usage (Endo-DA count) and entropy (Endo-DA entropy) for all syllable–mouse pairs. Finally, these estimates were averaged per mouse for Fig. 4b,c, and per syllable for Fig. 4d. Then, the $\log_2$ fold change in target counts on stimulation days relative to baseline days was used as an estimate of Opto-DA learning. To mitigate mouse-to-mouse variability, the $\log_2$ fold change in target counts was normalized by computing the $\log_2$ fold change in target counts against all pairs of non-stimulation days per mouse. The mean and s.d. of this distribution was used to $z$-score Opto-DA learning per mouse.

Bayesian linear regression models were used in Fig. 4b,c. A normal prior was placed on the regression coefficients, and an exponential prior on the variance. Samples from the posterior were drawn via the no u-turn sampler (NUTS) using NumPyro ($n$ = 1,000 warmup samples, $n$ = 2,000 samples)[75]. Performance was assessed using leave-two-out cross-validation. The linear regression model presented in Fig. 4f utilized a Huber regressor[73]. Performance of the Huber regressors was assessed using fivefold cross-validation repeated five times.

**Applying RL models to open field behaviour**
**Reinforcement-only RL model.** RL models have four key components: a reward signal, a state, a state-dependent set of available actions, and a policy (which governs how actions are chosen). Here, a simple Q-learning agent with a softmax policy was designed to model mouse behaviour in the open field as an RL process over endogenous dopamine levels[44]. Our model was recast (specifically a Q-learning agent with a softmax policy) to use endogenous dopamine (that is,

syllable-associated dLight) as a reward signal, behavioural syllables as states, and transitions between behavioural syllables as actions. Given a syllable at time $t+1$, the dLight peak occurring during the syllable at time $t$ is considered the 'reward'. The Q-table for the model was initialized with a uniform matrix with the diagonal set to 0, since by definition there are no self-transitions in our data. For every step of each simulation, given the currently expressed syllable (that is, the state), the model samples possible future syllables (actions) based on the behavioural policy and the expected dLight transient magnitude (expected reward, specified by the Q-table) associated with each syllable transition. Then, the model selected actions according to the softmax equation

$$p(a|s) = \frac{e^{Q_s(a)/\tau}}{\sum\limits_{b=1}^{n} e^{Q_s(b)/\tau}}$$

where $\tau$ is the temperature. The model is fed 30-min experiments of actual data. Data was formatted as a sequence of states and syllable-associated dopamine. Given the current state, the model selects an action according to the softmax equation. To update the Q-table and simulate the effect of endogenous dopamine as reward, the syllable-associated dopamine is presented to the model as reward in a standard Q-learning equation. Specifically, the Q-table was then updated according to

$$Q(s_t, a_t) \leftarrow Q(s_t, a_t) + \alpha[r_{t+1} + \gamma \max_a Q(s_{t+1}, a) - Q(s_t, a_t)]$$

where $Q$ is the Q-table that defines the probability of action $a$ while in state $s$, $\alpha$ is the learning rate, $r$ is the reward associated with action $a$ and state $s$ (the dLight peak value at the transition between syllable $a$ and syllable $s$), and $\gamma$ is the discount factor. Performance was assessed by taking the Pearson correlation between the model's resulting Q-table at the end of the simulation and the empirical transition matrix observed in the experimental data. Here, each row of the empirical transition matrix and the Q-table were separately $z$-scored prior to computing the Pearson correlation. Note that the learned Q-table is functionally equivalent to a transition matrix in this formulation. To avoid degradation in performance due to syllable sparsity, the top 10 syllables were used.

**Dynamic RL model.** To account for the short-term effect of dopamine on sequence randomness, a dopamine-dependent term was added to the baseline model's policy

$$p(a|s) = \frac{e^{Q_s(a)/\tau(t)}}{\sum\limits_{b=1}^{n} e^{Q_s(b)/\tau(t)}}$$

where temperature is now time-dependent and evolves according to,

$$\tau(t) = I(t)\exp\left(\frac{t-n}{\tau_{\text{decay}}}\right) + \tau_{\text{baseline}}$$

and,

$$I(t) = \upsilon \quad \text{if } r(t) \geq \lambda$$

Here, $\tau_{\text{decay}}$ corresponds to the time constant with which dopamine's effect on temperature decays, $\tau_{\text{baseline}}$ is the baseline temperature, $\upsilon$ is the amount by which temperature is increased if the $r(t)$ goes above the threshold $\lambda$, and $n$ is the number of timesteps after the threshold has been crossed. Experiments were split into training and test datasets via twofold cross-validation, and the training set was used to fit all free parameters. To compare the dynamic to the reinforcement-only model, $\upsilon$ was set to 0—this turns off the temperature varying component of the dynamic model. Note that we observe qualitatively similar results

under an alternative formulation. Rather than feeding the model 30-min sessions of actual data, we allow the model to freely select actions, and reward was randomly drawn from dLight peaks associated with that action in actual data.

**Reward-prediction error model variant.** Models were fit using observed dopamine magnitude as either the (1) reward term (see above) or (2) reward-prediction error term $[r_{t+1} + \gamma\max_a Q(s_{t+1}, a) - Q(s_t, a_t)]$. For each model type, a grid search was performed across values of $\alpha$ (learning rate), $\gamma$ (discount factor, used in the reward model only), and temperature (randomness of the next action). Held-out log likelihood was computed for each fit and $z$-scored using the mean and variance of the held-out log likelihood from models fit to data shuffled between experiments ($n = 10$ shuffles). This comparison is only valid for our particular model formulation. There are alternative formulations for which dopamine acting as a reward-prediction error are consistent with our data.

## Statistics

All hypothesis tests were non-parametric. Effect sizes for Mann–Whitney $U$ tests are presented as the common language effect size $f$. Correlations were established as significant by comparing to $n = 1,000$ shuffled correlations (referred to as the shuffle test throughout the manuscript). For shuffle test if all correlations exceeded the 1,000 shuffles, the $P$-value is listed as $P < 0.001$ rather than $P = 0$. P-values were adjusted to account for multiple comparisons where appropriate using the Holm–Bonferonni stepdown procedure. Sample sizes were not pre-determined but are consistent with sample sizes typically used in the field. For examples using similar techniques see[10,14]. Blinding was not performed, but MoSeq-based analysis of behaviour was automated.

## Plotting

Box plots (here and throughout) obey standard conventions: edges represent the first and third quartiles, whereas whiskers extend to include the furthest data point within 1.5 interquartile ranges of either the first or third quartile.

## Software packages

In addition to analysis-specific packages cited in the relevant sections above, the following packages were used for analysis: NumPy[83], Python[84], Seaborn[85], Matplotlib[86] and Python 3 (ref. [87]).

## Reporting summary

Further information on research design is available in the Nature Portfolio Reporting Summary linked to this article.

# Data availability

The data that support the findings of the current study are available on Zenodo at https://doi.org/10.5281/zenodo.7274803.

# Code availability

All code related to this study was developed in MATLAB or Python and is available at http://github.com/dattalab/dopamine-reinforces-spontaneous-behavior.

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

**Acknowledgements** S.R.D. is supported by NIH grants RF1AG073625 and R01NS114020, and the Brain Research Foundation. S.R.D., B.L.S. and S.W.L are supported by NIH grant U19NS113201 and and the Simons Collaboration on the Global Brain. J.E.M. is supported by a Career Award at the Scientific Interface (CASI) from the Burroughs Wellcome Fund. W.F.G. is supported by NIH grant F31NS113385. M.J. is supported by NIH grant F31NS122155. C.W. is a Fellow of the Jane Coffin Childs Memorial Fund for Medical Research. S.W.L is supported by the Alfred P. Sloan Foundation. We thank N. Bhagat, J. Araki and J. Wallace for administrative support; the Harvard Neurobiology Imaging Core for imaging support, which is supported as part of NINDS grant NS072030; the HMS Research Instrumentation Core, which is supported by the Bertarelli Program in Translational Neuroscience and Neuroengineering, and by NEI grant EY012196; S. Gershman, J. Assad, S. Shea and members of the Datta laboratory for useful comments on the paper; and S Knemeyer for mouse drawings. Portions of this research were conducted on the O2 High Performance Compute Cluster at Harvard Medical School. Brain diagrams were created with BioRender.com. We apologize to those whose work we were unable to cite due to length limits.

**Author contributions** J.E.M., W.F.G., M.J., B.L.S. and S.R.D. designed experiments and interpreted the data. J.E.M., W.F.G. and M.J. performed experiments and data analyses. J.W., R.W.H., R.C. and R.S. conducted behavioural experiments and histology. D.B. assisted with animal husbandry. C.W., M.A.M.O., S.R.P., N.U. and S.W.L. assisted with computational modelling. T.K. and D.K. performed HEK cell experiments. J.E.M., W.F.G., M.J. and S.R.D. wrote the manuscript with input from all authors.

**Competing interests** S.R.D. sits on the scientific advisory boards of Neumora and Gilgamesh Therapeutics, which have licensed or sub-licensed the MoSeq technology.

**Additional information**
**Correspondence and requests for materials** should be addressed to Sandeep Robert Datta.

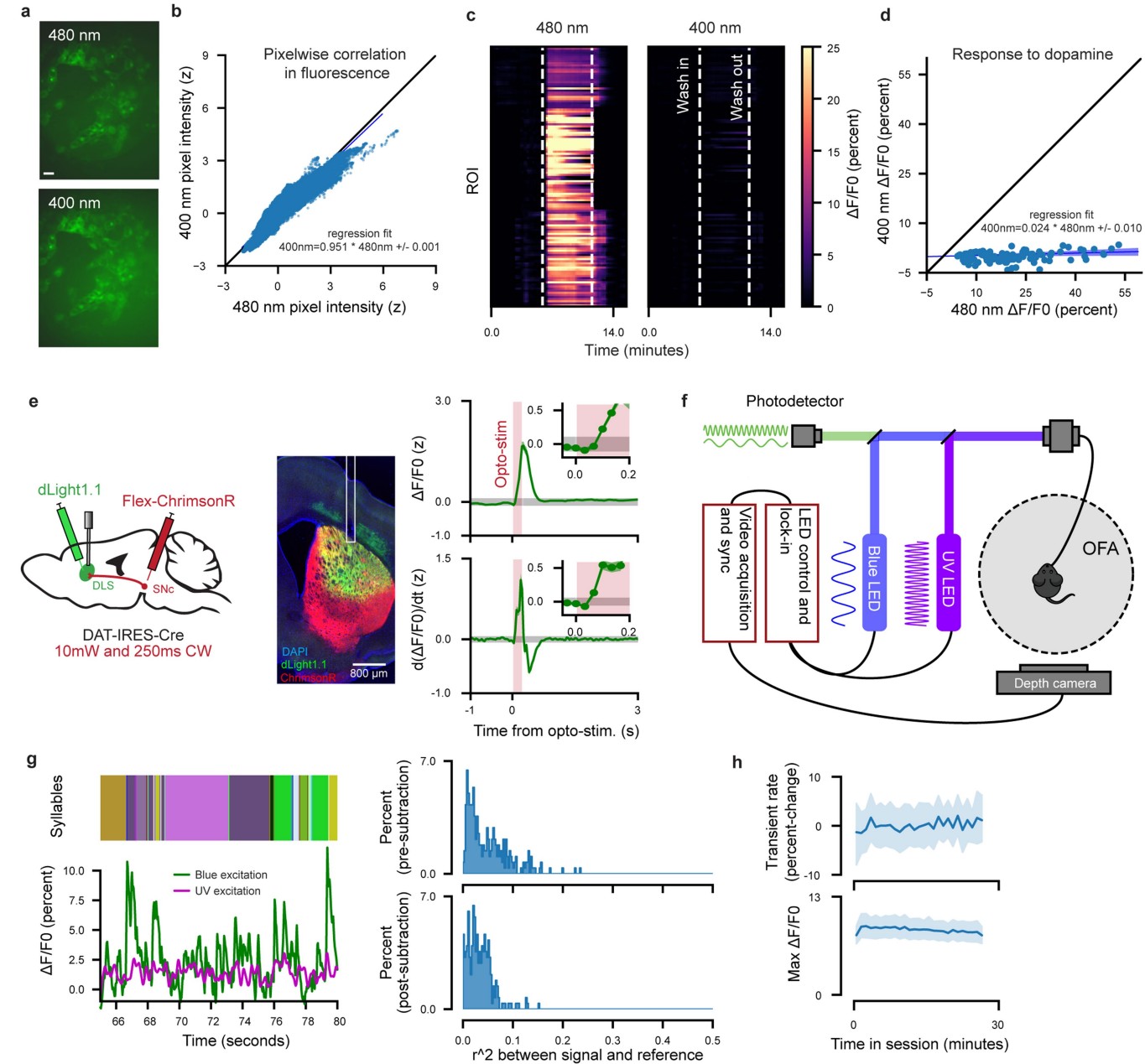

**Extended Data Fig. 1** | See next page for caption.

**Extended Data Fig. 1 | dLight validation, photometry setup and motion artifact removal. a**) Maximum projection epifluorescence images of HEK cells transfected with the dLight plasmid, and excited with either 480 nm blue (top) or 400 nm UV (bottom) light. Green emission (527 nm) was collected for both excitation wavelengths. Scale bar indicates 20 µm. Three separate experiments were performed with n = 28, n = 45, n = 42 regions of interest (ROIs) from the first, second, and third experiment respectively. **b**) Scatter plot of pixel fluorescence for each pixel location under both 400 nm and 480 nm excitation. A regression line was fit to the scatter plot (blue). The strength of the correlation demonstrates that both excitation wavelengths cause similar spatial patterns of dLight emission at 527 nm. **c**) Single cell dLight responses to perfused dopamine at 480 nm (left) or 400 nm (right) wavelengths. White dashed lines indicate time points where dopamine is washed into and out of the sample. Each row is an individual cell region of interest (ROI). **d**) Correlation in single cell dLight responses to perfused dopamine imaged with blue (x-axis) or UV light (y-axis). Each dot is an individual cell. Blue line indicates linear fit. The near-zero slope indicates that the UV-light-dependent green emission is almost entirely independent of dopamine concentration. **e**) Validation of dLight1.1 response to optogenetic stimulation. To assess how quickly DLS dLight reports dopamine transients in vivo, SNc axons were optogenetically stimulated using ChrimsonR (whose excitation spectrum is separated from that of dLight) while dLight fluorescence was recorded. Left: schematic illustrating viral injection and implant procedure to simultaneously record dLight transients and optogenetically stimulate dopamine axons in DAT-Cre animals. dLight was injected into the right dorsolateral striatum, Cre-dependent ChrimsonR was injected into the right SNc, and an optical fiber was placed above the dLight injection site in the DLS (see Methods). Middle: coronal slice depicting the expression of both dLight and ChrimsonR in the striatum. Right: mean stimulation-evoked dLight fluorescence using ChrimsonR in dopaminergic axons originating from SNc. The green shaded region indicates bootstrap SEM. The gray shaded area indicates the 95% bootstrap confidence interval of the mean trace pre-stimulation. Red shading indicates the duration of optogenetic stimulation. dLight transients are resolvable starting 67 ms from stimulation onset, suggesting that dLight can report rapid dopamine dynamics via photometry in DLS (p = 0.005, U = 2610, f = 0.32, one-sided Mann-Whitney U test). **f**) Schematic describing the photometry recording setup. A blue and UV LED were modulated at different frequencies, and light delivered to the mouse via a single fiber optic cable. Green fluorescence was acquired using a photodetector and the blue (dLight) and UV (isosbestic) components were separated using lock-in amplification. **g**) Example data depicting a dLight trace along with the simultaneously recorded reference signal acquired during free behaviour. The UV, or reference signal, represents the contributions of motion and mechanical artifacts and other non-ligand dependent changes in sensor fluorescence. Since UV and blue excitation of dLight cannot be perfectly matched, a gain and bias term were fit to match the UV-excited emission to the blue-excited emission per previously published papers[54]; this fitting process maximizes the correlation between the UV and blue signals, enabling us to effectively subtract the reference signal from the dLight signal. Left: example fit showing the UV trace scaled to match the dLight trace (bottom) aligned to syllables (top). Right: histogram of $r^2$ values between the photometry reference (UV component) and signal (dLight component), prior to fitting and subtracting the reference signal (top) and after fitting and subtracting the reference signal (bottom). Note that the baseline correlation between the reference and signal channels prior to subtraction is low. **h**) Top: the probability, on average, of observing a dopamine transient across all mice and experiments – defined as the ∆F/F0 trace crossing 1 standard deviation (computed per experiment) – at a given timepoint in the experiment. The probability was estimated in one minute bins. Shading indicates the 95% bootstrap confidence interval across per-mouse averages. Bottom: the maximum ∆F/F0 per 1 min time bin across all mice and experiments. As in the top plot, the average across all mice and experiments is shown, with shading indicating the 95% bootstrap confidence interval across per-mouse averages.

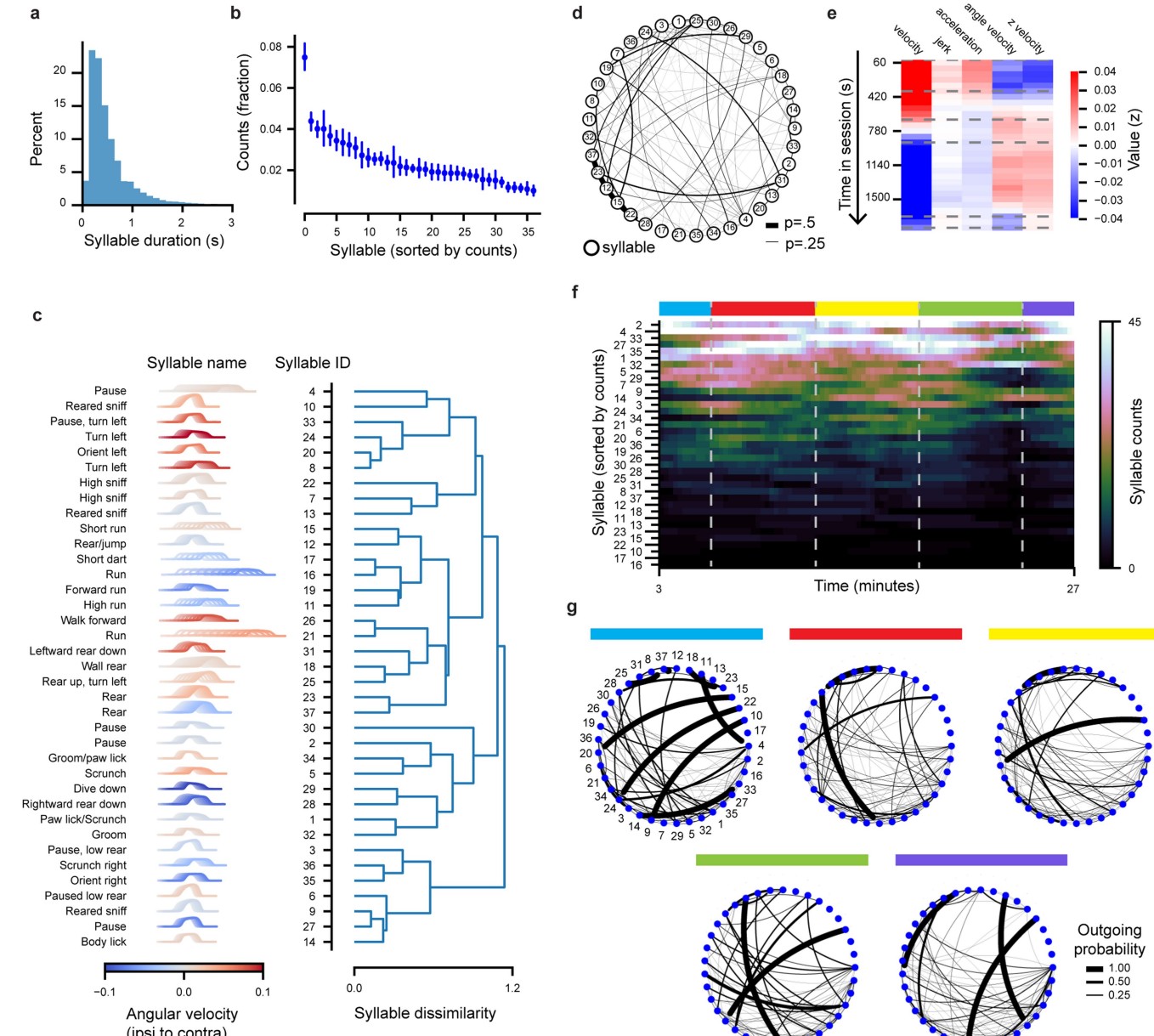

**Extended Data Fig. 2 | MoSeq captures subsecond structure in spontaneous mouse behaviour. a**) Distribution of syllable durations identified by MoSeq. The mean/median syllable duration was 566/400 ms +/− 636 ms SD. **b**) The average number of times each MoSeq-identified syllable is used during a 30-minute experiment per mouse (n = 16 mice). Error bars indicate bootstrap 95% confidence intervals across mice. **c**) Human-annotated descriptions of observed behavioural syllables. Left: semantic labels and "spinograms" of all behavioural syllables used more than 1% of the time, here to provide an illustration of movements associated with syllables. Each trace in the spinogram is an average height profile of the mouse computed by taking the pixel values along the center of the depth image across columns (note that MoSeq pre-processes depth images so that mice always face to the right of the cropped depth image). Each trace from left to right is the average of each frame of the behavioural syllable from the beginning to end. The distances between successive traces are proportional to the average x/y displacement from one frame to the next. Spinograms are color-coded by the average angular velocity of the syllable. Right: dendrogram computed using the pairwise MoSeq distance of all behavioural syllables (see Methods for a description of how MoSeq distances, which capture the average three-dimensional pose dynamics of each syllable, are computed). Spinograms are aligned to their corresponding leaf in the dendrogram. **d**) The average transition matrix visualized as a state map. Each

circle corresponds to a syllable, and each arrow corresponds to the likelihood that there is a transition from one syllable to the next. Arrow width indicates transition probability. All transitions with a probability below 0.1 are removed for visual clarity in this statemap and in all subsequent statemaps. **e**) Kinematic parameters over time averaged across all experiments and mice, demonstrating non-stationarities in kinematics across each recording experiment. Lines indicate boundaries derived via k-means clustering of this data. Note that these boundaries were used for analysis shown in Fig. 2e and k. Specifically, in order to prevent non-stationarities from impacting within-experiment correlations, correlations were computed within each of these segments and then averaged. **f**) Heatmap of syllable counts computed over a six-minute sliding window for the 37 syllables used >1% of the time in an example experiment. Syllables are sorted by total usage in the experiment, with the most-used syllable at the top and least used on the bottom. The colors above each segment of the plot indicate the time intervals used to compute the transition matrices in Extended Data Fig. 2g. **g**) State maps computed for each colored section of the example experiment shown in Extended Data Fig. 2f, summarizing the transition statistics between behavioural syllables, and demonstrating that transitions are also non-stationary over each imaging experiment. Each node is a syllable, and each line represents the transition from one syllable to the next (whose width specifies the observed likelihood of each transition, per the legend).

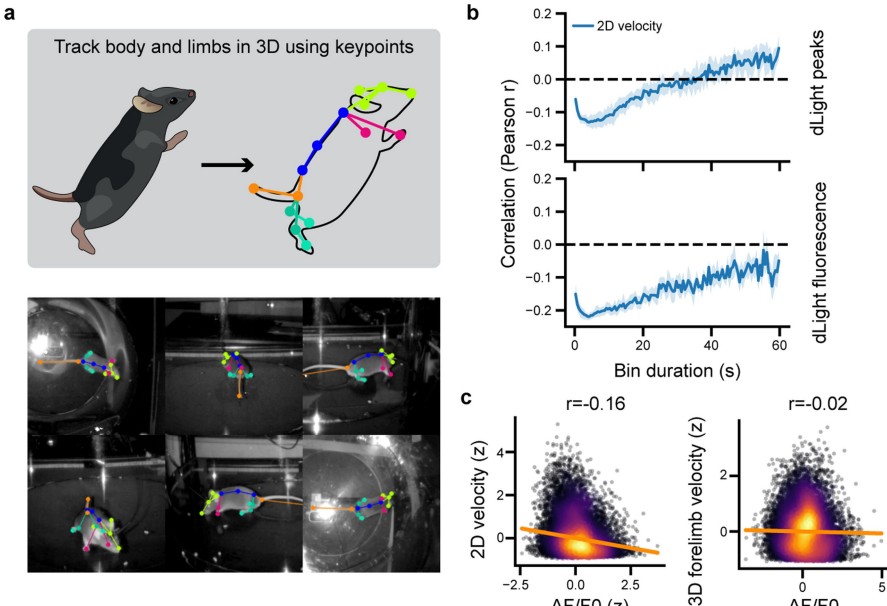

**Extended Data Fig. 3 | Validating correlations to kinematic variables through multi-camera keypoint tracking. a**) 15 keypoints were tracked in 3D using 6 infrared cameras (Azure Kinect, see Methods) positioned around the open field arena (n = 5 mice). A custom keypoint detection network was trained to identify all keypoints using manually labeled frames from each individual camera and integrated post-hoc with GIMBAL (see Methods). Top: schematic of keypoints positioned on the mouse. Bottom: example aligned frames from each of the 6 cameras with keypoints superimposed. **b**) Pearson correlation between 2D velocity and simultaneously recorded dLight at different timescales. 2D velocity was estimated by computing the centroid of the spinal keypoints in the X and Y plane for each frame (shown as the dark blue keypoints in Extended Data Fig. 3a) and taking the difference between centroid positions across frames. As in Fig. 1e, 2D velocity is negatively correlated with dLight transient

rates at short timescales, and positively correlated at long timescales. Top: Pearson correlations between 2D velocity and dLight transient rates across various time bins. Bottom: Pearson correlations between 2D velocity and average dLight fluorescence across various time bins. Shading represents 68% CI. **c**) Left: correlation between 2D velocity and dLight fluorescence after binning the data into 400ms time points (Pearson r = −0.16, p < .001 one-sided shuffle test). Each dot is a single 400ms time bin. Color represents point density, where brighter colors indicate denser points. Right: correlation between 3D forelimb velocity and dLight fluorescence after partialing out the relationship between dLight and other known kinematic parameters such as velocity and height (see Methods, Pearson r = −0.02, p < .001 one-sided shuffle test).

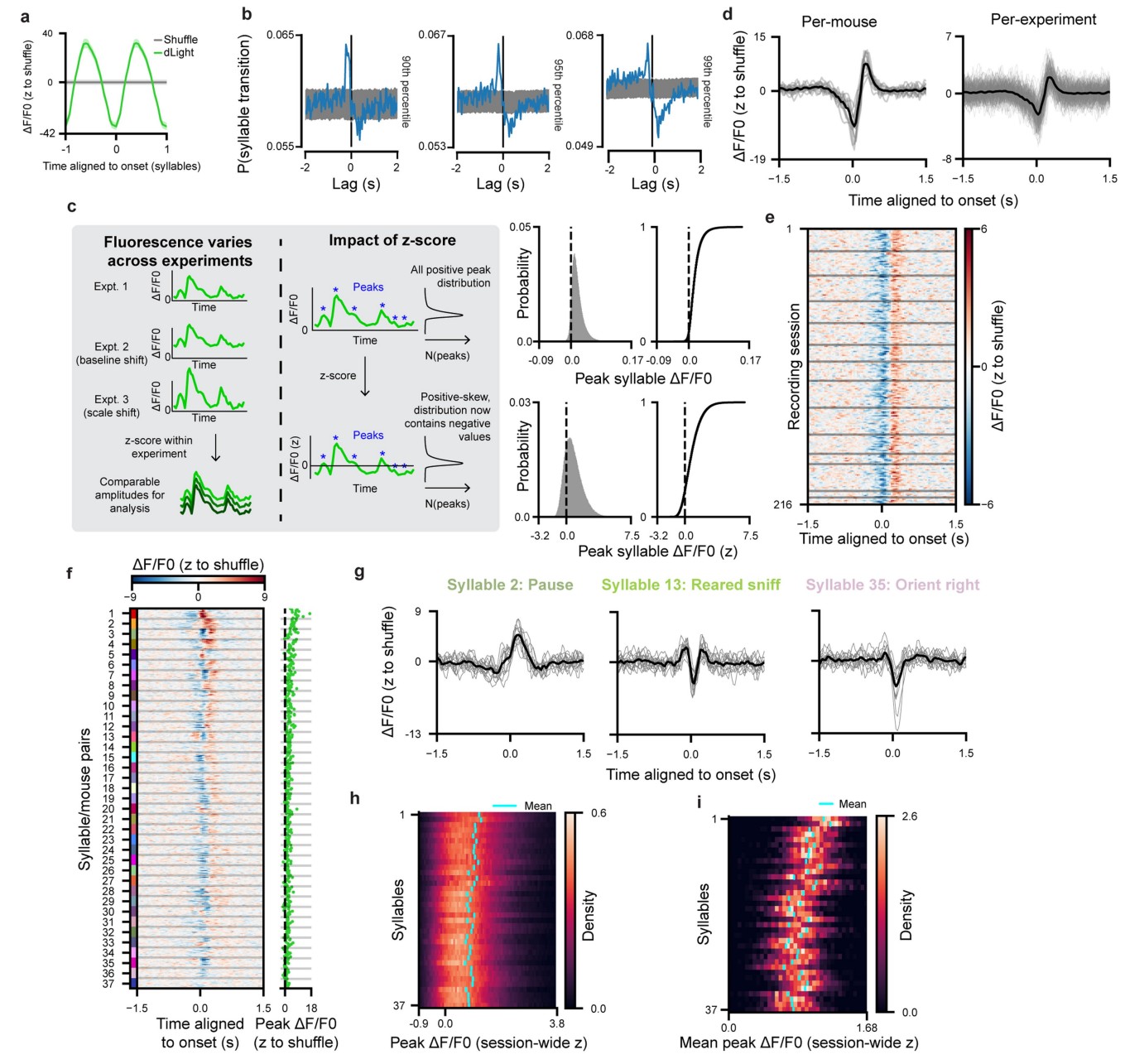

**Extended Data Fig. 4** | See next page for caption.

**Extended Data Fig. 4 | Variability of syllable-specific dLight waveforms across mice and experiments. a**) Average dLight fluorescence aligned to syllable onset after time warping (traces were warped using linear interpolation from syllable onset to syllable offset, see Methods). **b**) Average probability of a syllable transition occurring near a dLight peak across all experiments. Peaks in the dLight trace were identified by first computing and z-scoring the derivative of the ΔF/F0 trace, and identifying peaks as values that exceeded the 90th (left), 95th (middle), or 99th (right) percentile. We then plotted the probability that a syllable transition occurred given a dopamine peak. Gray shading indicates 95% bootstrap confidence interval of the shuffle. For the 95th percentile threshold, a syllable transition is likeliest to occur 200 ms prior to the dLight peak. However, the estimated dLight peak lags actual dopamine release by 10s of milliseconds (Extended Data Fig. 1e)[31]. **c**) Left: schematic showing forms of variability in dLight fluorescence measurements across experiments. dLight fluorescence assessed via photometry often exhibits baseline shifts and shifts in fluorescence scaling that can be normalized across experiments by z-scoring the fluorescence trace. Z-scoring dLight per experiment will have the effect of shifting the distribution leftward (and thus producing negative values). Top right: distribution of all syllable-associated dLight peaks across all mice and all experiments (left), and corresponding cumulative distribution (right). Bottom right: distribution of all syllable-associated dLight peaks across all mice and all experiments after z-scoring fluorescence traces from each experiment (left), and corresponding cumulative distribution (right). **d**) Left: assessing variability of the average dLight transient amplitudes from mouse to mouse. Shown is the average dLight amplitude aligned to syllable transitions Z-scored relative to a shuffle as in Fig. 1f. The thick black line indicates the average across all mice, and per-mouse averages are shown as thin gray lines. Right: same as left except averages across experiments are shown; the thick black line indicates the overall average, and the thin gray lines are per-experiment averages. **e**) Pseudo-color plots where each row depicts per-experiment average aligned to syllable onset for all experiments, grouped by mouse. Gray lines indicate boundaries between individual mice. **f**) Left: pseudo-color plot of all per-syllable dLight waveforms as in Fig. 1h, except shown for each mouse. The color bar on the left indicates which rows correspond to which syllables using the same sorting as Fig. 1h. Within a syllable-specific block, individual rows correspond to per-mouse average dLight waveforms (n = 518 syllable/mouse pairs). Average dLight waveforms are z-scored to a shuffle. Right: the syllable-associated peak dLight value for each row, computed from each waveform between 0–300 ms from syllable onset. **g**) Average dLight fluorescence aligned to syllable onset for three example syllables shown in Extended Data Fig. 4e; the thick black line indicates the mean across all experiments, with the thin grey lines indicating averages from each mouse. **h**) The probability of observing a syllable-specific dLight peak value across every syllable instance and across all mice and experiments. Syllable-specific peak values are computed using the maximum value in a 300 ms window after syllable onset. Here, color values indicate the likelihood of observing a specific peak dLight amplitude from trial to trial without averaging. Here, dLight is z-scored within each experiment. Cyan bars show the location of the overall average for each syllable. Rows are sorted in the same order as Fig. 1h and Extended Data Fig. 4f. **i**) The probability of syllable-specific average dLight peak values across experiments. Syllable-specific peak values are computed using the maximum value in a 300 ms window after syllable onset and averaged over the experiment, and thus do not correspond to the average waveform peaks in Extended Data Fig. 4e. Each row corresponds to a given syllable, and color values indicate the likelihood of observing a given peak dLight amplitude, on average, across experiments. Here, average dLight peaks are z-scored within each experiment. Cyan bars show the location of the overall average for each syllable. Rows are sorted in the same order as Fig. 1h and Extended Data Fig. 4f.

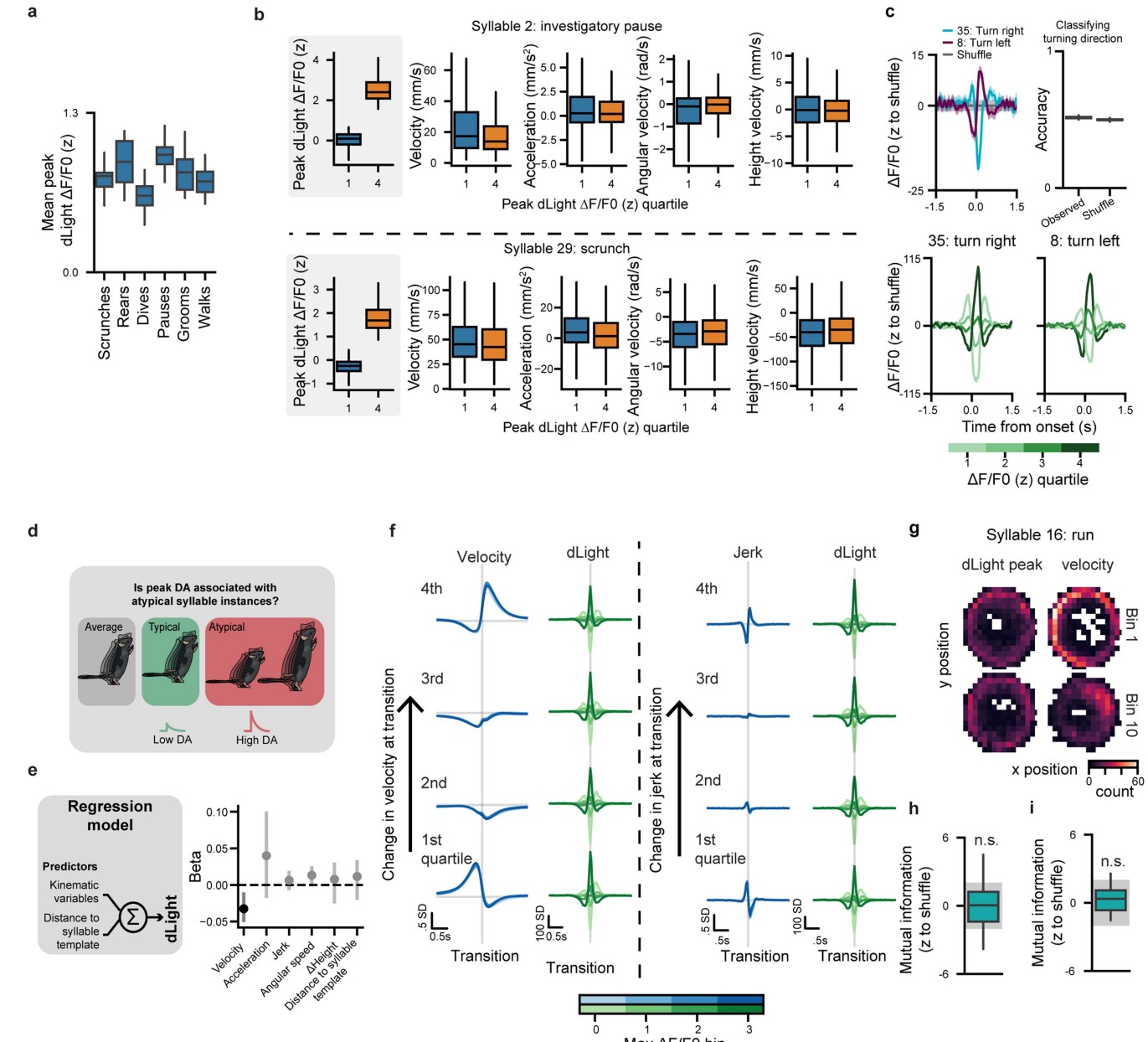

**Extended Data Fig. 5 |** See next page for caption.

**Extended Data Fig. 5 | Querying different possible sources of variability in dLight waveforms. a**) Average per-syllable dLight peaks associated with six behavioural categories (n = 7 dive syllables, 7 grooms, 9 pauses, 13 rears, 5 scrunches, 16 walks). Each category is associated with multiple syllables and were identified through human annotation. **b**) Within-syllable changes in kinematic parameters do not covary with peak dLight. Box plots of kinematic parameters binned by syllable-associated peak dLight – shown are the first and fourth quartiles. Kinematic variables were averaged from syllable onset to syllable offset, and box plots show the distribution of per-instance averages. Box plots for two examples syllables are shown, an investigatory pause (top; N = 15245 syllable instances) and a scrunch (bottom; N = 11838 syllable instances). **c**) Left top: average dLight fluorescence waveforms for two syllables that contain a left- (contralateral) and right-ward (ipsilateral) turning component. Consistent with prior studies indicating that elevated dopamine and striatal activity is associated with contralateral turning, we find higher average dLight levels are associated with contralateral turning[34,88]. Fluorescence traces were z-scored to a circular shuffle. Left bottom: dLight waveforms broken out into quartiles based on syllable-associated fluorescence, as in Fig. 1k. Right: performance of a linear SVM classifier predicting individual syllable instances as left or right turns. Average observed accuracy was 51%, indicating substantial instance-by-instance variability (p < .001, one-sided shuffle test). **d**) Schematic illustrating the hypothesis that dopamine fluctuations may reflect performance prediction errors; here "performance error" is defined as the degree to which a given syllable instance differs from its mean implementation (see Methods). **e**) Top: schematic describing the linear model used to characterize whether syllable rendition quality compared to a template provides additional information about dLight fluorescence on top of the kinematic parameters described in Fig. 1e. Bottom: model coefficient for each kinematic parameter. Significant parameters are shaded black (p < .001 two-sided shuffle test, n = 1000 bootstraps; error bars indicate 95% CI). **f**) Left half: Distribution of dLight waveforms across different velocity change bins. Syllable transitions were binned by the change in velocity from one syllable to the next. The peak magnitudes of dLight waveforms within each "velocity change" bin were then binned from lowest to highest; these binned dLight waveforms reveal the diversity of dLight transients associated with each behavioural transition type. Left: averaged velocity traces for each velocity change/dLight peak bin pair. Right: averaged dLight traces for each velocity change/dLight peak bin pair. Right half: Same as left half, but transitions were binned by their associated jerk, and waveform distributions are plotted as described above. Here inter-syllable jerk is used as a surrogate for the biomechanical difficulty mice are likely to experience as they transition across syllables. **g**) Syllable-associated dopamine peaks do not contain information about position in the open field. For each syllable, peak dLight and velocity were binned into ten equally spaced bins, and the animal's 2D centroid position in the arena was binned into four equally spaced bins. Then, mutual information was computed between the dopamine and the position bin. Shown are 2D histograms of mouse position for the highest and lowest bin for dLight peaks (left) and velocity (right) for an example syllable. **h**) Per-syllable mutual information between dLight per-syllable average peaks and position in the open field (p = .107, n = 57 syllables, one-sided test). The p-value was computed by comparing the average mutual information across all syllables against the mutual information computed on shuffled data. **i**) Specific syllable transitions do not contain information about the likelihood of a dopamine transient (p = .165, n = 14 mice, one-sided test). Here, we estimated the average likelihood of syllable-associated dLight peak crossing the 95th percentile for all syllable transitions. These likelihoods were used to build a 2D matrix, where cell $i, j$ was the likelihood of a transient for the transition from syllable $i$ to syllable $j$. Finally, we computed the mutual information of this matrix per-mouse, and estimated p-values by comparing with the mutual information computed on shuffled data.

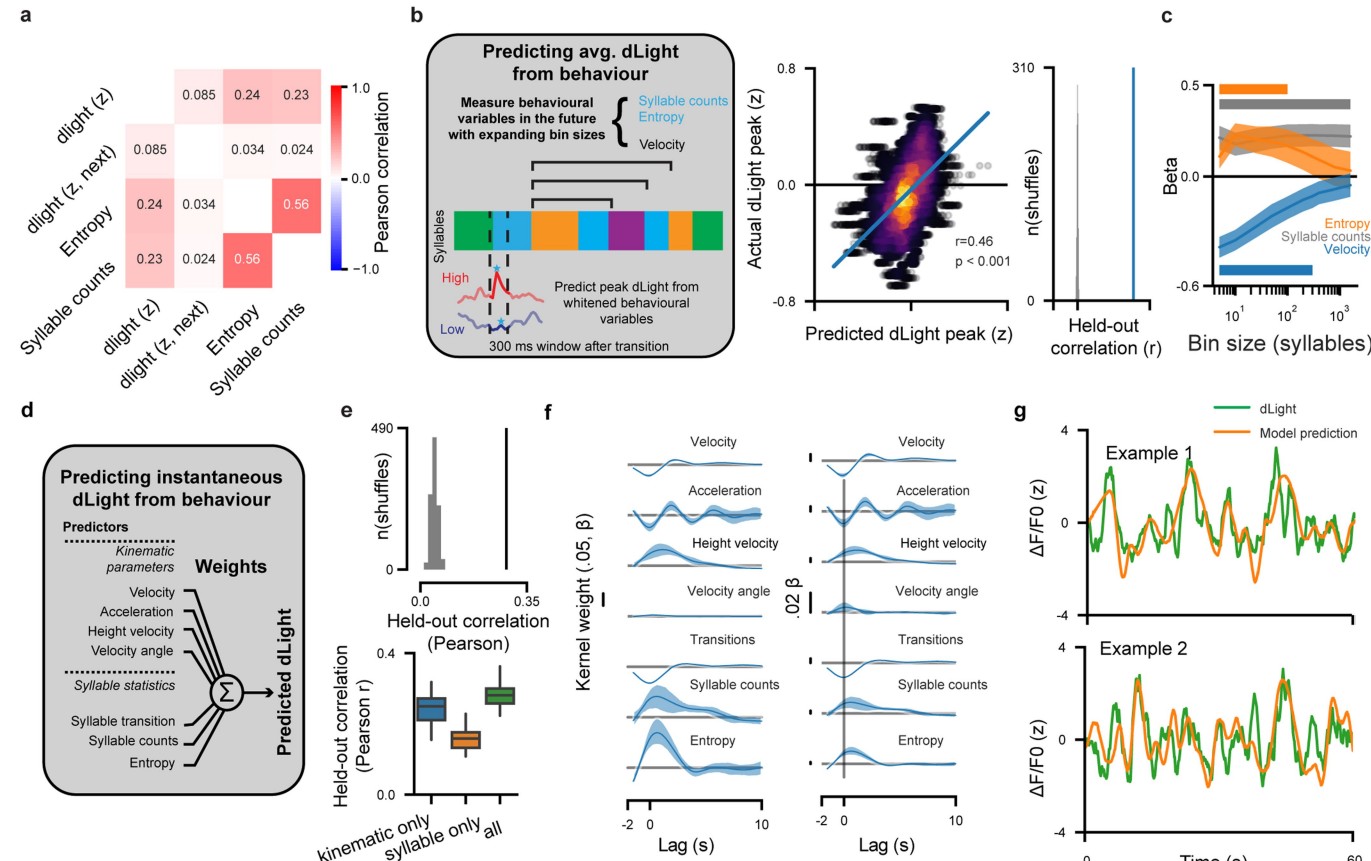

**Extended Data Fig. 6 | Dopamine predicts future syllable choices, and behaviour predicts prior dopamine dynamics. a)** Correlation matrix between dLight associated with a given syllable, entropy (which summarizes the variability of the subsequent syllable choice), syllable counts (for the syllable associated with dLight), and the dLight associated with the next syllable. Here, each feature was averaged per syllable/mouse pair, and the Pearson correlation was computed between feature averages (n = 760 syllable/mouse pairs). Syllable-associated dLight, entropy and syllable counts are all substantially correlated with each other, as described in the manuscript. Note that entropy (here defined as outbound entropy, the degree to which the subsequent syllable choice is predictable or variable) does not correlate with the amount of dLight on the subsequent syllable. This observation means that the amount of dopamine associated with a given syllable does not reflect whether that specific syllable was a more or less variable choice, given the preceding syllable; this contrasts with the correlation between syllable-associated dLight and outbound entropy, which demonstrates that the amount of dopamine associated with a given syllable predicts whether the next syllable choice will be deterministic or variable. **b)** Left: schematic for an encoding model which uses future behaviour to predict average syllable-associated dLight in the past (n = 760 syllable/mouse pairs, see Methods). Middle: plot of model predictions against actual dLight peak values on held-out data (5-fold cross-validation repeated 50 times). This model combines each feature at its best lag, lag = 10 syllables for velocity, 100 syllables for counts, and 10 syllables for entropy. Each point is a syllable/mouse pair, and the color of each point represents a kernel density estimate. Regression line is shown in blue. Right: the correlation between predicted syllable-associated dLight values and actual dLight values compared to n = 1000 shuffles (average Pearson correlation of held-out mouse/syllable pairs r = 0.46, p < .001; p-values for correlations throughout this figure were estimated by comparing observed correlation to Pearson correlation from shuffled data via a one-sided test). Performance

using kinematic parameters only, r = 0.39, counts and entropy only r = 0.22, both models p < .001 one-sided shuffle test. To evaluate model performance using feature subsets, we refit the model from scratch for each group of features using cross validation. **c)** Median beta coefficients of the encoding model shown in Extended Data Fig. 6b at increasing bin sizes. Shaded region indicates 95% confidence intervals for each behavioural variable across Markov-chain Monte Carlo samples. **d)** Schematic of a linear encoding model predicting instantaneous dLight fluorescence from future behaviour. In this model each behavioural variable is convolved with a learned kernel, with the result of each convolution summed to produce a predicted dLight trace (see Methods). **e)** Top: correlation between model predictions and true dLight fluorescence values (median correlation over all held-out experiments using all features r = 0.28, in black is model performance with experiment-permuted dLight traces, p < .001 shuffle test, n = 211 experiments). Bottom: model performance quantified as held-out correlation (2-fold cross-validation, Pearson r) shown using all behavioural variables ("all"), variables related to behavioural structure (syllable counts or transition entropy, "syllable only"), or kinematic parameters (velocity, angular velocity, height velocity, or acceleration, "kinematic only"). Held-out correlation was evaluated for each experiment (n = 211). To evaluate model performance using feature subsets, we refit the model for each group of features (median r over held-out experiments for kinematic parameters 0.23; syllable-related measures 0.16; all correlations p < .001, one-sided shuffle test). **f)** Representative kernels learned by the fitting procedure (with cross-validation, see Methods) for each behavioural variable. Left: kernels for all behavioural variables with the same scaling. Right: kernels y-axes are re-scaled according to the scalebar shown on the left to visualize temporal dynamics of each kernel. Error bars indicate 99% bootstrap confidence interval. **g)** Model prediction of instantaneous dLight fluorescence for two example held-out experiments. Green indicates observed dLight fluctuations over time, orange indicates model-predicted fluctuations.

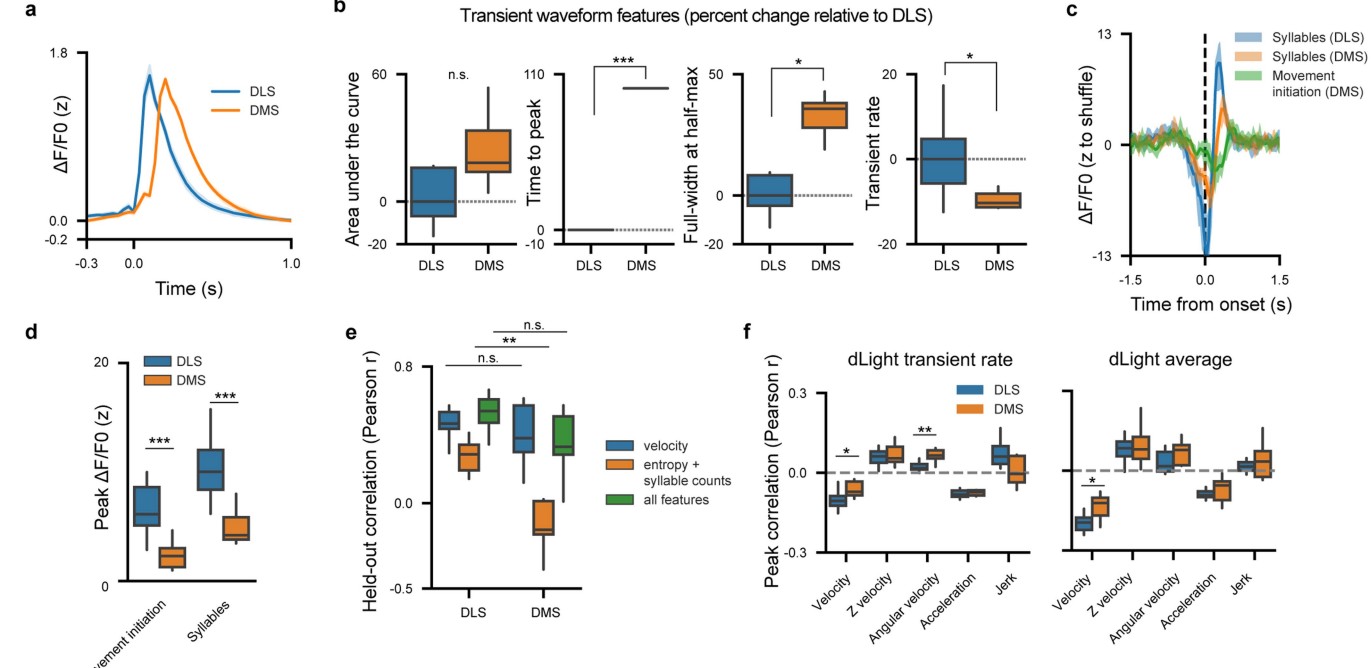

**Extended Data Fig. 7 | DMS dopamine does not correlate with syllable usage or entropy. a**) Average dopamine transient waveform for DMS and DLS. The average dopamine transient was computed for each DMS (n = 8) and DLS (n = 14) mouse, and then the per-mouse means were averaged to form the grand average. Shaded region indicates the 95% bootstrap confidence interval. As in Extended Data Fig. 1h, a transient is defined as when the dLight trace crosses 1 standard deviation above the mean. Note that DMS and DLS data are z-scored independently of each other. **b**) Summary statistics of DLS and DMS dopamine transients. Values were averaged per-mouse, thus leaving n = 8 for DMS and n = 14 for DLS. Box plots summarize per-mouse averages of each statistic. *Indicates p < .05, ** indicates p < .01, two-sided Mann-Whitney U test. Area under the curve, p = .052, U = 27, f=.24.; time to peak, p = .0001, U = 4, f = .036; Full-width at half-maximum p = .017, U = 16, f = .14; transient rate p = .017, U = 97, f = .87. **c**) As in Fig. 1f, average dorsomedial striatum (DMS) dLight fluorescence aligned to all syllable transitions (green) or movement initiations (orange) and z-scored to shuffle (n = 100 shuffles, see Methods; DLS shown for reference in blue). Shaded regions represent bootstrap SEM. **d**) Peak dLight values from per-mouse average waveforms aligned to either syllable transitions or movement initiations (n = 14 DLS mice, n = 8 DMS mice). *** Indicates p < .01, two-sided Mann-Whitney U test (movement initiations, p = 2.5e-5, U = 110, f = .98; syllables, p = 4.4e-5, U = 109, f = .97). **e**) Encoding model performance for dLight peak values (see Extended Data Fig. 6b, Methods) recorded in DLS (left) and in DMS (right) using different feature subsets. Each point is the average per-mouse heldout performance (n = 14 DLS mice, n = 8 DMS mice). ** Indicates p < .01, two-sided Mann-Whitney U test. Comparison of syllable feature subsets between DLS and DMS: entropy and syllable counts p = .002, U = 102, f = 0.91; velocity p = 0.87, U = 59, f = 0.53; all p = 0.25, U = 79, f = 0.71. **f**) Correlation for all kinematic parameters for DLS (n = 14 mice) and DMS (n = 8 mice) photometry mice. Here we computed the correlation between dLight and kinematic parameters and a variety of bin sizes as in Fig. 1e. The maximum correlation across all bin sizes per mouse is shown. *, p < .05 and **, p < .01, two-sided Mann-Whitney U test. Angular velocity comparison for dLight transient rate, p = 0.006, U = 9, f = 0.08; velocity comparison for dLight transient rate, p = .03, U = 15, f = 0.13; velocity comparison for dLight average, p = 0.026, U = 14, f = 0.125.

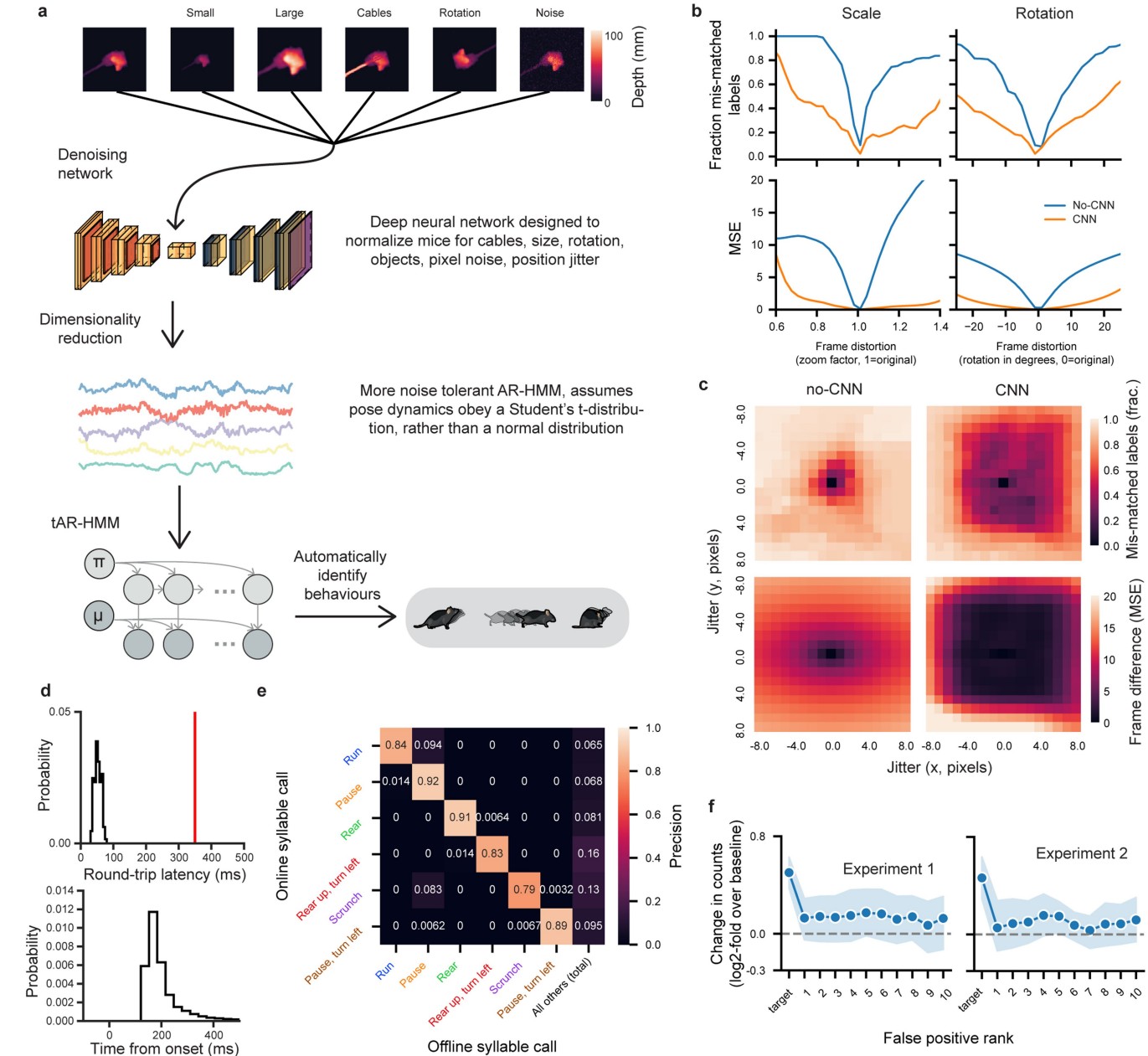

**Extended Data Fig. 8** | See next page for caption.

**Extended Data Fig. 8 | Closed-loop Motion Sequencing. a)** Schematic of the closed-loop MoSeq pipeline. A deep neural network (a convolutional neural net (CNN)), structured as a denoising autoencoder, was used to remove small artifacts related to photometric or optogenetic fiber optics. The network takes a depth image of a mouse to 1) remove artifacts like cables, 2) eliminate rotation or translation jitter, and 3) resize the mouse to a standard size for closed loop MoSeq. This network was trained to minimize the reconstruction loss of images of mice after those images were corrupted through artificial rotation, rescaling or imposition of noise (see Methods). After being passed through this denoising network, depth frames were dimensionally reduced by applying principal components analysis, where principal components were estimated using a size-and-age-matched "clean" dataset. Finally, the principal component scores were modeled using an autoregressive hidden Markov model (AR-HMM). For offline syllable detection, discrete latent states (i.e. behavioural syllables) were estimated using the Viterbi algorithm. For online syllable detection, the probabilities of discrete latent states were estimated using a forward pass estimated on a rolling basis (see Methods). **b)** Average pixel-wise or syllable label corruption plotted as a function of mouse depth image distortion via either a change in size or a change in rotation. Top: the impact of image corruption on syllable labels either with (orange) or without (blue) applying the CNN. Mouse depth images were corrupted through applying either a zoom factor (left, >1 indicates enlargement and <1 shrinking) or a rotation (right, in degrees), and syllable labels were compared between the corrupted and uncorrupted image using the Viterbi algorithm. Here, depth videos from a size-and-age-matched dataset were fed to the CNN after image scaling or rotation. This analysis reveals that the CNN effectively mitigates the effects of scale and rotation on the depth image. Bottom: the impact of image corruption measuring using the mean-squared-error (MSE) between the original depth image and the corrupted depth image. **c)** Similar to b, but measuring the robustness of depth images and syllable labels to jittering the mouse's position in X or Y (in units of pixels). Top: impact of position jitter on syllable labels without (left) or with (right) applying the CNN. Bottom: impact of position jitter measured using the MSE between the original depth image and the corrupted depth image without (left) or with (right) applying the CNN. **d)** Top: histogram of round-trip latency between receiving a depth frame and running all computations associated with the closed-loop pipeline (CNN, image processing, and AR-HMM likelihood estimation). Red line indicates the median syllable duration. Bottom: prediction time relative to the onset of the six syllables targeted for optogenetic reinforcement in this work. **e)** Degree to which the online system for syllable classification used during opto-DA stimulation confused the targeted syllable with other syllables. Shown is the row-normalized confusion matrix comparing online syllable calls (from actual experiments) against offline classification using traditional MoSeq (as in the remainder of the paper). The last column is the sum total of false alarms across all syllables that were not targeted for closed-loop reinforcement. **f)** Opto-DA learning is minimally impacted by false positives. Here, we show per-mouse average opto-DA learning for the targeted syllable (leftmost point of each plot), along with per-mouse averages of the 10 off-target syllables with the most false positives, ranked from highest to lowest (1 to 10). Off-target learning was smoothed with a 3-point rolling average. Results from the first stimulation experiment are shown on top, and results from the second experiment are shown on the bottom.

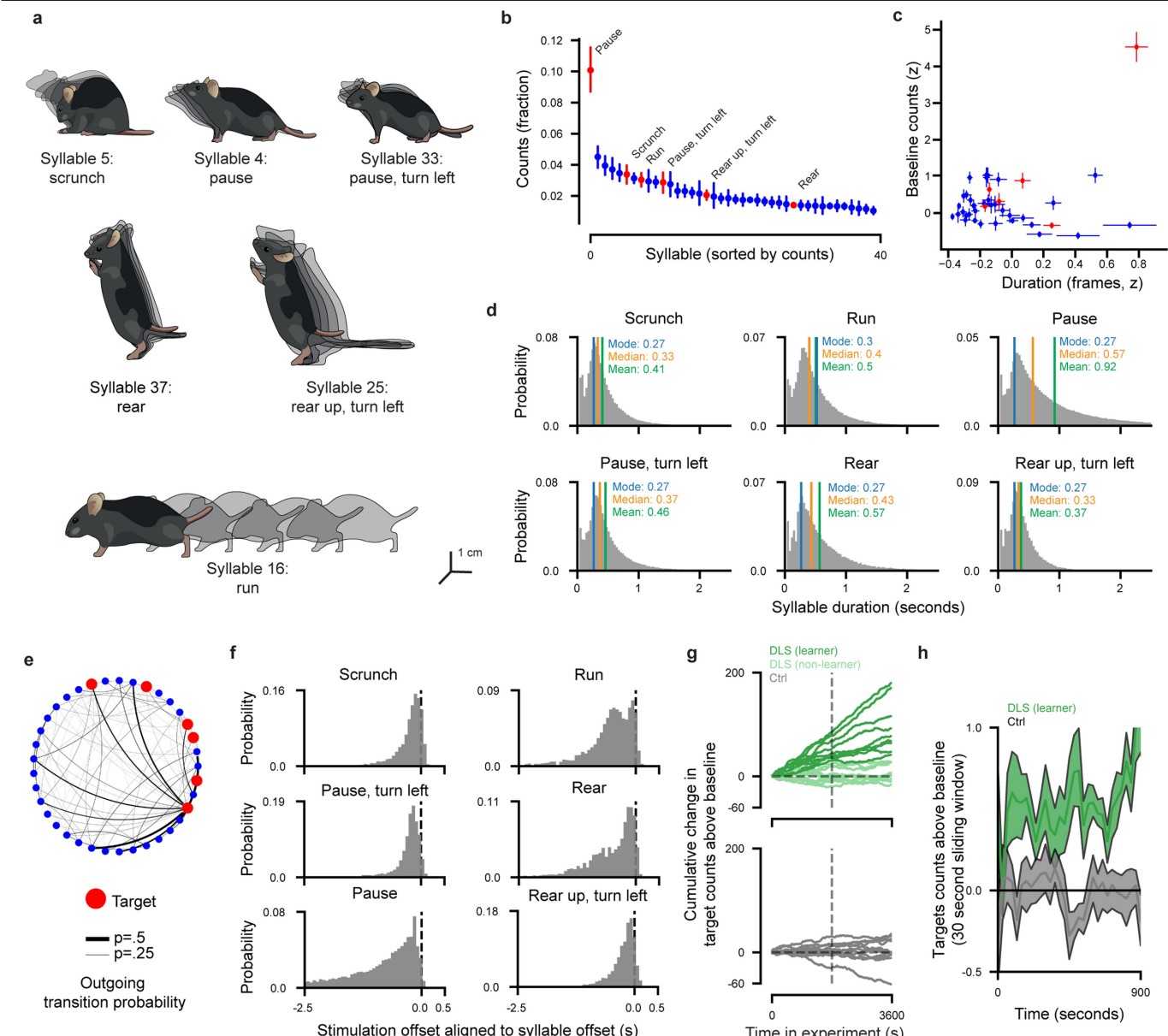

**Extended Data Fig. 9 | Closed-loop reinforcement of targeted syllables.**
**a)** Cartoons depicting the mouse pose dynamics expressed during the six syllables targeted for optogenetic reinforcement. **b)** Per-mouse average usage plot depicting the top 40 most used syllables identified by closed-loop MoSeq (used >1% of the time), rank-ordered by baseline usage, with target syllables are outlined and highlighted. Syllable usages were computed in counts for each mouse-experiment pair, and then averaged across these pairs for each syllable (n = 32 mice total, n = 20 opto-DA mice and n = 12 controls). Target syllables are labeled in red. Error bars represent 95% bootstrapped CI. **c)** Relationship between baseline syllable usage and syllable expression duration in no-opsin controls. Each point is a syllable, whose durations and usage counts were averaged across mouse-experiment pairs, and subsequently normalized across pairs (n = 40 syllables, Spearman r = −0.08, p = 0.61). Target syllables are labeled in red. Error bars represent SEM. **d)** Probability distributions for the duration of each target syllable across all behavioural experiments and mice. Mean, median, and mode values (in seconds) are reported. **e)** Circular state map computed for the full repertoire of behaviours the closed-loop system was able to faithfully detect. Each node is a syllable, and each line represents the transition from one syllable to the next (whose width specifies the observed likelihood of each transition). Each syllable targeted for optogenetic reinforcement is shown in red; each such node is associated with a different set of sequences in which it participates. **f)** Probability distributions describing the relative timing of optogenetic stimulation offset and the offset of the syllable instance for all target syllables. Note that optogenetic stimulation across targets rarely extends into the subsequent syllable. **g)** Cumulative target syllable counts over time. Lines are averaged over the six target syllables for each mouse. Dark green indicates "learners" that used the targeted syllables significantly above controls (n = 9/20, learners are defined as mice whose average cumulative change in counts across all syllables exceeds all control mice, see Methods). **h)** Timecourse of target syllable use during the first thirteen minutes of opto-DA. Depicted here is the usage of the target syllable (in counts) above baseline in a 30-second long non-overlapping bins. Mice quickly learn the contingency between expressing the targeted syllable and opto-DA, and then perform the target syllable at a near-constant rate above baseline.

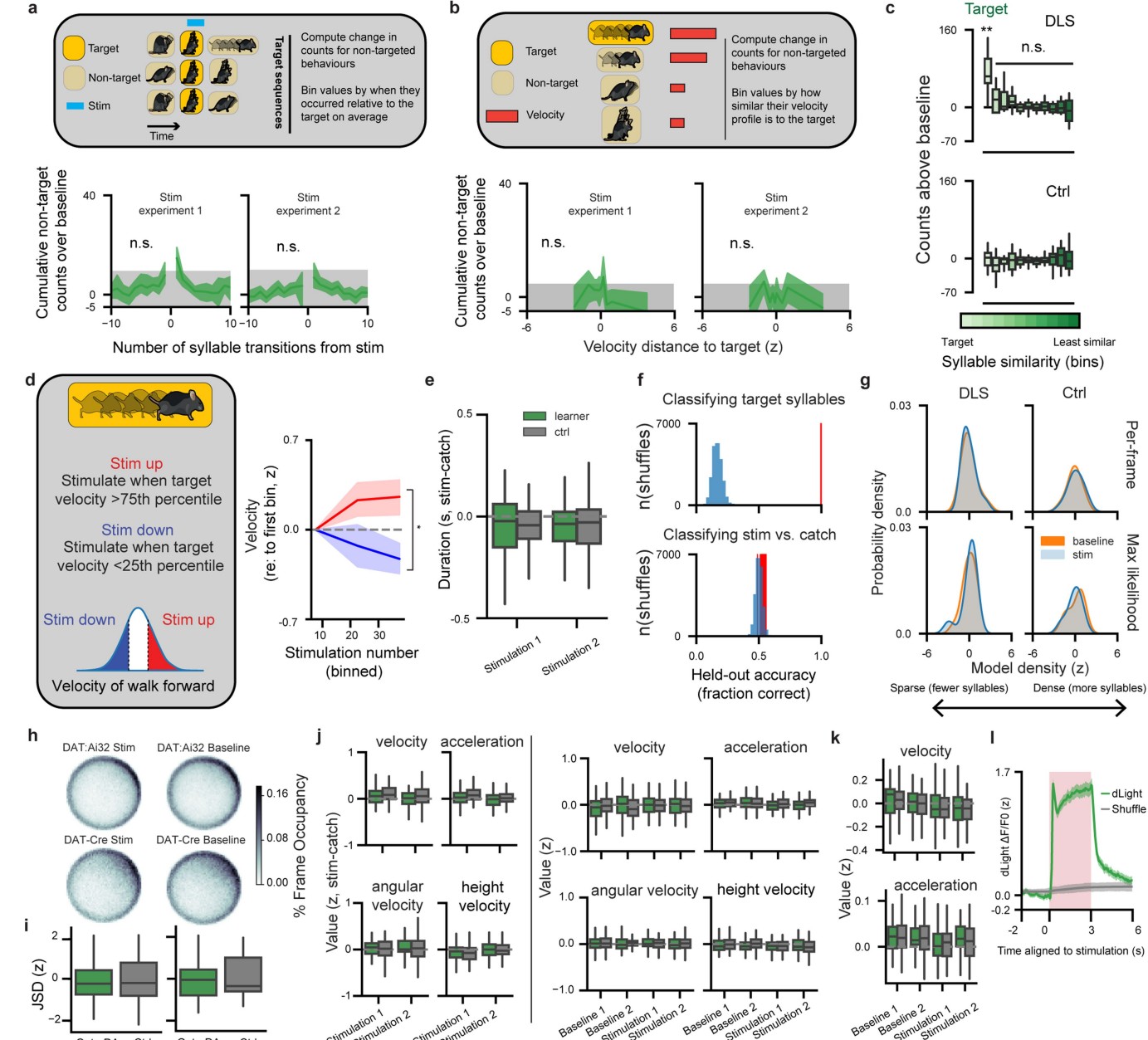

**Extended Data Fig. 10** | See next page for caption.

**Extended Data Fig. 10 | Reinforcement of the target syllable is spatiotemporally precise. a**) Top: schematic describing the hypothesis that optogenetically-evoked DA release influences syllable counts of temporally-adjacent non-targeted syllables. Bottom: weighted average of syllable counts over baseline for non-targeted syllables for the first (left) and second (right) stimulation experiments in learner mice (n = 9). Green and gray shading indicates 95% bootstrap CI for weighted average and time-shuffled data, respectively (n = 1000 shuffles, p > 0.05 for all comparisons, two-sided Mann-Whitney U test). **b**) Top: schematic describing the hypothesis that opto-DA reinforces similar-velocity syllables to the target. Bottom: average syllable counts over baseline for similar-velocity syllables for stimulation experiment 1 (left) and 2 (right) in learner mice. Green and gray shading indicates 95% bootstrap CI for weighted average and time-shuffled data, respectively (n = 1000 shuffles, p > 0.05 for all comparisons, two-sided Mann-Whitney U test). **c**) Relative usage change (in syllable counts) of syllables of varying behavioural similarity to the target syllable, with syllables grouped into 10 bins given their relative similarity to the target. Shown are per-mouse-and-bin medians. Top: learner mice. Bottom: no-opsin controls. ** Indicates a significant difference between opto-DA learners (n = 9) and control mice (n = 12) (p = 0.006, U = 103, f = 0.95 two-sided Mann-Whitney U test between median change in counts per learner mouse), all other comparisons p > 0.05. **d**) Left: schematic of velocity modulation experiment (see Methods). Right: mouse/experiment averages of the targeted syllable's velocity binned by stimulation number (for velocity, p = 0.013, u = 167, f = 0.77; n = 18 up experiments and n = 12 down experiments, two-sided Mann-Whitney U test). Error bars indicate bootstrap SEM. **e**) Per-mouse and per-target average target syllable duration, comparing learner mice to controls. Shown is the average duration on stim trials relative to catch trials (stim − catch); no statistically significant differences in duration distributions were identified (p = .98 for both sessions; session one U = 1995 and f = .52; session two U = 1804 and f = .46; two-sided Mann-Whitney U test, n = 144 mouse/target control pairs, n = 107 mouse/target learner pairs). **f**) Kinematic parameters associated with each target syllable were not altered as a result of opto-DA. Top: a linear classifier (linear discriminant analysis) was trained to use syllable-associated pose dynamics (measured using the mean and variance of the 10 principal components derived from the mouse depth data, see Methods) to predict the identity of the 6 target syllables; p < .001 established via a one-sided shuffle test. Bottom: linear classifiers trained on syllable-associated pose dynamics were unable to distinguish between stimulated and catch trials of single syllables in learner mice. Blue shows classifier performance on shuffled data, and red shows classifier accuracy over repeated cross validation splits; p = 0.069 established via a one-sided shuffle test. **g**) Stimulation of target syllables did not result in fractionated syllables or lowered detection confidence. Top: distribution showing entropy of cross-likelihoods for syllable detection for each frame, averaged across each experiment. Cross-likelihoods are a quantitative measure of confidence in assigning a given frame of behavioural data to a particular syllable. Distributions show density of average entropy of cross-likelihoods for baseline vs. stimulation experiments; these distributions show no evidence of changes in model confidence on experiments where syllables were targeted with optogenetic stimulation, consistent with opto-DA not substantially changing the kinematics associated with any given syllable in mice that learned. Bottom: distributions show probability density across baseline vs. stimulation experiments of entropy across maximum likelihoods of every syllable. No significant differences were found between stimulation and baseline distributions (all comparisons p > .05, two-sided 2-sample Kolmogorov-Smirnov test). **h**) Spatial histogram of frame occupancy of the centroid of the animal across stimulation and baseline experiments. Opto-DA mice (DAT-IRES-Cre::Ai32) on the left, no-opsin controls on the right. **i**) Left: Jensen-Shannon Divergence (JSD) of centroid location probability distributions across mice based on locations during stimulation trials (target performance) on stimulation day and simulated stimulation trials on baseline days (n = 192 mouse/target syllable pairs, p = 0.44, U = 4262, f = 0.49, two-sided Mann-Whitney U test across opto-DA mice and no-opsin controls). Right: JSD of centroid location distributions computed over experiment-wide centroid locations for each mouse (n = 32 mice, p = 0.41, U = 114, f = 0.48, two-sided Mann-Whitney U test). **j**) Distribution of kinematic parameters averaged per-mouse and per-target for the target syllable on both baseline and stimulation experiments. Left: difference between stimulation and catch trials for the targeted syllable on stimulation day. Right: magnitude of kinematic parameters for all trials across baseline and stimulation experiments. No significant differences were observed between learners and controls (p > .05, two-sided Mann-Whitney U test). **k**) Same as right half of Extended Data Fig. 10j (for velocity and acceleration), but for all non-target syllables. No significant differences were observed between learners and controls (p > .05, two-sided Mann-Whitney U test). **l**) Average dLight waveform aligned to the onset of 3-second pulsed stimulation (as elicited by ChrimsonR stimulation). Gray line indicates circular shuffle. Shaded error bars indicate 95% CI. Shaded red region indicates the duration of ChrimsonR stimulation.

**Extended Data Table 1 | List of reagents and resources**

| Reagent or Resource | Source | Identifier |
|---|---|---|
| Virus Strains | | |
| AAV5.CAG.dLight1.1 | Addgene | #111066 |
| AAV1.Syn.Flex.ChrimsonR.tdTomato | UNC Vector Core | |
| Experimental Models | | |
| Mouse: C57BL/6J | The Jackson Laboratory | JAX #000664 |
| Mouse: DAT-IRES-Cre [Slc6a3$^{tm1.1(cre)Bkmn}$] | The Jackson Laboratory | JAX #006660 |
| Mouse: Ai32 [Gt(ROSA)26Sor$^{tm32(CAG\text{-}COP4*H134/eYFP)Hze}$] | The Jackson Laboratory | JAX #012569 |
| Software | | |
| Motion Sequencing (MoSeq) | [4] | |
| NumPy | [60] | |
| Scikit-Learn | [61] | |
| TensorFlow | [62] | |
| JAX | [63] | |
| NumPyro | [64] | |
| SciPy | [65] | |
| Pandas | [66] | |
| RAPIDS AI | [67] | |
| Cellpose | [68] | |
| Seaborn | [69] | |
| Matplotlib | [70] | |
| Python | [71] | |

Table includes viruses, mouse lines, and software packages that were used in this study.

# Reporting Summary

## Statistics

For all statistical analyses, confirm that the following items are present in the figure legend, table legend, main text, or Methods section.

| n/a | Confirmed | |
|---|---|---|
| ☐ | ☒ | The exact sample size (*n*) for each experimental group/condition, given as a discrete number and unit of measurement |
| ☐ | ☒ | A statement on whether measurements were taken from distinct samples or whether the same sample was measured repeatedly |
| ☐ | ☒ | The statistical test(s) used AND whether they are one- or two-sided<br>*Only common tests should be described solely by name; describe more complex techniques in the Methods section.* |
| ☐ | ☒ | A description of all covariates tested |
| ☐ | ☒ | A description of any assumptions or corrections, such as tests of normality and adjustment for multiple comparisons |
| ☐ | ☒ | A full description of the statistical parameters including central tendency (e.g. means) or other basic estimates (e.g. regression coefficient) AND variation (e.g. standard deviation) or associated estimates of uncertainty (e.g. confidence intervals) |
| ☐ | ☒ | For null hypothesis testing, the test statistic (e.g. $F$, $t$, $r$) with confidence intervals, effect sizes, degrees of freedom and $P$ value noted<br>*Give P values as exact values whenever suitable.* |
| ☐ | ☒ | For Bayesian analysis, information on the choice of priors and Markov chain Monte Carlo settings |
| ☒ | ☐ | For hierarchical and complex designs, identification of the appropriate level for tests and full reporting of outcomes |
| ☐ | ☒ | Estimates of effect sizes (e.g. Cohen's *d*, Pearson's *r*), indicating how they were calculated |

*Our web collection on statistics for biologists contains articles on many of the points above.*

## Software and code

Policy information about availability of computer code

| Data collection | Custom code for data acquisition and pre-processing was written in MATLAB and Python (details in Methods) and is available at http://github.com/dattalab/dopamine-reinforces-spontaneous-behavior . |
|---|---|
| Data analysis | The following custom and publicly-available software packages were used, details are included in Methods section: Python 3+, Motion Sequencing, Numpy 1.21.5, 1.21.6 , Scikit-learn 1.0.2, TensorFlow, Jax 0.3.15, NumPyro 0.9.1, SciPy 1.7.3, 1.8.0, Pandas 1.3.5, 1.5.0, RAPIDS AI 21.12, Seaborn 0.11.2, 0.12.1, Matplotlib 3.5.1, 3.5.3, GIMBAL 0.0.1, Cellpose 1.0.2 |

For manuscripts utilizing custom algorithms or software that are central to the research but not yet described in published literature, software must be made available to editors and reviewers. We strongly encourage code deposition in a community repository (e.g. GitHub). See the Nature Portfolio guidelines for submitting code & software for further information.

## Data

Policy information about availability of data

All manuscripts must include a data availability statement. This statement should provide the following information, where applicable:
- Accession codes, unique identifiers, or web links for publicly available datasets
- A description of any restrictions on data availability
- For clinical datasets or third party data, please ensure that the statement adheres to our policy

The data that support the findings of the current study are available on Zenodo at DOI 10.5281/zenodo.7274803

# Human research participants

Policy information about studies involving human research participants and Sex and Gender in Research.

| | |
|---|---|
| Reporting on sex and gender | N/A |
| Population characteristics | N/A |
| Recruitment | N/A |
| Ethics oversight | N/A |

Note that full information on the approval of the study protocol must also be provided in the manuscript.

# Field-specific reporting

Please select the one below that is the best fit for your research. If you are not sure, read the appropriate sections before making your selection.

☒ Life sciences     ☐ Behavioural & social sciences     ☐ Ecological, evolutionary & environmental sciences

For a reference copy of the document with all sections, see nature.com/documents/nr-reporting-summary-flat.pdf

# Life sciences study design

All studies must disclose on these points even when the disclosure is negative.

| | |
|---|---|
| Sample size | Sample sizes were not pre-determined and are consistent with sample sizes typically used in the field for photometry and optogenetics experiments involving striatal dopamine release. For examples see: Howe and Dombeck, Nature 2016; Coddington and Dudman, Nature Neuroscience 2018. |
| Data exclusions | Photometry sessions were excluded if the signal did not exceed a pre-defined threshold (1.5 % dF/F), or if the Pearson correlation between signal and reference channels exceeded .6. |
| Replication | Results were replicated across animals. |
| Randomization | There were no treatment groups in our study, thus randomization was not required. |
| Blinding | Analysis was carried out automatically using Motion Sequencing, thus blinding was not necessary. |

# Reporting for specific materials, systems and methods

We require information from authors about some types of materials, experimental systems and methods used in many studies. Here, indicate whether each material, system or method listed is relevant to your study. If you are not sure if a list item applies to your research, read the appropriate section before selecting a response.

## Materials & experimental systems

| n/a | Involved in the study |
|---|---|
| ☒ | ☐ Antibodies |
| ☐ | ☒ Eukaryotic cell lines |
| ☒ | ☐ Palaeontology and archaeology |
| ☐ | ☒ Animals and other organisms |
| ☒ | ☐ Clinical data |
| ☒ | ☐ Dual use research of concern |

## Methods

| n/a | Involved in the study |
|---|---|
| ☒ | ☐ ChIP-seq |
| ☒ | ☐ Flow cytometry |
| ☒ | ☐ MRI-based neuroimaging |

# Eukaryotic cell lines

Policy information about cell lines and Sex and Gender in Research

| | |
|---|---|
| Cell line source(s) | HEK 293 cells (ATCC) |
| Authentication | Cells were obtained from ATCC, which authenticates their provenance via STR (short tandem repeat) analysis. |

| Mycoplasma contamination | Cells were not tested for mycoplasma contamination. |
|---|---|
| Commonly misidentified lines<br>(See ICLAC register) | No commonly misidentified cell lines were used in this study. |

# Animals and other research organisms

Policy information about studies involving animals; ARRIVE guidelines recommended for reporting animal research, and Sex and Gender in Research

| Laboratory animals | This study utilized wild-type, DAT-IRES-Cre (The Jackson Laboratory 006660) and Ai32 (The Jackson Laboratory 012569) mice, both male and female, between 6-15 weeks of age. Ambient temperature and humidity were maintained at 71 +/- 3 degrees Fahrenheit and 50% +/- 5 relative humidity, respectively. |
|---|---|
| Wild animals | This study did not involve wild animals. |
| Reporting on sex | For all experiments, both males and female mice were used, though sex was not an explicit factor considered for analytics. For dLight recording experiments, a total of n=18 males and n=4 females were used across groups. For reinforcement experiments, n=16 males and n=16 female mice were used across control and experimental groups. |
| Field-collected samples | This study did not involve samples collected from the field. |
| Ethics oversight | All experimental procedures were approved by the Harvard Medical School IACUC (Protocol # 04930) and were performed in compliance with ethical regulations of Harvard University and the Guide fir Animal Care and Use of Laboratory Animals. |

Note that full information on the approval of the study protocol must also be provided in the manuscript.

