## [Peer Review File · Nature]

Manuscript Title: Spontaneous behavior is structured by reinforcement without explicit reward

Reviewer Comments & Author Rebuttals

Reviewer Reports on the Initial Version:

Referees' comments:

Referee #1 (Remarks to the Author):

Markowitz, Gills, Jay, Datta and colleagues image DA signals in the dorsolateral striatum (DLS) to test if (and if so how) phasic DA signals are correlated with natural behavioral transitions in the absence of primary rewards or experimentally imposed task goals. They discover that DA transients (activations, suppressions) are common, exhibit magnitudes similar to those observed in reward tasks, and are non-randomly timed relative to behavioral modules, or 'syllables' - defined as units that cluster together in an unsupervised network analysis. Phasic activations predict future syllable usage over minutes and, seemingly independently and at a different timescale, promote sequence variability over seconds. Photoactivation of the DLS DA yields effects on behavior consistent with the spontaneous transients. An RL model captures the relationship between DA dynamics and behavior - with the specific feature that DA promotes reinforcement in the OFT (basically the same way it does in tasks) and also that DA promotes brief increases in behavioral variability that reduce the types of preservation observed when DA photostimulation reinforces an action, (e.g. a place or a lever press).

This paper will have its share of supporters and detractors - supporters will emphasize that this is the first study to show (in a mammal, at least (e.g. Gadagkar et al, 2016; Duffy et al, 2022 in songbirds))) that DA transients of reward-magnitude occur in the absence of primary rewards in ways that are more more sophisticated (and include a reinforcement component) than just about invigorating movement (e.g. as in findings from Yin, Costa labs in mice and early Romo/Scultz work in primates). Detractors will argue that the animal has ever-shifting objectives even in an open field and the experimental design is simply failing to provide the experimentalists with a sense as to what those objectives are, reducing interpretability of the results. Detractors may also point to the absence of circuit mechanisms of these DA transients or the DA-independent dynamics of the behavioral switching. Despite these caveats, we see these results as an important and timely contribution to the field. Given the pervasive impact of reward prediction error (RPE) based theories of DA function and RL on the study of biological and artificial learning, and the potential compatibility of these perspectives on the behavior of a mouse in the open field - this study is, in our opinion, important because it bridges decades of work on DA and RL implementation with the less accessible aspects of animal behavior outside the narrow task frameworks (e.g. single drives of hunger, thirst), typically imposed in experiments.

We see several important yet addressable problems that, if addressed, may increase the paper's impact and favor.

(1) MORE DETAIL ON SYLLABLE TYPES - and address differential conditionability

The paper has a big data approach that leverages statistical methods to connect DA transients to any behavioral correlate or consequence. This makes it very difficult to visualize what is actually happening - syllables are used throughout the manuscript as units of behavior but the actual behaviors (rearing, walking, stopping, sniffing, grooming) are so varied in their probable purposes (e.g. to scratch an itch for a groom versus to visualize the surround for a rear) - that it's hard to swallow the punchline (reinforcement and variability promotion) that the authors argue apply to all syllables. The reason this claim is met with skepticism is that the authors ignore and fail to cite relevant work from a relatively large field of the reinforcement literature known as differential conditionability. Since Thorndike, we've known that not all behaviors that animals exhibit are equally reinforceable (known as differential conditionability (e.g. scratching and rearing are readily reinforceable, yawning and grooming are not), e.g. Louboungou & Anderson, 1987; Charlton, 1983; Iverson et al, 1982; Seligman, 1970). Neither of these nor other relevant papers are cited nor is the phenomenon considered.

Because of a failure to address differential conditionability, it's not clearly ruled out if the observed effects on variability have to do with a dominating effect of some syllables in some animals that - in the aggregate - drive statistical significance of the findings. Were there some syllables that were 'immune' to reinforcement? For example, was groom probability in any way linked to DA transients? Such a result is not discernible from the paper in its present form. Taken in the context of a century's work on animal reinforcement - it's likely that syllables in the OFT vary dramatically in their reinforceability in a way that is related both to their association with DA signals and their ability be shaped by these signals.

To address this point, the authors could in principle compute for each syllable in each animal a reinforceability index to determine which syllables are consistently (across animals) reinforceable, which are not, or (possible but unlikely) find that they are all equally reinforceable. To clarify: the point here is not to steer the authors manuscript towards 'discovery' of which syllables are which in this context - the point here is for the authors to give the reader - especially those readers who care about animal behavior - a clearer picture of the actual behaviors that are driving their conclusions.

(2) Variation of DA transients across mice, across syllables, and within syllables within mice

2.1.1. It was confusing which results applied across versus within animals. Together, Figure 1g - where diverse syllables types are associated with diverse DA transients - and Fig 1h where distinct variations of the same syllable are associated with distinct DA transients, make for confusion as it's unclear if these plots are from a single or multiple animals. Please clarify if the relationship between syllable identity and average DA transient waveform was consistent or variable across animals - and please report if (and if so which) some were syllables-DA waveform relationships were consistent across animals whereas others were not? Within-animal within-syllable variation in DA magnitude (see more below) related to a syllable variation could reflect differences in evaluation alone, with variation scaled by predicted syllable quality. But this value would not generalize across animals - each of which has its own behavior and 'error' history. Thus if some syllables are reliably associated with phasic activations across animals - this would not be consistent with a performance prediction

error - which takes each animals' idiosyncratic error history into account.

2.1.2. In an across animal analysis - were syllables associated with phasic increases more or less likely to occur in a session?

2.1.3. Photoactivation experiments revealed that some mice were 'learners' and some were not. Were similar findings present in the DA imaging experiments? Did some mice have a stronger tendency for DA transients to drive future usage or variability than others? Ideally these two results (DA transient predicting future syllable usage and variability) could be summarized as two terms, and plotted for each mouse.

2.2 Distinct renditions of single syllables are also associated with distinct DA transients. Some simple additional analyses may help the reader interpret these interesting variations. For example, if you compare the kinematics of a rear that was associated with the upper 25% of DA transient with one associated with a lower 25% of DA transient (for that rear type) - do you see kinematic differences? Along these lines, some syllables are lateralized (e.g., Rear up, turn left; Orient left; etc.) - given that we know the inputs to striatum are lateralized and topographic, could the authors analyze whether there are stronger correlates between kinematics and DA transients relative to the side the authors recorded from in each animal? For example, we might expect to see stronger correlations between DA transients and orient left, if the authors recorded from right DLS, etc. One potential reason for low correlations between kinematics and syllables may be that they were collapsed across all syllables (Fig 1). Additionally, were some transitions from a specific syllable, say 'syllable X', to 'syllable Y' correlated with DA release, whereas transitions from another syllable, 'syllable Z', to 'syllable Y' correlated with suppression of DA? Was release dependent upon the time in session the syllable was performed? Finally, if you post-hoc ask mo-Seq to plot the position of a syllable, then color code each point by DA transient magnitude - does pattern emerge?

These are just examples/suggestions, but any additional analyses the authors can provide to clarify rendition-to-rendition changes in DA, aside from entropy and average syllable use in Fig 2, would strengthen the author's claims. Broadly, the goal of point 2.2 is for the authors to help the reader visualize possible behavioral correlates of the within-syllable DA variations. If all these analyses fail to capture any of these correlates - that's fine - it should just be made clear.

3. Consider moving Fig 3 to supplement to make space for a main figure on differential conditionability (just a recommendation)

It is not clear to us what the models in Fig 3 are teaching us beyond what was already shown in Figs 1-2. Given the correlation of DA transients with syllable identity and transition probability - could these models have possibly given us a different answer? What hypotheses are being ruled in or ruled out here? It reads like more evidence for correlation between DA and behavior and vice versa - but does not clarify the direction of the association (ie causality) nor does it give the reader a better picture of what animals are actually doing. Fig 3 title is DA predicts syll and behavior predicts DA. Though technically true, the correlations are relatively low and predict feels too strongly worded. Having just demonstrated that there are associations with DA and behavior and vice versa, it seems trivial that a decoding model would be able to make some predictions with non-zero accuracy. This

reader has no way of 'calibrating' or interpreting the magnitude of the correlation as behaviorally meaningful. What exactly is at stake if the value is 0.2, 0.3, or 0.4? And as these numbers are washed over all syllables - is it possible that some syllables have higher vals than others? How would these disintct outcomes affect the paper's claims / conclusions? As impressive as these correlations may be - they also highlight how many other mysterious and yet-to-be determined circuits/dynamics/states are influencing the behavior.

Fig 4c . These controls - adding the optoDA to the dynamic decoding model - are fine but again it's not clear what the model really provides / adds - is the punchline that optoDA works via similar mechanism as natural DA? If that's all this is - then I suggest moving it to supplement.

4. The DLS DMS comparison may be more fit for a main figure - it warrants more attention than the models because these data are actually testing an impotant hypothesis of distinct mesostriatal DA systems. Some questions about DMS/DLS comparisons:

Could the authors provide a comparison in the relationship between kinematics and DA from DMS vs. DLS? Was DA release correlated more strongly with acceleration - either movement in general, or acceleration of specific syllables, especially ones like locomotion - in DMS relative to DLS (Howe & Dombeck)?

It is surprising that there is no correlation between DA transients and syllables in DMS, given that the average dF/F plot in Extended Data Fig 9a shows qualitatively similar average transient magnitudes in DMS as in DLS (in Fig 1). By eye, the number of points for the plots showing a lack of correlation between DMS DA and syllable counts in Extended Data Fig. 9b appears to have less samples by around ~ 1 order of magnitude from those shown for DLS in Fig 2. Please just clarify these stats.

It would help the reader if a colormap plot for DMS aligned to syllable onset were included, as well as dF/F plots with example syllables as in Fig 1g for DMS. Additionally, if the others could compare the DA transients for the example syllables shown in Fig 1g for DMS data, this would help the reader visualize a dissociation (if it exists) in DA transients between DMS and DLS example syllable data. It would be helpful if the authors could also provide the number of animals and data point pairs for Extended Data Fig 9b, as they did in Fig 2d, to allow for a more direct comparison.

5. Online MoSeq was impressive and we have clarifying technical questions about the method and associated results

The optoDA experiments appear to lack a dependence of the transition that led into target syllable. This is fine but it's unclear - please just clarify that this was randomized (e.g. specific transitions were or were not not photoactivated or reinforced).

The labeling accuracy of the online closed-loop version of MoSeq in Extended Data Fig. 5 is also unclear. While the precision-recall curves in Extended Data Fig. 5e get to this point, could the authors show the error rate of syllable labeling across syllables (rather than within each syllable, as shown here) relative to the offline classification as well? For example, in Extended Data Fig. 3, there are two types of syllables labeled 'rear' among the 37 common syllables across animals. Was the

online version of MoSeq able to classify these two distinct rears, and was only 1 reinforced in these optoDA experiments or was both? Did labeling accuracy differ across animals?

In Extended Data Fig 5e, the authors state that 'Threshold per syllable were empirically chosen to avoid missing possible detections of the target' - while this makes sense to do, but this comes at the cost of potentially photoactivating on syllables that are not the target syllable. While the precision-recall analyses get to this point, could the authors provide descriptive statistics showing the accuracy of online MoSeq for the real sessions used in Fig 4? What was the probability of MoSeq accidentally labeling a target syllable as another syllable?

The across-animal error rates for online MoSeq are also unclear. Did labeling accuracy differ from mouse-to mouse? Might this also explain why $\sim\frac{1}{2}$ of the mice used in these opto experiments were non-learners? Was there a higher probability of online MoSeq confusing the target syllables with other non-target syllables for non-learners relative to learners?

Were there also any differences between non-learner and learner mice in terms of behavior? Could the authors provide additional analyses or speculate on why many mice did not 'learn' from the photoactivation? Perhaps baseline syllable usage of the targeted syllables was lower for these mice?

It would also benefit the reader if the authors could point this out in the paper as well - while the data are present in all of the figures associated with these experiments, there is no mention of the non-learner mice in text, and the paragraphs are written as if all of the mice learned.

Extended Data Fig. 7c - While the authors plot the photostimulation data sorted by syllable similarity, it would also be interesting to examine these data sorted by the syllables that online MoSeq most frequently (and incorrectly) labeled as the target syllable (e.g., the ones that were most often accidentally photostimulated due to labeling error) for each mouse? This may differ from mouse-to-mouse if there were differences in online MoSeq's labeling accuracy from mouse-to-mouse, so seeing this data per mouse would be helpful. Also, why are only 11 syllables plotted here? It would be helpful to see all syllables.

Minor / Technical Issues

It would be tragic if some future study discovered that the DA transient-syllable relationships observed here were not due to what the authors are claiming but instead due to some simpler kinematic parameter. The authors attempted to control for this with whole-body kinematic measurements but the forelimb may be a better control. Past work implicates DMS with projections to downstream MLR to drive or halt locomotion (e.g. Kreitzer) and a lateralized role for DLS in forelimb movements, and the topography of cortical areas projecting to DLS consists of frontal cortical areas associated with forelimb movement (Haber, Hintiryan/Dong, Peters/Carandini). Would there be stronger correlations between DA transients and kinematics if the authors examined kinematics of specific limbs instead of whole-body movements/postures? Might behaviors more heavily involving forelimb usage (grooming, paw-licking, etc.) correlate more strongly with DA transients than the individual syllables themselves? The authors could use DeepLabCut, SLEAP, or some other tracking software to track specific limbs in some open-field sessions and correlate the

kinematics of these limbs to DA transients to rule out this possibility, or rule it in in such a way that could be integrated into the present results.

Supp fig 2d. peak as threshold crossing on 95th percentile in the deriv of the df/f trace. it'd be nice to see this trace alongside a regular DF/F signal to gut-check how well this captures peaks; and which peaks are missed (is this grabbing only the biggest peaks? all peaks)? Do results change if you choose 90 or 99?

Fig 1g - Can the authors label the names of each of the two example syllables (e.g., rear 1, rear 2, etc.) shown in the middle and right panels? In fact, it would be useful to know the exact syllable paired with each number for this figure - this would provide the reader with some intuition as to what syllables were associated with increases vs. dips in dopamine.

Fig 1e plots an xcorr value of DA signal with a variety of kinematic parameters over an order of magnitude range of time bins (0-60 seconds). It's not clear what hypothesis this analysis is aiming to test. It has been suggested that DC offset of DA signals (e.g. 'tonic dopamine') fluctuates with experimental conditions / reward availability (e.g. Niv, Berke). Are the authors testing if tonic DA relates to any kinematic variable? Some more setup is necessary - as it is it just reads like a random analysis without a conclusion - other than the vague and timescale-independent summary that DA may 'integrate movement,' But even this was the case, why is there a zero crossing at 10 seconds?

For all figures where there are averages across animals (e.g., Fig 1f, 1g, etc.), it would be helpful to see individual average traces per animal either in a lighter shade behind the across-animal averages or in a supplemental figure. This would be helpful to get a sense of the across-animal variability in DA release for specific syllables.

It would also be helpful to show across session changes in average DA release across syllables. Did the average DA signal for the same syllable flip signs from session to session? If so, were there any quantifiable differences in behavior that differed from session to session that could help explain why?

Extended Data Fig. 2c (right) - Could the authors include in this plot the distribution of R^2 's post-fitting for comparison as well? It appears there are two shades of blue in this histogram - not sure if this figure was already comparing pre- & post-fitting with different shades of blue, or if this is a visual artifact from plotting. If it already does depict pre- & post-fitting with two shades of blue, this should be pointed out in the figure legend.

Extended Data Fig 3c, d, f, g - Could the authors label each syllable in these figures? It would be helpful for the reader to build an intuition on which syllable transitions were most probable. It would also be helpful to understand the directionality of these transitions. Some might be labeled in Extended Data Fig 6, but it would be helpful to label here as well.

Fig 2e & 3a: These figures depict that the syllable transitions analyzed in subsequent panels included time prior to the onset of the syllable. However, in the Methods, it states that transitions were

analyzed from the onset of the syllable to 300 ms after syllable onset. Which one is correct?

In Fig 1f and 1g, it appears that some syllables had phasic activity that preceded the onset of the syllable. If syllable transitions were analyzed from syllable onset to 300ms post-onset, would the results differ if time prior to the onset (the actual moment of transition) were included in the analysis window?

“Given that DA transients promote variability...” This sentence seems tailored to set up upcoming result and does not quite capture the cited papers or the role that DA is known to have on 'selecting variations' - be it place preference or syllable rendition. This is not the same as reinforcing variability - in fact DA transients (natural or photactivation) in past studies do not increase variability of choice - they reinforce a choice. So this section and the following result is confusing and not as in line with past work as the authors are suggesting. I think just minor re-wording of this setup would help.

Discussion

The idea that DA both reinforces and drives local variability to reduce perseveration is a compelling one. In fact the experimental and analytical sequence to derive this idea is beautiful. But it raises some questions that, if addressed in the discussion, could strengthen it. First, I don't understand how the results in Fig5 and the idea of maximizing DA are compatible with Fig 1 showing that more than half of the syllables are reliably associated with significant phasic decreases at time of transition. The simple RL model would predict that these syllables would be extinguished, and a few syllables would come to dominate the repertoire. But clearly there's some other DA independent process that is ensuring that the mouse distributes its behavior over the identified space of syllables.

It's also still unclear if the authors are imagining syllables are being evaluated (e.g. a good rear vs a bad rear) or if they have some value (rears are better than grooms). Multiple re-reads of the text do not clarify the author's arguments here. And it's not clear from the RL model in Fig 5 - at least in the main text - what is giving rise to the DA signals.

The most parsimonious explanation is that the DA system is always evaluating behavior - inducing local variability and reinforcing. But if this is the case, then each syllable should carry both its own predicted value signal as well as its outcome signal. This would predict phasic signals to occur within a syllable. Additionally, the DA transient could be evaluating the quality of the entry / transition into the syllable, which would predict DA transients closer to syllable onset. Finally, the whole syllable could be evaluated after its completion, which would predict modulations (dips or increases) at the end of a syllable. Fig 5 and the discussion leaves the reader confused as to which of the above interpretations are favored by the authors / results.

Minor discussion point: the discussion focuses on DA-modulated corticostriatal activity but thalamic inputs to striatum are also important to evoke - as I'm curious if open field behavior even looks different in decorticate mice. Most of the behaviors (locomotion, grooming, rearing) are known to be subcortically generated and depend on brainstem-thal-striatal-snr-brainstem loops (e.g. Berridge).

Referee #2 (Remarks to the Author):

Dopamine levels in the striatum have long been shown to reinforce task-structured behavior but a role for striatal dopamine signaling in shaping spontaneous, naturalistic behavior has not been demonstrated. Here, Markowitz et al. provide strong correlational, sufficiency, and modeling evidence that fluctuations in dorsolateral striatum (DLS) dopamine levels modulate behavior in the absence of task structure or reward.

Specifically, spontaneous behavior is decomposed into sequences of brief syllables and dopamine activity in the DLS is shown to fluctuate during syllable transitions. The peak amplitude of dopamine transients during a syllable predicts future syllable frequency and variability in sequence transitions. Using encoding models, Markowitz et al. predict average dopamine amplitude for a syllable, or dopamine amplitude at a given moment, from features of the behavior that follow. In a compelling experiment, closed-loop optogenetic stimulation of dopamine inputs to the DLS during particular syllables increases the occurrence of those syllables and the entropy of syllable transitions. Finally, Markowitz et al. build reinforcement learning models to test the hypothesis that DLS dopamine serves as a reward signal while increasing syllable variability to preserve behavioral flexibility.

The work is exciting and significantly broadens our understanding of dopamine's role in guiding behavior. The discovery that dopamine promotes the repetition of spontaneous behavioral syllables while also increasing behavioral variability is particularly novel. Below are a few points for the authors to consider:

1) In figure 1, DLS dopamine activity is shown to fluctuate during syllable transitions, often showing a phasic decrease in activity. Yet, in figures 2-4 the authors focus on the peak dopamine level measured during each syllable. The decision to focus on the peak amplitude during each syllable is at odds with figure 1g, which suggests that dopamine signaling is most associated with syllable transitions when there is usually a phasic decrease in signaling.

Why focus on dopamine signaling during specific syllables rather than syllable transitions? And why focus on the peak amplitude of dopamine signaling during syllables rather than on the larger and more prevalent phasic decreases in dopamine signaling? Does the amplitude of phasic dips in dopamine signaling during syllables and syllable transitions bear no consequence on syllable usage?

2) Less than half of the mice in figure 4e-f (the "learners") showed an increased occurrence of syllables that were paired with optogenetic DLS dopamine activity. An explanation for why a majority mice ("non-learners") did not demonstrate this effect is absent. Also absent is an explanation for why non-learners were excluded from further analyses. Why were mice split into learners and non-learners and what justification permits the exclusion of non-learners from analysis?

3) The authors paired optogenetic stimulation of dopamine terminals in the DLS with 6 specific behavioral syllables, reporting generally that these syllables subsequently increased in frequency (figure 4e-f). Can the authors show the relative increases in frequency of the 6 syllables for representative learners (and non-learners)? I'm curious if some syllables are more liable to increases

in frequency than others. Is there an underlying difference in the waveform of dopamine transients for syllables that did and did not increase in frequency? A figure identifying the syllables (e.g. rear, pause) that correspond to rows in figure 1g would be informative. I wonder, for example, if the syllables that increased after stimulation are also the syllables in the top few rows of figure 1g.

4) For mice that increased the frequency of syllables paired with optogenetic DLS dopamine stimulation, were all transitions to the target syllable increased, or only transitions that received optogenetic stimulation?

Minor corrections:

1) In figure 1e the right y-axis labels are reversed relative to what is indicated in the figure caption.

2) Figure 2a indicates that the mice are food restricted but the methods (bottom of page 39) indicate that mice were water restricted for 3-5 hours before the task.

3) In general, the methods are confusingly organized – it is difficult to discern which sections of the methods correspond to which figures.

4) In the discussion, the description of the types of computations that DLS dopamine transients might reflect are confusing. The paragraph at the end of page 14 could be edited for clarity.

Referee #3 (Remarks to the Author):

The manuscript by Markowitz et al., report the exciting results of a thorough and carefully conducted study of transient dopamine release in the dorsolateral striatum of mice during spontaneous behavior in a (relatively) cue-free environment. The authors make parallels between the role that dopamine in the striatum plays in the control of behavior in reward-maximization contexts, and the shaping of spontaneous behavior in the absence of external overt reward. They use fiber photometry of dopamine release, combined with their previously published discretization method of ongoing mice behavior into syllables, to show that during unstructured spontaneous behavior, dopamine released in the DLS appears to exert similar long term and short-term effects on its target striatal circuitry to those shown in more structured tasks. Physiologically, it is well established that dopamine has a dual effect: an immediate effect on excitability of SPNs (differentially impacting direct and indirect projections from the striatum), and a long-term effect on synaptic plasticity. In this study, the authors demonstrate both effects, and discuss their impact on behavioral choices with terminology derived from the framework of action choice in reinforcement learning.

The authors use the inter-syllable and intra-syllable variability of the amplitude of fluctuations in dLight fluorescence to show a correlation between the amount of dopamine release tied to a particular syllable and its subsequent occurrence over a minutes' time scale, and a negative correlation to overall velocity and to predictability of the following syllable on a seconds' time scale. They further examine causality by closed-loop optogenetic stimulation of dopaminergic axons,

triggered upon detection of individual syllables. The described experiments are meticulous, the analysis robust and the manuscript is well-written. The detailed Methods section is particularly commendable, as is the use of explanatory panels in the different figures. My issues, detailed below, have mainly to do with phrasing and interpretation.

1. First, I am a little confused about the main phenomenon. According to Fig. 1, syllable-associated responses are primarily reductions in dopamine release (in fact, it appears that on a per-syllable average, syllables are associated with either an elevation of DA release following the syllable transition, or (more frequently) reduction in release preceding transition. In that respect the average trace depicted in Fig. 1f is misleading). On the other hand, the reward-like time course of dLight peaks depicted in Fig. 2b shows clear elevations (as expected from the alignment to the time of the peak). The narrative also seems to implicitly regard DA fluctuations as upward deflections (hence the reinforcement lingo). Could it be that most fluctuations during behavior are not related to syllable transitions per-se, but rather to the syllable-related behavior itself? This is also in line with the authors' use of syllable detection to trigger excitation in dopaminergic axons during the syllables.
2. The combined results of the free-behavior and optogenetic experiments imply that dopaminergic surges in the DLS serve to reinforce activity in DLS circuits that coincided with this surge. As the authors and others have previously shown, activity in the DLS is tightly connected to the identity of specific behaviors and the vigor with which they are expressed. It therefore stands to reason that these forms of activity, and the associated behaviors, would be reinforced upon elevations of DA and diminished upon DA decreases. In that respect, it is unsurprising that the authors failed to detect a similar connection with DMS DA. It is likely that these fluctuations in DA (which may or may not coincide with those in the DLS – an interesting issue in itself) change the statistics of DMS activity, which would not necessarily be reflected by syllable-related statistics. Unfortunately, it is unknown (as far as I am aware) what DMS SPN activity looks like during spontaneous, task free and cue free behavior. I therefore find extended Fig. 9 unnecessary. However, if the authors do wish to keep DMS data in the paper, I would suggest showing the full distribution of dF/F_0 for different syllables (equivalent to Fig. 1g), as well as a more thorough attempt at characterization of the signal as a function of other variables that may be represented in DMS activity (e.g., action sequences, spatial variables (allocentric or egocentric), etc).
3. I am curious about the authors' thoughts about the nature of the DA signal in the context of spontaneous exploratory behavior. Is it merely random noise that hijacks the reinforcement-learning circuitry, thereby causing similar effects, or does the syllable-associated release encode a meaningful parameter?
4. I find the description of the RL model in Fig 5 lacking. Specifically, I do not understand which parameters were estimated in each model version, and how models with different numbers of parameters were compared. Generally, I like the notion of leveraging the dopamine signal to implement both the reinforcement and exploration elements of RL on two different timescales. As I mentioned above, it is in line with well-known physiological properties of DA innervation to the corticostriatal circuitry. However, I find the Q-learning model depicted in Fig. 5 to cause more confusion than add insight. If I understood the model correctly, the basic model does not contain an effective temperature term in the action selection function. If this is indeed the case, it is unsurprising that large rewards would lead the model to perseverate, and that introducing stochasticity to the action selection process would implement exploration, thereby rescuing behavior. The analysis presented in Fig. 5f-g does not convincingly show that another method of

dynamically changing tau that is unrelated to dopamine, or even simply tuning a constant temperature parameter to an intermediate value, would be inferior. Finally, I am not sure what to make of the finding that the DA transients correspond to the reward term rather than RPE (in the current formulation of the model), and I do not know how to interpret differences in z-scored log-likelihoods. However, it seems intuitively likely that the inability to detect RPE-like activity may have something to do with the fact that the DA signals were sampled randomly per syllable, and therefore could not track trial to trial fluctuations in prediction error.

Minor:

1. Why are the actual dLight values in Fig. 3a larger than in Fig. 2?
2. Legend to Fig. 3j – correlation between ...?
3. I am not convinced that the inability of a classifier to distinguish between opto-stimulated syllables and spontaneous syllables is sufficient proof that opto-stimulation did not affect kinematic properties (extended Fig. 8C)

Author Rebuttals to Initial Comments:

We thank the reviewers for their enthusiasm for our paper and especially for their excellent suggestions, which have substantially improved this manuscript. We have fully addressed all of the issues raised by the reviewers through new experiments and text revisions, and hope that our revised manuscript is now acceptable for publication. We address each of the reviewers' points in detail below; because we have added a number of analyses we offer a summary of the revision highlights here:

- All three reviewers were concerned that we were only including/analyzing instances in which syllable-associated DA increased rather than decreased. This was a simple misunderstanding due to the way we Z-scored the data in Figure 1, which misleadingly made it seem like dopamine falls during some syllables. This was an error of depiction only, and we thank the reviewers for encouraging us to switch to a more transparent description of the data. Essentially every syllable instance is associated with a positive dopamine transient (whose amplitude varies); it is the relative variation in the size of that positive dopamine transient that determines syllable use and sequence variability. We now present the data in Figure 1 and associated supplementals in such a way that the association between syllables and positive dopamine deflections is clear, and also include a cartoon in the supplemental figure to describe the consequences of z scoring (which is unavoidable with these sorts of data when merging across mice). Note that these changes to data depiction do not affect any of the downstream analyses or conclusions that were presented in our initial submission.
- All three reviewers asked that we investigate whether individual syllables differ in terms of their ability to be reinforced by opto-DA. This was a really fruitful suggestion — we find that indeed there is variability in the ability of opto-DA to reinforce syllables, and furthermore, we have determined why some syllables are more reinforceable: those syllables that are most reinforceable by opto-DA (across mice) are precisely those syllables that are associated with the highest endogenous dopamine. This finding is explained by our additive model (which predicts behavior from dopamine), which shows that the behavioral effects of exogenous DA essentially add to the effects of endogenous DA. We include a new main figure summarizing these findings. This finding provides a potential explanation for prior work (cited by Reviewer 1) regarding differential conditionability, which we now cite in the revision.
- Two reviewers asked about why some mice learn to associate syllables with opto-DA and others fail to make this association. To put this question in context, there is an extensive literature in which animals are trained in tasks in which some substantial fraction of animals fail to “meet criterion” and therefore are excluded from downstream analysis. While focusing on brain-behavior relationships in only those animals that meet criterion in a learning task is standard in the field, it remains unclear why some animals can easily learn a task while other (genetically identical and similarly housed and handled) animals fail to learn. Reviewer 1 asked a similar question, but about endogenous dopamine: are all mice equally sensitive

to endogenous dopamine? To address both of these questions we performed an extensive analysis of the subset of mice in which we both recorded DLS dopamine transients and manipulated dopamine through closed-loop optogenetics.

Gratifyingly, we find that those mice that are best able to learn from the exogenous dopamine afforded by opto-DA are also those that are most susceptible to the reinforcing effects of endogenous dopamine. This finding, which we include in a new main figure, suggests that mice are differentially sensitive to DLS dopamine (from any source, endogenous or exogenous), which in turn determines the extent of learning. There are many implications of this exciting finding that lie outside of the purview of this initial paper (which, thanks to the reviewers' excellent suggestions, we are now pursuing); that said, we are delighted to report this relationship in the revised manuscript, as it provides an affirmative explanation for why some mice learn and others do not.

- Reviewer 1 asked a series of questions whose theme was: can we discern any relationship between the specific behavior being expressed during a syllable and average or instance-by-instance dopamine transients? In other words, is our data really driven by grooms being associated with lots of dopamine rather than rears? We provide many additional new analyses — including all of the analyses explicitly asked for by the reviewer — and show that there is no apparent relationship between the type of movement being generated per se and dopamine. The reviewer also wondered whether forelimb movements were correlated with dopamine transients. To address this question we initiated a completely new set of experiments in which we recorded DLS dopamine while we performed three-dimensional point tracking using six cameras in an open field (rather than MoSeq analyses); these data validate our observation that velocity and dopamine are correlated, and further reveal that forelimb movements per se (when velocity is partialled out) do not substantially correlate with dopamine transient rates or amplitudes. These findings are consistent with our claim that endogenous fluctuations in DLS dopamine is important for syllable selection and sequencing rather than kinematic control.

- Concerns were raised about the relative amount of data that went into our comparisons of DMS versus DLS; to address this we have more than doubled the size of our DMS dataset.

We again thank the reviewers for their insightful and helpful comments – they have substantially improved and clarified the manuscript.

Referee #1 (Remarks to the Author):

Markowitz, Gills, Jay, Datta and colleagues image DA signals in the dorsolateral striatum (DLS) to test if (and if so how) phasic DA signals are correlated with natural behavioral transitions in the absence of primary rewards or experimentally imposed task goals. They discover that DA transients

(activations, suppressions) are common, exhibit magnitudes similar to those observed in reward tasks, and are non-randomly timed relative to behavioral modules, or 'syllables' - defined as units that cluster together in an unsupervised network analysis. Phasic activations predict future syllable usage over minutes and, seemingly independently and at a different timescale, promote sequence variability over seconds.

Photoactivation of the DLS DA yields effects on behavior consistent with the spontaneous transients. An RL model captures the relationship between DA dynamics and behavior - with the specific feature that DA promotes reinforcement in the OFT (basically the same way it does in tasks) and also that DA promotes brief increases in behavioral variability that reduce the types of preservation observed when DA photostimulation reinforces an action, (e.g. a place or a lever press).

This paper will have its share of supporters and detractors - supporters will emphasize that this is the first study to show (in a mammal, at least (e.g. Gadagkar et al, 2016; Duffy et al, 2022 in songbirds))) that DA transients of reward-magnitude occur in the absence of primary rewards in ways that are more sophisticated (and include a reinforcement component) than just about invigorating movement (e.g. as in findings from Yin, Costa labs in mice and early Romo/Scultz work in primates). Detractors will argue that the animal has ever-shifting objectives even in an open field and the experimental design is simply failing to provide the experimentalists with a sense as to what those objectives are, reducing interpretability of the results. Detractors may also point to the absence of circuit mechanisms of these DA transients or the DA-independent dynamics of the behavioral switching. Despite these caveats, we see these results as an important and timely contribution to the field. Given the pervasive impact of reward prediction error (RPE) based theories of DA function and RL on the study of biological and artificial learning, and the potential compatibility of these perspectives on the behavior of a mouse in the open field - this study is, in our opinion, important because it bridges decades of work on DA and RL implementation with the less accessible aspects of animal behavior outside the narrow task frameworks (e.g. single drives of hunger, thirst), typically imposed in experiments.

We see several important yet addressable problems that, if addressed, may increase the paper's impact and favor.

(1) MORE DETAIL ON SYLLABLE TYPES - and address differential conditionability

The paper has a big data approach that leverages statistical methods to connect DA transients to any behavioral correlate or consequence. This makes it very difficult to visualize what is actually happening - syllables are used throughout the manuscript as units of behavior but the actual behaviors (rearing, walking, stopping, sniffing, grooming) are so varied in their probable purposes (e.g. to

scratch an itch for a groom versus to visualize the surround for a rear) - that it's hard to swallow the punchline (reinforcement and variability promotion) that the authors argue apply to all syllables. The reason this claim is met with skepticism is that the authors ignore and fail to cite relevant work from a relatively large field of the reinforcement literature known as differential conditionability. Since Thorndike, we've known that not all behaviors that animals exhibit are equally reinforceable (known as differential conditionability (e.g. scratching and rearing are readily reinforceable, yawning and grooming are not), e.g. Louboungou & Anderson, 1987; Charlton, 1983; Iverson et al, 1982; Seligman, 1970). Neither of these nor other relevant papers are cited nor is the phenomenon considered.

Because of a failure to address differential conditionability, it's not clearly ruled out if the observed effects on variability have to do with a dominating effect of some syllables in some animals that - in the aggregate - drive statistical significance of the findings. Were there some syllables that were 'immune' to reinforcement? For example, was groom probability in any way linked to DA transients? Such a result is not discernible from the paper in its present form. Taken in the context of a century's work on animal reinforcement - it's likely that syllables in the OFT vary dramatically in their reinforceability in a way that is related both to their association with DA signals and their ability be shaped by these signals.

To address this point, the authors could in principle compute for each syllable in each animal a reinforceability index to determine which syllables are consistently (across animals) reinforceable, which are not, or (possible but unlikely) find that they are all equally reinforceable. To clarify: the point here is not to steer the authors manuscript towards 'discovery' of which syllables are which in this context - the point here is for the authors to give the reader - especially those readers who care about animal behavior - a clearer picture of the actual behaviors that are driving their conclusions.

There are two important points being raised here:

1. To what extent are the correlations we observe between endogenous DA and behavioral syllables homogenous across syllables? As requested by the reviewer, we now compute a reinforcement index per syllable; this index measures the maximum correlation between syllable use and dopamine on a syllable-by-syllable basis. As one might predict, there is variability in the degree to which different syllables are susceptible to endogenous dopamine; importantly, there is no clear relationship between the kinematics of each syllable (i.e., rears, runs, grooms) and reinforceability. We now report these results (and cite the differential conditioning literature) in the revised manuscript.
2. Are there syllables more or less susceptible to reinforcement by opto-DA?

We have now done the suggested analysis, and as for endogenous dopamine, there appears to be no relationship between the ability of a syllable to be reinforced and the type of behavior encoded by the syllable. Instead, we show that those syllables that are most susceptible to reinforcement by opto-DA are also associated with higher average endogenous DA. This is consistent with a simple model in which endogenous and exogenous dopamine add together to reinforce the expression of associated syllables. We present these results and the additive model explaining the interaction between endogenous and exogenous dopamine in our revised manuscript.

(2) *Variation of DA transients across mice, across syllables, and within syllables within mice*

2.1.1. *It was confusing which results applied across versus within animals. Together, Figure 1g - where diverse syllables types are associated with diverse DA transients - and Fig 1h where distinct variations of the same syllable are associated with distinct DA transients, make for confusion as it's unclear if these plots are from a single or multiple animals. Please clarify if the relationship between syllable identity and average DA transient waveform was consistent or variable across animals - and please report if (and if so which) some were syllables-DA waveform relationships were consistent across animals whereas others were not?*

Everything shown in Figure 1 is averaged across all mice and all sessions; we now make that clear in the main text and legends in the revised manuscript.

Here and elsewhere the reviewer asks about inter-session, inter-mouse and inter-syllable variability; we have generated compact representations of this variability and included plots summarizing this variation in the main and supplemental figures. Note that while these analyses will provide important context for our conclusions, none of these new analyses change the basic message: syllables are associated with different average dopamine transients (regardless of whether the data are averaged across mice, experiments or both), but these average associations mask a huge amount of instance-by-instance variability in the amplitude of the dopamine transients.

Within-animal within-syllable variation in DA magnitude (see more below) related to a syllable variation could reflect differences in evaluation alone, with variation scaled by predicted syllable quality. But this value would not generalize across animals - each of which has its own behavior and 'error' history. Thus if some syllables are reliably associated with phasic activations across animals - this would not be

consistent with a performance prediction error - which takes each animals' idiosyncratic error history into account.

There are many possible models for what causes the dopamine transients associated with each syllable, including performance prediction errors (PPE) as suggested here by the reviewer. The idea of the PPE derives from the birdsong literature; in that setting, particularly “good” versions of a syllable are associated with more DA, which then reinforce “correct” versions of the syllable. The analogy here in the mouse would be that DA should be high when the kinematic implementation of a given behavioral syllable is close to its mean implementation. We have done a new analysis exploring this exact question, which suggests that in the mouse there is no obvious relationship between the quality of implementation of a syllable (as assessed by how far a given implementation of a syllable deviates from the mean implementation) and the associated levels of DA. We include this analysis in the revised manuscript. We will also include in our revised discussion additional thinking about how a modified PPE (one that encodes variability in sequences rather than kinematics) might relate to the dopamine signals we observe in the mouse.

2.1.2. In an across animal analysis - were syllables associated with phasic increases more or less likely to occur in a session?

We address this directly in Figure 2d; we clarify in the revised figure legend that this panel includes multiple mice.

2.1.3. Photoactivation experiments revealed that some mice were 'learners' and some were not. Were similar findings present in the DA imaging experiments? Did some mice have a stronger tendency for DA transients to drive future usage or variability than others? Ideally these two results (DA transient predicting future syllable usage and variability) could be summarized as two terms, and plotted for each mouse.

This is a fabulous question. Per the reviewer's request, we have computed a reinforceability index (as we did for individual syllables), but here summarizing on a per mouse basis the strength of the correlation between dopamine and future syllable use. Importantly in all cases this association was positive, although as might have been predicted there was substantial variability in the degree to which endogenous dopamine influenced syllable usage across mice. We also performed this analysis in a subset of mice in which we also performed opto-DA; this allowed us to ask whether sensitivity to endogenous dopamine related to the ability of opto-DA to reinforce syllables at the mouse level. We find that those mice whose behavior was particularly susceptible to endogenous dopamine fluctuations were also those in which opto-DA was most effective; this suggests that mice are differentially sensitive to the behavioral effects of DLS dopamine, which explains

difference in opto-DA learning. We now present these data in the revised manuscript.

In addition, we have computed an index that summarizes the relative influence of dopamine on sequence variability. As for syllable usage, this index varied across mice. Importantly, the ability of endogenous dopamine to influence syllable usage in an individual mouse was positively correlated with its ability to influence sequence variability. We include this plot (as requested by the reviewer) as a main figure in the manuscript.

2.2 Distinct renditions of single syllables are also associated with distinct DA transients. Some simple additional analyses may help the reader interpret these interesting variations. For example, if you compare the kinematics of a rear that was associated with the upper 25% of DA transient with one associated with a lower 25% of DA transient (for that rear type) - do you see kinematic differences? Along these lines, some syllables are lateralized (e.g., Rear up, turn left; Orient left; etc.) - given that we know the inputs to striatum are lateralized and topographic, could the authors analyze whether there are stronger correlates between kinematics and DA transients relative to the side the authors recorded from in each animal? For example, we might expect to see stronger correlations between DA transients and orient left, if the authors recorded from right DLS, etc. One potential reason for low correlations between kinematics and syllables may be that they were collapsed across all syllables (Fig 1).

We have performed the exact analysis suggested by the reviewer, and find that there is no relationship between the kinematics of a given syllable instance and the size of the dopamine transient during that instance. This analysis is included in the revised manuscript. We always record from the right hemisphere, and as you might predict, there is more dopamine associated with left turning syllables than right turning syllables on average, and more dopamine associated with left turns than right turns on average (assessed independent of MoSeq). However, there is still substantial variation in the dopamine transients observed within left and right turns, and this variation is so substantial that a classifier can only marginally predict a left or right turn syllable from dopamine transients alone; these data are included in the revised manuscript.

Additionally, were some transitions from a specific syllable, say 'syllable X', to 'syllable Y' correlated with DA release, whereas transitions from another syllable, 'syllable Z', to 'syllable Y' correlated with suppression of DA?

As we show in the manuscript, on average predictable syllable sequences are associated with less dopamine, so broadly speaking X->Y and Z->Y transitions will differ in their associated dopamine depending upon their relative probabilities. However, the relationships between dopamine and

predictability are independent of the specific identity of the syllables participating in any given sequence; we now include a supplemental panel making this clear.

Was release dependent upon the time in session the syllable was performed?

We do not observe systematic changes in the number of dopamine transients or their average amplitudes over time during an experiment; we now report this in the revised manuscript.

Finally, if you post-hoc ask mo-Seq to plot the position of a syllable, then color code each point by DA transient magnitude - does pattern emerge?

We do not observe any clear relationship between where in space a given syllable is performed and its associated dopamine transient; we include these data in the revised manuscript.

These are just examples/suggestions, but any additional analyses the authors can provide to clarify rendition-to-rendition changes in DA, aside from entropy and average syllable use in Fig 2, would strengthen the author's claims. Broadly, the goal of point 2.2 is for the authors to help the reader visualize possible behavioral correlates of the within-syllable DA variations. If all these analyses fail to capture any of these correlates - that's fine - it should just be made clear.

We have included all of the suggested analyses in the revised manuscript, and are confident at this point that there is nothing about the identity or kinematics of a given syllable that meaningfully relates (within the level of resolution that is possible given our analysis) with the average dopamine amplitude (except left/right variation, which is a consequence of the lateralization of our recordings); this is consistent with the alternative explanation that instance-by-instance variation in endogenous dopamine acts as a reinforcer.

It is not clear to us what the models in Fig 3 are teaching us beyond what was already shown in Figs 1-2. Given the correlation of DA transients with syllable identity and transition probability - could these models have possibly given us a different answer? What hypotheses are being ruled in or ruled out here? It reads like more evidence for correlation between DA and behavior and vice versa - but does not clarify the direction of the association (ie causality) nor does it give the reader a better picture of what animals are actually doing. Fig 3 title is DA predicts syll and behavior predicts DA. Though technically true, the correlations are relatively low and predict feels too strongly worded. Having just demonstrated that there are associations with DA and behavior and vice versa, it seems trivial that a decoding model would be able to make some predictions with non-zero accuracy. This reader has no way of 'calibrating' or interpreting the magnitude of

the correlation as behaviorally meaningful. What exactly is at stake if the value is 0.2, 0.3, or 0.4? And as these numbers are washed over all syllables - is it possible that some syllables have higher vals than others? How would these disintct outcomes affect the paper's claims / conclusions? As impressive as these correlations may be - they also highlight how many other mysterious and yet-to-be determined circuits/dynamics/states are influencing the behavior.

The main thing we get from the models shown in the old Figure 3 (but not correlation analysis alone) is a. the ability to orthogonalize e.g., usage, entropy and velocity from each other and to therefore understand their relative influence, and b. the ability to make moment-to-moment predictions of dopamine-behavior relationships. We feel these points are important but agree they are a bit for the afficianados, and so we have moved this figure to supplemental data to make space for the new main figure suggested by the reviewer.

Fig 4c . These controls - adding the optoDA to the dynamic decoding model - are fine but again it's not clear what the model really provides / adds - is the punchline that optoDA works via similar mechanism as natural DA? If that's all this is - then I suggest moving it to supplement.

Although we agree that in our initial submission the importance of the “additive” model was unclear, given the experiments assessing differential conditionability during opto-DA it now has new relevance, as it explains why those syllables that are associated with the highest average endogenous dopamine transient amplitudes are those that are most susceptible to opto-DA reinforcement. We have the moved the additive model into the new figure describing those findings.

3. *The DLS DMS comparison may be more fit for a main figure - it warrants more attention than the models because these data are actually testing an impotant hypothesis of distinct mesostriatal DA systems. Some questions about DMS/DLS comparisons:*

Could the authors provide a comparison in the relationship between kinematics and DA from DMS vs. DLS? Was DA release correlated more strongly with acceleration - either movement in general, or acceleration of specific syllables, especially ones like locomotion - in DMS relative to DLS (Howe & Dombeck)?

It is surprising that there is no correlation between DA transients and syllables in DMS, given that the average dF/F plot in Extended Data Fig 9a shows qualitatively similar average transient magnitudes in DMS as in DLS (in Fig 1). By eye, the number of points for the plots showing a lack of correlation between DMS DA and syllable counts in Extended Data Fig. 9b appears to have less samples by around ~1 order of magnitude from those shown for DLS in Fig 2.

Please just clarify these stats.

It would help the reader if a colormap plot for DMS aligned to syllable onset were included, as well as dF/F plots with example syllables as in Fig 1g for DMS. Additionally, if the others could compare the DA transients for the example syllables shown in Fig 1g for DMS data, this would help the reader visualize a dissociation (if it exists) in DA transients between DMS and DLS example syllable data. It would be helpful if the authors could also provide the number of animals and data point pairs for Extended Data Fig 9b, as they did in Fig 2d, to allow for a more direct comparison.

There was something of a difference of opinions amongst the reviewers, with another reviewer advocating for eliminating the DMS panels entirely given our focus on DLS. After much thought and additional experiments, we have decided for clarity to include as a supplemental figure a more detailed analysis of average endogenous DMS dopamine-syllable relationships. To perform this analysis, we nearly tripled the size of our DMS recording dataset. The results reveal that the timescale at which dopamine fluctuates in DMS is slower than in DLS, and that DMS waveforms are broader than in DLS; similar results have recently been reported on bioRxiv by Josh Burke. Because of this difference in timescales, average syllable-associated DMS dopamine amplitudes are much smaller than in DLS and do not predict syllable usage in the future. Taken together, these data argue that thinking about “syllable-associated” dopamine fluctuations in DMS isn’t really possible in a meaningful sense, because there are fewer dopamine transients in DMS, and those that are present do not fluctuate at the syllable timescale. Because of the differences in timescales and waveforms between DLS and DMS, we have decided to remove our opto-DA experiments in DMS, which were difficult to interpret because limiting opto-DA stimulation to a single targeted syllable requires us to use brief DLS-like stimulation conditions. We have instead performed a full analysis of the relationship between endogenous DMS dopamine and continuous kinematic variables (mostly for reference). We have generated a new supplemental figure which we include in our revised manuscript that details these results.

4. Online MoSeq was impressive and we have clarifying technical questions about the method and associated results

The optoDA experiments appear to lack a dependence of the transition that led into target syllable. This is fine but it’s unclear - please just clarify that this was randomized (e.g. specific transitions were or were not photoactivated or reinforced).

The optoDA experiments are not conditioned on any prior syllables. We clarify this in the revised methods.

The labeling accuracy of the online closed-loop version of MoSeq in Extended Data Fig. 5 is also unclear. While the precision-recall curves in Extended Data

Fig. 5e get to this point, could the authors show the error rate of syllable labeling across syllables (rather than within each syllable, as shown here) relative to the offline classification as well? For example, in Extended Data Fig. 3, there are two types of syllables labeled 'rear' among the 37 common syllables across animals. Was the online version of MoSeq able to classify these two distinct rears, and was only 1 reinforced in these optoDA experiments or was both? Did labeling accuracy differ across animals?

In Extended Data Fig 5e, the authors state that 'Threshold per syllable were empirically chosen to avoid missing possible detections of the target' - while this makes sense to do, but this comes at the cost of potentially photoactivating on syllables that are not the target syllable. While the precision-recall analyses get to this point, could the authors provide descriptive statistics showing the accuracy of online MoSeq for the real sessions used in Fig 4? What was the probability of MoSeq accidentally labeling a target syllable as another syllable?

The across-animal error rates for online MoSeq are also unclear. Did labeling accuracy differ from mouse-to mouse? Might this also explain why ~½ of the mice used in these opto experiments were non-learners? Was there a higher probability of online MoSeq confusing the target syllables with other non-target syllables for non-learners relative to learners?

Were there also any differences between non-learner and learner mice in terms of behavior? Could the authors provide additional analyses or speculate on why many mice did not 'learn' from the photoactivation? Perhaps baseline syllable usage of the targeted syllables was lower for these mice? It would also benefit the reader if the authors could point this out in the paper as well - while the data are present in all of the figures associated with these experiments, there is no mention of the non-learner mice in text, and the paragraphs are written as if all of the mice learned.

Extended Data Fig. 7c - While the authors plot the photostimulation data sorted by syllable similarity, it would also be interesting to examine these data sorted by the syllables that online MoSeq most frequently (and incorrectly) labeled as the target syllable (e.g., the ones that were most often accidentally photostimulated due to labeling error) for each mouse? This may differ from mouse-to-mouse if there were differences in online MoSeq's labeling accuracy from mouse-to-mouse, so seeing this data per mouse would be helpful. Also, why are only 11 syllables plotted here? It would be helpful to see all syllables.

We thank the reviewer for raising these points, as it highlights some confusion in the way we described the accuracy of the system. As suggested by the reviewer we now provide descriptive statistics in which we quantify the accuracy of online MoSeq as assessed by offline calls with traditional MoSeq. This reveals that the true positive rate for individual

targeted syllables is very high, which in turn suggests that mis-targeted syllables (whether kinematically related to the target or not) should not be contributing substantially to our results. To directly assess this, we also now as requested by the reviewer ask whether the syllables most likely to be hit erroneously are reinforced, and find they are not. We further assess the specificity of our manipulation by asking whether syllables that are kinematically similar to the target (i.e., if we are targeting a rear, then another rear) are reinforced, and again find that they are not. We include all of these analyses in the revised manuscript. Because off-target rates are low and do not lead to reinforcement, analyzing effects per mouse aren't really informative (and so to avoid paper bloat we do not include these data) — and as detailed above we already have a much more compelling explanation for inter-mouse variation in opto-DA learning. Note that we now also make clear that the 11 “syllables” are actually bins of morphologically-related syllables.

Minor / Technical Issues

It would be tragic if some future study discovered that the DA transient-syllable relationships observed here were not due to what the authors are claiming but instead due to some simpler kinematic parameter. The authors attempted to control for this with whole-body kinematic measurements but the forelimb may be a better control. Past work implicates DMS with projections to downstream MLR to drive or halt locomotion (e.g. Kreitzer) and a lateralized role for DLS in forelimb movements, and the topography of cortical areas projecting to DLS consists of frontal cortical areas associated with forelimb movement (Haber, Hintiryan/Dong, Peters/Carandini). Would there be stronger correlations between DA transients and kinematics if the authors examined kinematics of specific limbs instead of whole-body movements/postures? Might behaviors more heavily involving forelimb usage (grooming, paw-licking, etc.) correlate more strongly with DA transients than the individual syllables themselves?

The authors could use DeepLabCut, SLEAP, or some other tracking software to track specific limbs in some open-field sessions and correlate the kinematics of these limbs to DA transients to rule out this possibility, or rule it in in such a way that could be integrated into the present results.

We have done both experiments suggested — we have fractionated syllables based upon those that are more forelimb dependent (like grooming), and asked whether these syllables are associated with more DA; we find there is no systematic relationship between forelimb-enriched syllables and average dopamine levels. We have also implemented 3D point tracking, which shows that forelimb movements per se do not seem to predict dopamine levels (when considered independent of translational velocity, which as we have shown does predict dopamine levels).

Supp fig 2d. peak as threshold crossing on 95th percentile in the deriv of the df/f trace. it'd be nice to see this trace alongside a regular DF/F signal to gut-check how well this captures peaks; and which peaks are missed (is this grabbing only the biggest peaks? all peaks)? Do results change if you choose 90 or 99?

Our results don't qualitatively depend upon the specific threshold chosen; we have added traces reflecting the 90 and 99th percentile as the reviewer requested to demonstrate this.

Fig 1g - Can the authors label the names of each of the two example syllables (e.g., rear 1, rear 2, etc.) shown in the middle and right panels? In fact, it would be useful to know the exact syllable paired with each number for this figure - this would provide the reader with some intuition as to what syllables were associated with increases vs. dips in dopamine.

As noted above, there are actually no dips — this was an artifact of the way in which we z scored those data for the plots in Figure 1; we have revised Figure 1 to correct this mis-communication. We have also as requested labeled the waveforms in that Figure with the associated behavior for clarity.

Fig 1e plots an xcorr value of DA signal with a variety of kinematic parameters over an order of magnitude range of time bins (0-60 seconds). It's not clear what hypothesis this analysis is aiming to test. It has been suggested that DC offset of DA signals (e.g. 'tonic dopamine') fluctuates with experimental conditions / reward availability (e.g. Niv, Berke). Are the authors testing if tonic DA relates to any kinematic variable? Some more setup is necessary - as it is it just reads like a random analysis without a conclusion - other than the vague and timescale-independent summary that DA may 'integrate movement,' But even this was the case, why is there a zero crossing at 10 seconds?

In the revised manuscript we clarify the writing describing Figure 1, which we hope helps to better convey our intent in showing those analyses. In the panel the author is wondering about, we are simply trying to characterize how dopamine, when binned at different timescales, correlates with various aspects of the mouse's kinematics (independent of syllables), since that is what people have looked at in the past. Our data are interesting and worth reporting, as they reconcile one set of observations — that dopamine seems to pause when animals initiate a movement — with another set of observations — that systematically increasing dopamine increases movement. We see both effects, but at different timescales. We never use language about tonic dopamine (or contrast tonic and phasic) because photometry by its nature is typically not appropriate for making such comparisons.

For all figures where there are averages across animals (e.g., Fig 1f, 1g, etc.), it would be helpful to see individual average traces per animal either in a lighter shade behind the across-animal averages or in a supplemental figure. This

would be helpful to get a sense of the across-animal variability in DA release for specific syllables. It would also be helpful to show across session changes in average DA release across syllables. Did the average DA signal for the same syllable flip signs from session to session? If so, were there any quantifiable differences in behavior that differed from session to session that could help explain why?

As mentioned above, we now include in the revised manuscript plots that compactly represent the degree to which DA-syllable relationships are variable across syllables, sessions and mice. In general average relationships between syllables and dopamine are stable, but there is substantial overlap between the distribution of dopamine transient amplitudes across syllables, which explains our inability to predict specific behaviors based upon dopamine transients.

Extended Data Fig. 2c (right) - Could the authors include in this plot the distribution of R^2 's post-fitting for comparison as well? It appears there are two shades of blue in this histogram - not sure if this figure was already comparing pre- & post-fitting with different shades of blue, or if this is a visual artifact from plotting. If it already does depict pre- & post-fitting with two shades of blue, this should be pointed out in the figure legend.

We have addressed these points in the revised version of the manuscript.

Extended Data Fig 3c, d, f, g - Could the authors label each syllable in these figures? It would be helpful for the reader to build an intuition on which syllable transitions were most probable. It would also be helpful to understand the directionality of these transitions. Some might be labeled in Extended Data Fig 6, but it would be helpful to label here as well.

Yes – we now provide a numerical code keyed to the list in Extended Data Fig. 2b.

Fig 2e & 3a: These figures depict that the syllable transitions analyzed in subsequent panels included time prior to the onset of the syllable. However, in the Methods, it states that transitions were analyzed from the onset of the syllable to 300 ms after syllable onset. Which one is correct?

In all cases we characterize transients from onset to 300ms after onset; we clarify this in the revised methods.

In Fig 1f and 1g, it appears that some syllables had phasic activity that preceded the onset of the syllable. If syllable transitions were analyzed from syllable onset to 300ms post-onset, would the results differ if time prior to the onset (the actual moment of transition) were included in the analysis window?

As addressed above and in our revised manuscript, in nearly all cases syllable-associated DA transients are positive; the appearance of a “negative transient” is the consequence of the specific way in which we depicted the data in Figure 1 in the original manuscript (through z scoring relative to the overall distribution of transients). Given that — as we now clarify in the revised manuscript — this positive transient peaks during syllable expression, we focus on the 0-300 ms bin that captures the peak in our analyses.

“Given that DA transients promote variability...” This sentence seems tailored to set up upcoming result and does not quite capture the cited papers or the role that DA is known to have on 'selecting variations' - be it place preference or syllable rendition. This is not the same as reinforcing variability - in fact DA transients (natural or photactivation) in past studies do not increase variability of choice - they reinforce a choice. So this section and the following result is confusing and not as in line with past work as the authors are suggesting. I think just minor re-wording of this setup would help.

We apologize for our poor choice of language in that particular sentence, and we have revised it to more accurately reflect the literature. What has been shown previously (and what we cite) is that elevation of dopamine (through d-amphetamine or through D2 agonists) can increase the variation (e.g., entropy) of ongoing behavioral sequences; this role for an elevation of dopamine in behavioral variability is thought to be independent of a role for dopamine in reinforcement of future behaviors. Our use of the word “transient” in the original description mis-stated the current state of the field; indeed our work is (as far as we know) the first to show that dopamine transients (rather than chronic elevations of dopamine) can influence sequence variability.

Discussion

The idea that DA both reinforces and drives local variability to reduce perseveration is a compelling one. In fact the experimental and analytical sequence to derive this idea is beautiful. But it raises some questions that, if addressed in the discussion, could strengthen it. First, I don't understand how the results in Fig5 and the idea of maximizing DA are compatible with Fig 1 showing that more than half of the syllables are reliably associated with significant phasic decreases at time of transition. The simple RL model would predict that these syllables would be extinguished, and a few syllables would come to dominate the repertoire. But clearly there's some other DA independent process that is ensuring that the mouse distributes its behavior over the identified space of syllables.

This is an important misunderstanding that we have clarified in the revised text; as mentioned above, this was about how we depicted the data in Figure 1, which we have since changed and made much more transparent.

It's also still unclear if the authors are imagining syllables are being evaluated (e.g. a good rear vs a bad rear) or if they have some value (rears are better than grooms). Multiple re-reads of the text do not clarify the author's arguments here. And it's not clear from the RL model in Fig 5 - at least in the main text - what is giving rise to the DA signals.

In the discussion we propose several models for how the DA signals might arise, and articulate evidence for and against each; we have expanded this discussion in a revised manuscript for clarity. One idea supported by this reviewer is that DA serves to “evaluate” behavioral performance; this idea originates from the birdsong literature, where syllables that are too high- or low-pitched are associated with less DA, and accurate syllable renditions are associated with more DA. It is important to note that our new analysis suggests that the accuracy of individual behavioral syllable renditions is unlikely to be specifying levels of dopamine; instead, as we mention in the discussion, it is possible that the mouse is trying to match an idealized template sequence, where “errors” in sequencing correspond to more dopamine. It is also important to note that RL models can only speak to how DA fluctuations are interpreted — they provide no insight per se into how they arise (and accordingly we do not use the RL model to propose a mechanistic origin for the DA transients that are observed).

The most parsimonious explanation is that the DA system is always evaluating behavior - inducing local variability and reinforcing. But if this is the case, then each syllable should carry both its own predicted value signal as well as its outcome signal. This would predict phasic signals to occur within a syllable. Additionally, the DA transient could be evaluating the quality of the entry / transition into the syllable, which would predict DA transients closer to syllable onset. Finally, the whole syllable could be evaluated after its completion, which would predict modulations (dips or increases) at the end of a syllable. Fig 5 and the discussion leaves the reader confused as to which of the above interpretations are favored by the authors / results.

There are many possible explanations for why dopamine varies during spontaneous behavior, including those mentioned by the reviewer above; we have expanded and clarified our discussion of these possibilities in the revised manuscript.

Minor discussion point: the discussion focuses on DA-modulated corticostriatal activity but thalamic inputs to striatum are also important to evoke - as I'm curious if open field behavior even looks different in decorticate mice. Most of the behaviors (locomotion, grooming, rearing) are known to be subcortically generated and depend on brainstem-thal-striatal- snr-brainstem loops (e.g. Berridge).

Referee #2 (Remarks to the Author):

Dopamine levels in the striatum have long been shown to reinforce task-structured behavior but a role for striatal dopamine signaling in shaping spontaneous, naturalistic behavior has not been demonstrated. Here, Markowitz et al. provide strong correlational, sufficiency, and modeling evidence that fluctuations in dorsolateral striatum (DLS) dopamine levels modulate behavior in the absence of task structure or reward. Specifically, spontaneous behavior is decomposed into sequences of brief syllables and dopamine activity in the DLS is shown to fluctuate during syllable transitions. The peak amplitude of dopamine transients during a syllable predicts future syllable frequency and variability in sequence transitions. Using encoding models, Markowitz et al. predict average dopamine amplitude for a syllable, or dopamine amplitude at a given moment, from features of the behavior that follow. In a compelling experiment, closed-loop optogenetic stimulation of dopamine inputs to the DLS during particular syllables increases the occurrence of those syllables and the entropy of syllable transitions. Finally, Markowitz et al. build reinforcement learning models to test the hypothesis that DLS dopamine serves as a reward signal while increasing syllable variability to preserve behavioral flexibility.

The work is exciting and significantly broadens our understanding of dopamine's role in guiding behavior. The discovery that dopamine promotes the repetition of spontaneous behavioral syllables while also increasing behavioral variability is particularly novel. Below are a few points for the authors to consider:

1) In figure 1, DLS dopamine activity is shown to fluctuate during syllable transitions, often showing a phasic decrease in activity. Yet, in figures 2-4 the authors focus on the peak dopamine level measured during each syllable. The decision to focus on the peak amplitude during each syllable is at odds with figure 1g, which suggests that dopamine signaling is most associated with syllable transitions when there is usually a phasic decrease in signaling.

Why focus on dopamine signaling during specific syllables rather than syllable transitions? And why focus on the peak amplitude of dopamine signaling during syllables rather than on the larger and more prevalent phasic decreases in dopamine signaling? Does the amplitude of phasic dips in dopamine signaling during syllables and syllable transitions bear no consequence on syllable usage?

We thank the reviewer for pointing this out. As mentioned above, this

misunderstanding is the consequence of the specific (and confusing) way in which we z-scored the data in the initially submitted Figure 1, which we have now corrected for clarity. In actuality, nearly all syllable instances are associated with positive DA transients whose amplitudes vary (and whose amplitudes predict syllable usage and variability); we make this clear through additional figures and cartoons in the revised manuscript.

2) Less than half of the mice in figure 4e-f (the “learners”) showed an increased occurrence of syllables that were paired with optogenetic DLS dopamine activity. An explanation for why a majority mice (“non-learners”) did not demonstrate this effect is absent. Also absent is an explanation for why non-learners were excluded from further analyses. Why were mice split into learners and non-learners and what justification permits the exclusion of non-learners from analysis?

As mentioned above, there is a great deal of variability in the ability of animals to learn tasks, and for the most part we have little explanation for why. Given this context, the question raised by the reviewer is both interesting and important — what accounts for the variability in opto-DA reinforcement across mice? To address this question, we took advantage of a subset of mice in which we both recorded and manipulated dopamine in DLS. These analyses revealed that mice that are particularly susceptible to endogenous dopamine fluctuations (in terms of how much syllable reinforcement or sequence variability is induced by endogenous dopamine transients) are also the most sensitive to opto-DA-induced learning. These observations suggest that individual mice vary in their sensitivity to the behavioral effects of DLS dopamine, including effects related to reinforcement and learning. This finding has important implications for our understanding of why some mice “get” reward-driven tasks in the lab, while others do not, and we include a new main text figure exploring these ideas. As mentioned above, this observation has also prompted a whole new line of research in the lab, and so we again thank the reviewer for raising this interesting question.

3) The authors paired optogenetic stimulation of dopamine terminals in the DLS with 6 specific behavioral syllables, reporting generally that these syllables subsequently increased in frequency (figure 4e-f). Can the authors show the relative increases in frequency of the 6 syllables for representative learners (and non-learners)? I’m curious if some syllables are more liable to increases in frequency than others. Is there an underlying difference in the waveform of dopamine transients for syllables that did and did not increase in frequency? A figure identifying the syllables (e.g. rear, pause) that correspond to rows in figure 1g would be informative. I wonder, for example, if the syllables that increased after stimulation are also the syllables in the top few rows of figure 1g.

This is an excellent question, one also raised by Reviewer 1 — are there syllables more or less susceptible to reinforcement by opto-DA? We have done this analysis — the answer is yes, different syllables are differentially reinforced by opto-DA. We also show that this variation does not map onto different types of behavior (with grooms, for example, more reinforced than rears); instead, those syllables that are associated with higher endogenous DA are precisely that are more reinforced in the opto-DA experiments. This finding is consistent with a simple model in which endogenous and opto-DA add together to reinforce the expression of associated syllables. We present these results and the additive model in a revised manuscript, and thank the reviewer for raising this important question.

For mice that increased the frequency of syllables paired with optogenetic DLS dopamine stimulation, were all transitions to the target syllable increased, or only transitions that received optogenetic stimulation?

Our opto-DA stimulation is timed to coincide with individual targeted syllables rather than transitions; in nearly all cases opto-DA starts and terminates within the targeted syllable (as shown in our supplementary data). We show that those transitions into a given syllable that were most likely become even more likely after opto-DA learning is complete; we include these data in the revised manuscript.

Minor corrections:

1) *In figure 1e the right y-axis labels are reversed relative to what is indicated in the figure caption.*

We thank the reviewer for noting this error, which we have fixed.

2) *Figure 2a indicates that the mice are food restricted but the methods (bottom of page 39) indicate that mice were water restricted for 3-5 hours before the task.*

We have clarified this point in the revised manuscript.

3) *In general, the methods are confusingly organized – it is difficult to discern which sections of the methods correspond to which figures.*

We have gone through the methods, and organized them in the order in which they are referred to in the paper itself. We have also adopted the excellent suggestion from this reviewer to include explicit call-outs in the methods to the associated figures

and panels — we plan to do this in all our subsequent papers as well, as it really helps with clarity.

4) *In the discussion, the description of the types of computations that DLS dopamine transients might reflect are confusing. The paragraph at the end of page 14 could be edited for clarity.*

We have revised the part of the discussion in which we articulate different computations that could be supported by DA for clarity.

Referee #3 (Remarks to the Author):

The manuscript by Markowitz et al., report the exciting results of a thorough and carefully conducted study of transient dopamine release in the dorsolateral striatum of mice during spontaneous behavior in a (relatively) cue-free environment. The authors make parallels between the role that dopamine in the striatum plays in the control of behavior in reward- maximization contexts, and the shaping of spontaneous behavior in the absence of external overt reward. They use fiber photometry of dopamine release, combined with their previously published discretization method of ongoing mice behavior into syllables, to show that during unstructured spontaneous behavior, dopamine released in the DLS appears to exert similar long term and short-term effects on its target striatal circuitry to those shown in more structured tasks.

Physiologically, it is well established that dopamine has a dual effect: an immediate effect on excitability of SPNs (differentially impacting direct and indirect

projections from the striatum), and a long-term effect on synaptic plasticity. In this study, the authors demonstrate both effects, and discuss their impact on behavioral choices with terminology derived from the framework of action choice in reinforcement learning.

The authors use the inter-syllable and intra-syllable variability of the amplitude of fluctuations in dLight fluorescence to show a correlation between the amount of dopamine release tied to a particular syllable and its subsequent occurrence over a minutes' time scale, and a negative correlation to overall velocity and to predictability of the following syllable on a seconds' time scale. They further examine causality by closed-loop optogenetic stimulation of dopaminergic axons, triggered upon detection of individual syllables. The described experiments are meticulous, the analysis robust and the manuscript is well-written. The detailed Methods section is particularly commendable, as is the use of explanatory panels in the different figures. My issues, detailed below, have mainly to do with phrasing and interpretation.

1. *First, I am a little confused about the main phenomenon. According to Fig.*

1, syllable-associated responses are primarily reductions in dopamine release (in fact, it appears that on a per-syllable average, syllables are associated with either an elevation of DA release following the syllable transition, or (more frequently) reduction in release preceding transition. In that respect the average trace depicted in Fig. 1f is misleading). On the other hand, the reward-like time course of dLight peaks depicted in Fig. 2b shows clear elevations (as expected from the alignment to the time of the peak). The narrative also seems to implicitly regard DA fluctuations as upward deflections (hence the reinforcement lingo). Could it be that most fluctuations during behavior are not related to syllable transitions per-se, but rather to the syllable-related behavior itself? This is also in line with the authors' use of syllable detection to trigger excitation in dopaminergic axons during the syllables.

We thank the reviewer for pointing this out. As mentioned above, this misunderstanding is the consequence of the specific (and confusing) way in which we z-scored the data in the initially submitted Figure 1, which we have now corrected for clarity. In actuality, nearly all syllable instances are associated with positive DA transients whose amplitudes vary (and whose amplitudes predict syllable usage and variability); we make this clear through additional figures and cartoons in the revised manuscript.

2. The combined results of the free-behavior and optogenetic experiments imply that dopaminergic surges in the DLS serve to reinforce activity in DLS circuits that coincided with this surge. As the authors and others have previously shown, activity in the DLS is tightly connected to the identity of specific behaviors and the vigor with which they are expressed. It therefore stands to reason that these forms of activity, and the associated behaviors, would be reinforced upon elevations of DA and diminished upon DA decreases. In that respect, it is unsurprising that the authors failed to detect a similar connection with DMS DA. It is likely that these fluctuations in DA (which may or may not coincide with those in the DLS – an interesting issue in itself) change the statistics of DMS activity, which would not necessarily be reflected by syllable-related statistics. Unfortunately, it is unknown (as far as I am aware) what DMS SPN activity looks like during spontaneous, task free and cue free behavior. I therefore find extended Fig. 9 unnecessary. However, if the authors do wish to keep DMS data in the paper, I would suggest showing the full distribution of dF/F_0 for different syllables (equivalent to Fig. 1g), as well as a more thorough attempt at characterization of the signal as a function of other variables that may be represented in DMS activity (e.g., action sequences, spatial variables (allocentric or egocentric), etc.

There was something of a difference of opinions amongst the reviewers, with another reviewer advocating for moving the DMS panels into the main figures. After much thought and additional experiments, we have decided for clarity to include as a

supplemental figure a more detailed analysis of average endogenous DMS dopamine-syllable relationships. To perform this analysis, we nearly tripled the size of our DMS recording dataset. The results reveal that the timescale at which dopamine fluctuates in DMS is slower than in DLS, and that DMS waveforms are broader than in DLS; similar results have recently been reported on bioRxiv by Josh Burke. Because of this difference in timescales, average syllable-associated DMS dopamine amplitudes are much smaller than in DLS and do not predict syllable usage in the future. Taken together, these data argue that thinking about “syllable-associated” dopamine fluctuations in DMS isn’t really possible in a meaningful sense, because there are fewer dopamine transients in DMS, and those that are present do not fluctuate at the syllable timescale. Because of the differences in timescales and waveforms between DLS and DMS, we have decided to remove our opto-DA experiments in DMS, which were difficult to interpret because limiting opto-DA stimulation to a single targeted syllable requires us to use brief DLS-like stimulation conditions. We have instead performed a full analysis of the relationship between endogenous DMS dopamine and continuous kinematic variables (mostly for reference as requested by this reviewer). We have generated a new supplemental figure which we include in our revised manuscript that details these results.

3. I am curious about the authors’ thoughts about the nature of the DA signal in the context of spontaneous exploratory behavior. Is it merely random noise that hijacks the reinforcement-learning circuitry, thereby causing similar effects, or does the syllable-associated release encode a meaningful parameter?

We discuss the possible origins of the DA fluctuations we observe in our revised discussion, including the possibility — as this reviewer mentions — that it is noise.

4. I find the description of the RL model in Fig 5 lacking. Specifically, I do not understand which parameters were estimated in each model version, and how models with different numbers of parameters were compared. Generally, I like the notion of leveraging the dopamine signal to implement both the reinforcement and exploration elements of RL on two different timescales. As I mentioned above, it is in line with well-known physiological properties of DA innervation to the corticostriatal circuitry. However, I find the Q-learning model depicted in Fig. 5 to cause more confusion than add insight. If I understood the model correctly, the basic model does not contain an effective temperature term in the action selection function. If this is indeed the case, it is unsurprising that large rewards would lead the model to perseverate, and that introducing stochasticity to the action selection process would implement exploration, thereby rescuing behavior.

The analysis presented in Fig. 5f-g does not convincingly show that another method of dynamically changing

tau that is unrelated to dopamine, or even simply tuning a constant temperature parameter to an intermediate value, would be inferior. Finally, I am not sure what to make of the finding that the DA transients correspond to the reward term rather than RPE (in the current formulation of the model), and I do not know how to interpret differences in z-scored log-likelihoods. However, it seems intuitively likely that the inability to detect RPE-like activity may have something to do with the fact that the DA signals were sampled randomly per syllable, and therefore could not track trial to trial fluctuations in prediction error.

We thank the reviewer for raising these important concerns. There are three issues here. First, we agree with the reviewer that the perseveration analysis is too susceptible to parameter engineering, so we have eliminated it from the paper. Second, we failed to effectively describe the model in the main text and methods, which does include a temperature parameter that is learned from the data; we apologize for our confusing description of this, which we now clarify in the revised main text. Third, per the reviewer's suggestion, we have modified the model such that the training phase is not based upon random sampling but instead on actual syllable sequences and the ensuing dopamine transients. This modified model recapitulates what we initially reported, and we include this modified version of the model in the revised manuscript. In sum we have simplified our description and use of the model to highlight just the essential point — that an RL model trained on syllable sequences and associated dopamine transients generates an output consistent with mice interpreting DLS dopamine fluctuations as a reward-like signal — and have eliminated other confusing aspects of this analysis. We think this clarifies the importance of these results, and we thank the reviewer for encouraging us to do this.

Minor:

1. *Why are the actual dLight values in Fig. 3a larger than in Fig. 2?*

We thank the reviewer for catching this – it was a mistake in the legend, which we have now corrected.

2. *Legend to Fig. 3j – correlation between ...?*

We thank the reviewer for catching this error, which we have fixed in the revised manuscript.

3. *I am not convinced that the inability of a classifier to distinguish between opto-stimulated syllables and spontaneous syllables is sufficient proof that opto-stimulation did not affect kinematic properties (extended Fig. 8C)*

In addition to the classifier, we show extensive data demonstrating directly that we could not observe any change in kinematics (e.g., velocity, acceleration) associated with opto-stimulation; we emphasize these complementary analyses in the revised text.

Reviewer Reports on the First Revision:

Referees' comments:

Referee #1 (Remarks to the Author):

Markowitz et al provide a strong resubmission that substantially addressed each one of our major and minor issues.

First, by taking a deeper dive into trial-by-trial correlations between DA and behavior, they address our main issue of differential conditionability of syllables. These new analyses rule out the major concern - that the correlations evident on average were simply driven by a small number of small correlations.

Second, by computing indices for DA-behavioral correlation on a syllable- and mice-specific level, we now see a satisfying and profoundly important relationship between DA signals and how well mice learn in the optoDA experiments. We agree with the authors that this is a substantial new discovery with promising future research directions.

Third, the authors have done a substantial amount of analytical and experimental work to address our questions about the relationship between DA transient variance and syllable variance, including analysis of forelimb kinematics. Though the results of these analyses were mostly negative, we think they fortify the paper as - in the past version - some fraction of the readership would likely have (misattributed) their transients to these kinematic variations.

Fourth, the authors have dramatically expanded their DMS dataset, and have replicated recent findings about different timescales of transients there and why this brain region is weakly, if at all, correlated with syllables and transitions. Given the amount of work here, and the largely negative results, we raise the possibility that the paper could be published without them - if the authors wanted to re-package these findings for a different (and likely lower impact) publication. In other words, we now agree with the other reviewer that the paper is publishable without the DMS data if the authors choose this path in order to secure an additional publication.

Finally, ample technical details are provided on the closed loop tracking that strengthen all of the associated findings, and minor requests for clarification in the discussion were addressed.

Overall this is a very important and timely paper and should be published with minimal delay.

Referee #2 (Remarks to the Author):

The authors addressed all of my concerns. The manuscript is improved and now clearly highlights the most novel and important findings. The results are exciting and significantly broaden our understanding of dopamine's role in guiding behavior. I fully support publication.

Referee #3 (Remarks to the Author):

In their revised manuscript, Markowitz et al. address most of the concerns raised by myself and the other reviewers. I am particularly excited about the new figure 4. I think it has the potential to provide much insight into the dynamics of spontaneous behavior on different time scales. In particular, the relation between average amplitude of dopamine transients on catch trials and the ease with which these syllables lend themselves to being reinforced by opto-DA is very impressive. I also wish to thank the authors for the additional data of DMS recordings. I list below some points which I am still uneasy about.

First, I still remain puzzled about the analysis of the transients and its presentation in the various figures.

1. Depiction of example dLight traces: As the authors explain, the impression of negative dopamine responses in Figure 1 and extended Figure 4 stem from the z-scoring procedure they employed. As I understand it, in these figures the traces are z-scored for each short 3 sec frame shown in the figure. This indeed makes it likely that every peak of a substantial duration will be balanced out by a negative deflection. However, judging from Extended Fig. 4, it seems that a large proportion of syllables are associated with significant reductions in the dLight signal, that seem better aligned with the stimulus onset than the peaks. As we know from responses of dopamine neurons in primates and rodents in structured reward-related tasks, as well as from theoretical work, pauses in dopamine neuron firing during omission of rewards, for example, carry information as well, and it seems wrong not to consider those. It may be found, for example, that these responses show the flip side of the reinforcement: that syllables which are associated with reduced dopamine would decrease their usage.

2. Analysis of the correlation of dLight magnitude and various behavioral measures: If I understand correctly, for all the variant of this type of analysis, the dLight signal was taken differently: it was z scored in 20 sec sliding windows, and the magnitude of the signal was then defined as the maximum value within the 300 ms after syllable onset. If this is so, then indeed the troughs are not taken into consideration in analysis at all. Apart from missing out on the potential of the negative responses, this poses a methodological problem. In cases where indeed the syllable is associated with a reduced signal, it may be that the maximum value taken here is just noise. Consider, for example, the light-green traces in Fig. 2f or 2m. Which value would be taken for those? This is particularly evident in Extended Fig. 4e: The bottom syllables have no discernable temporally aligned peaks, and yet the points reflecting the mean syllable associated peak, depicted as green dots on the right, show substantial scores of a completely different scale.

3. The relation between dopamine, usage and entropy. According to the results of this paper, these variables are inter-correlated. It would be useful to give the readers an overall sense of these. I would like to see a visual depiction of the relationship between entropy and frequency of use, on average per syllable, and perhaps the corresponding mean dLight signals (similar to Fig. 4f shown for the optoDA experiments for 6 syllables).

4. The role of velocity and other kinematic parameters. The narrative in the paragraph relating to the encoding and decoding analysis (Fig. 2, highlights syllable usage and variability. However, it appears that velocity adds a substantial amount of explanatory power to the prediction. This information is hidden in the extended data and not highlighted enough in the main text. Please make a note of this there. This is particularly striking when one notices that although DMS dopamine does not encode syllable usage or entropy, the ability to predict dLight in DMS is not significantly different from DLS.

Minor:

1. Extended Figs. 2f,g, refer to a single example experiment. Please provide this information in the beginning of the legend to f, not the end of g.
2. The legend to Extended Fig. 4a is very cumbersome. Please clarify that the state-transition precedes the dopamine peak by 230 ms.
3. Extended Fig. 4e. Is the color scale correct? The values seem very low, compared to what we're used to seeing (and to the values on the right).
4. RL model (Figure 5). Why is the transition matrix z-scored? These are probabilities, all in the [0,1] range. What are they z-scored over?
5. Discussion, first paragraph: Although I like in principle the idea that dopamine enables DLS to dynamically assemble contextually appropriate behavioral sequences, the model and findings have nothing to do with contextual appropriateness
6. Methods: dLight average waveform z-scoring - 'to account for differences in the number of trials in each average...' what are trials here?
7. Extended Figure 8b: legend says DNN, figure says CNN
8. Extended figure 8d: Please add some ticks on the time axis. It would be useful to get an idea of the actual latency

Author Rebuttals to First Revision:

We thank the reviewers for their support of our revised manuscript, and Reviewer #3 for his/her additional comments. We have comprehensively addressed these concerns, which have helped to further clarify and improve the paper. We hope that with the inclusion of these revisions the manuscript will now be acceptable for publication. We include a point-by-point response to Reviewer #3's comments below.

In their revised manuscript, Markowitz et al. address most of the concerns raised by myself and the other reviewers. I am particularly excited about the new figure 4. I think it has the potential to provide much insight into the dynamics of spontaneous behavior on different time scales. In particular, the relation between average amplitude of dopamine transients on catch trials and the ease with which these syllables lend themselves to being reinforced by opto-DA is very impressive.

I also wish to thank the authors for the additional data of DMS recordings. I list below some points which I am still uneasy about.

First, I still remain puzzled about the analysis of the transients and its presentation in the various figures.

1. Depiction of example dLight traces: As the authors explain, the impression of negative dopamine responses in Figure 1 and extended Figure 4 stem from the z-scoring procedure they employed. As I understand it, in these figures the traces are z-scored for each short 3 sec frame shown in the figure.

Throughout the paper and in the figures in question we Z score the data using 20 second bins (10 seconds before and after each transition). We now clarify this in the figure legends and methods.

This indeed makes it likely that every peak of a substantial duration will be balanced out by a negative deflection. However, judging from Extended Fig. 4, it seems that a large proportion of syllables are associated with significant reductions in the dLight signal, that seem better aligned with the stimulus onset than the peaks. As we know from responses of dopamine neurons in primates and rodents in structured reward-related tasks, as well as from theoretical work, pauses in dopamine neuron firing during omission of rewards, for example, carry information as well, and it seems wrong not to consider those. It may be found, for example, that these responses show the flip side of the reinforcement: that syllables which are associated with reduced dopamine would decrease their usage.

We thank the reviewer for raising this question, which speaks to the dopamine dynamics coincident with syllable transitions. We first asked whether or not the dip in dopamine is “real” – that is, is this dip a consequence of the way in which we normalized or plotted the data, or is it a real and pervasive fluctuation present in the data itself? We have done a number of analyses (most of which we will not discuss in detail here, but some of which are included in the revised manuscript) consistent with the idea that indeed there is a systematic dip in dopamine that occurs near the beginning of each transition. We say “near” this transition because the dopamine photometry signal is a lagging indicator (by 10s of milliseconds; Extended Data Figure

1e argues the number is about 70 ms); given that dopamine is minimized near time “zero,” it is likely that the nadir of the dip is associated with the end of the previous syllable (rather than strictly being associated with the beginning of the new syllable).

What might account for the ends of syllables being associated with a dopamine dip? Given our observation that each syllable is also associated with a dopamine peak (whose amplitude varies), the most reasonable explanation for the dip phenomenon is the burst-then-pause dopamine dynamics that have been described previously in association with various with movements (which we now refer to in the revised manuscript). Explicitly addressing whether dopamine dynamics associated with syllables exhibit a burst then a pause requires us to normalize time, given that each syllable varies in terms of its length. Indeed, if we subject our data to time warping (so the X axis is no longer seconds, but rather syllables) we observe that each syllable starts with low dopamine, followed by a dopamine peak, and then — importantly — that peak falls back to baseline before the initiation of the next syllable (again, not accounting for photometry lags). We include this time-warped data now in the revised paper, along with a plot of the derivative of the photometry signal, which shows that the transition between the pause and the burst (which we expect accompanies the actual syllable transition) is the moment at which dopamine changes the most during the experiment. We think our revisions better describe the pattern of dopamine fluctuations, center our thinking about this pattern in terms of the published literature, and convincingly demonstrate that there is a peak within each syllable.

2. Analysis of the correlation of dLight magnitude and various behavioral measures: If I understand correctly, for all the variant of this type of analysis, the dLight signal was taken differently: it was z scored in 20 sec sliding windows, and the magnitude of the signal was then defined as the maximum value within the 300 ms after syllable onset. If this is so, then indeed the troughs are not taken into consideration in analysis at all.

To address this concern, we recomputed dopamine-behavior correlations not just considering the amplitude of the peak, but rather the difference between the pause and the peak. As shown here (Response Figure 1), using this revised metric did not improve the correlation

between dopamine and either syllable usage or sequence entropy. Consistent with this finding, if we consider only the depth of the pause occurring near each syllable alone, we observe only negligible correlations (either positive or negative) with syllable usage (data not shown). Given that the dopamine trough is likely associated with the end of the previous syllable, these results suggest that the most relevant event for predicting usage/entropy is the dopamine peak associated with each syllable. To keep the paper as simple as possible we chose not to include these analyses in the paper, but instead have modified the main text to clarify that the pause occurs before syllable initiation given the imaging lag (and as mentioned above include the warping figures to show the actual dynamics).

Apart from missing out on the potential of the negative responses, this poses a methodological problem. In cases where indeed the syllable is associated with a reduced signal, it may be that the maximum value taken here is just π noise. Consider, for example, the light-green traces in Fig. 2f or 2m. Which value would be taken for those? This is particularly evident in Extended Fig. 4e: The bottom syllables have no discernable temporally aligned peaks, and yet the points reflecting the mean syllable associated peak, depicted as green dots on the right, show substantial scores of a completely different scale.

Dopamine exhibits a peak within each syllable, even for syllables that are at the bottom end of the average dopamine distribution, because dopamine starts off lower than average during each syllable (due to the previously discussed dip dynamics). Thus for each syllable there is a real rise in dopamine; for low dopamine syllables this means dopamine rises to something close to the average level of dopamine, which is associated with a z score near zero. Importantly, this does not mean there wasn't a rise in dopamine during the syllable, just that this rise is less than that observed for other syllables. We thank the reviewer for catching an error we made in plotting, which added to the confusion – the green points in the submitted manuscript was actually averaged in a different way than the adjacent plot (we were juggling lots of averaging by animals, by sessions, and by syllables to address a reviewer concern, and made an inadvertent mistake in what we plotted); to reconcile this we plot syllable/mouse pairs in both sides of this plot, which now makes sense.

3. The relation between dopamine, usage and entropy. According to the results of this paper, these variables are inter-correlated. It would be useful to give the readers an overall sense of these. I would like to see a visual depiction of the relationship between entropy and frequency of use, on average per syllable, and perhaps the corresponding mean dLight signals (similar to Fig. 4f shown for the optoDA experiments for 6 syllables.

We have performed the requested analyses, and included them as a new Extended Data figure (which, as predicted, reveals that dopamine, usage, and entropy are all correlated to some extent). We thank the reviewer for this request, as this is a really useful plot for another reason — it points out the value in our modeling, which allows us to ask about the *independent* contributions of syllable usage and entropy to the dopamine signal. As expected (and reported in the prior version of the manuscript)

both syllable usage and sequence entropy independently contribute information useful to predicting dopamine dynamics from behavior. We now explicitly point this out in the revised manuscript.

4. The role of velocity and other kinematic parameters. The narrative in the paragraph relating to the encoding and decoding analysis (Fig. 2, highlights syllable usage and variability. However, it appears that velocity adds a substantial amount of explanatory power to the prediction. This information is hidden in the extended data and not highlighted enough in the main text. Please make a note of this there. This is particularly striking when one notices that although DMS dopamine does not encode syllable usage or entropy, the ability to predict dLight in DMS is not significantly different from DLS.

As requested we have revised the main text to point this out.

Minor:

1. Extended Figs. 2f,g, refer to a single example experiment. Please provide this information in the beginning of the legend to f, not the end of g.

We have now clarified this in the legend.

2. The legend to Extended Fig. 4a is very cumbersome. Please clarify that the state-transition precedes the dopamine peak by 230 ms.

We have rewritten the legend to Ext. Fig. 4a, and included a description of the relationship between syllable transitions and dopamine peaks. Note that the most likely relationship occurs at 200 ms (which does not account for photometry lags; note this number has slightly changed due to us switching from the 97.5 percentile – where we calculated 230 ms – to 90th, 95th, and 99th per a reviewer request, all of which are 200 ms. Of course these differences are in the noise, we just wanted to clarify where the slight change in value came from).

3. Extended Fig. 4e. Is the color scale correct? The values seem very low, compared to what we're used to seeing (and to the values on the right).

We thank the reviewer for pointing this out; as noted above we had made an averaging error in the submitted figure — the revised panel now makes sense.

4. RL model (Figure 5). Why is the transition matrix z-scored? These are probabilities, all in the [0,1] range. What are they z-scored over?

We agree there is no need to z score these probabilities, and we now depict them directly as suggested.

5. Discussion, first paragraph: Although I like in principle the idea that dopamine enables DLS to dynamically assemble contextually appropriate behavioral sequences, the model and findings have nothing to do with contextual appropriateness

We agree, and have deleted that phrasing.

6. Methods: dLight average waveform z-scoring - 'to account for differences in the number of trials in each average...' what are trials here?

Here we are referring to syllable instances; we have clarified that in the revised manuscript.

7. Extended Figure 8b: legend says DNN, figure says CNN

We have corrected this error.

8. Extended figure 8d: Please add some ticks on the time axis. It would be useful to get an idea of the actual latency

We have added the requested tics.

Reviewer Reports on the Second Revision:

Referees' comments:

Referee #3 (Remarks to the Author):

The last revision has alleviated all of my concerns.

This is an excellent and exciting paper. It provides important insights into the molding of continuous everyday behaviors, highlighting novel functions for dopamine - at the same time expanding and solidifying the behavioral repertoire. These findings complement our understanding of dopamine's role in goal-directed behavior, while fitting beautifully with known cellular and network properties of dopamine in the basal ganglia.

I look forward to seeing the paper in print.

Author Rebuttals to Second Revision:

We again thank the reviewers for their support of our manuscript and their useful comments.